# ML4CO-Bench-101: Benchmark Machine Learning for Classic Combinatorial Problems on Graphs

**Jiale Ma**[12], **Wenzheng Pan**[1], **Yang Li**[12], **Junchi Yan**[12*]
[1]Sch. of Computer Science & Sch. of Artificial Intelligence, Shanghai Jiao Tong University
[2]Shanghai Innovation Institute.
{heatingma,pwz1121,yanglily,yanjunchi}@sjtu.edu.cn

## Abstract

Combinatorial problems on graphs have attracted extensive efforts from the machine learning community over the past decade. Despite notable progress in this area under the umbrella of ML4CO, a comprehensive categorization, unified reproducibility, and transparent evaluation protocols are still lacking for the emerging and immense pool of neural CO solvers. In this paper, we establish a modular and streamlined framework benchmarking prevalent neural CO methods, dissecting their design choices via a tri-leveled "paradigm-model-learning" taxonomy to better characterize different approaches. Further, we integrate their shared features and respective strengths to form 3 unified solvers representing global prediction (GP), local construction (LC), and adaptive expansion (AE) mannered neural solvers. We also collate a total of 65 datasets for 7 mainstream CO problems (including both edge-oriented tasks: TSP, ATSP, CVRP, as well as node-oriented: MIS, MCl, MVC, MCut) across scales to facilitate more comparable results among literature. Extensive experiments upon our benchmark reveal a fair and exact performance exhibition indicative of the raw contribution of the learning components in each method, rethinking and insisting that pre- and post-inference heuristic tricks are not supposed to compensate for sub-par capability of the data-driven counterparts. Under this unified benchmark, an up-to-date replication of typical ML4CO methods is maintained, hoping to provide convenient reference and insightful guidelines for both engineering development and academic exploration of the ML4CO community in the future. Code is available at https://github.com/Thinklab-SJTU/ML4CO-Bench-101, and the dataset is at https://huggingface.co/datasets/ML4CO/ML4CO-Bench-101-SL.

## 1 Introduction

Combinatorial optimization (often in the form of edge- or node-oriented tasks on graphs) plays a pivotal role in operations research, with wide-ranging applications in logistics systems [1], transportation planning [2], supply chain management [3], and network design [4], where discrete decision-making is essential for optimizing operations and reducing costs. Traditional approaches typically rely on mathematical programming [5] or heuristic methods [6, 7] to obtain exact or approximate solutions. Recently, machine learning (ML) has emerged as a powerful paradigm for tackling combinatorial optimization problems (COPs), offering data-driven efficiency and near-optimal solutions [8, 9, 10, 11, 12, 13], which has fostered the growth of *Machine Learning for Combinatorial Optimization* (ML4CO) as a distinct research community [14, 15]. Despite the promising advancements, the ML4CO community faces critical challenges regarding model design, reproducibility, inconsistent datasets, heterogeneous evaluation and reporting protocols, thus highlighting the pressing need for a comprehensive, standardized, and user-friendly benchmark.

---

*Correspondence to: Junchi Yan. This work was partly supported by NSFC (92370201).

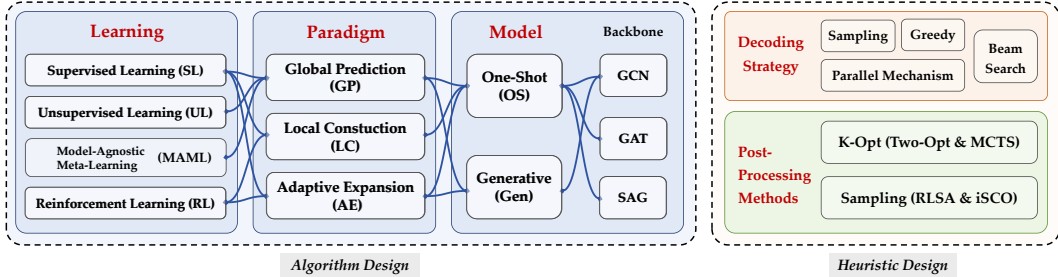

Figure 1: Overview of our proposed ML4CO-Bench-101.

Specifically, major issues lie in the following aspects. **1) Methodological fragmentation.** Existing works lack a unified taxonomy, with baselines inconsistently categorized, e.g., by learning type (SL: supervised learning, RL: reinforcement learning, UL: unsupervised learning, MAML: Model-Agnostic Meta-Learning, etc.), construction method [16] (AR: autoregressive, NAR: non-autoregressive), or backbone architecture [17] (GCN, GAT, SAG, etc.). This obscures comparative insights and impedes systematic advancements. **2) Implementation Inconsistency.** Reproducibility is hampered by ad-hoc implementations. For instance, training and testing datasets vary in format (TSPLIB file, pickle file, txt file, etc.) or framework (PyG objects, DGL objects, raw PyTorch tensors, etc.), creating redundant overhead for replication and extension. Also, a number of works [18, 19, 20, 21] incorporate complicated environment dependencies or intricately nested code modules, which hinder the applicability of transferring the approaches to new tasks. Moreover, the decoding heuristics, as simple as the greedy algorithm, evolve and vary from one work to another, however, an easy reuse of the notations (e.g., "SL+Grdy") is not supposed to conceal inherent performance variation from readers. **3) Evaluation irregularity.** A standardized protocol for evaluating CO problems remains absent, and current results suffer from: **i) Dataset heterogeneity.** Beyond formatting discrepancies, some works often use arbitrary problem settings and self-generated synthetic data. For instance, [22] experimented on BA [23] graphs for MVC and MCut, [24] merely tested MCl and MIS on real-world datasets; [19, 20, 25, 26] all conducted tests on RB [27] graphs but for different tasks and with different settings of node ranges. This makes it nearly infeasible to ensure model-to-model fair comparison. **ii) Post-processing abuse.** Techniques like MCTS are widely adopted to further improve the solution quality for TSP [18, 28, 29]. While effective, these methods are under inconsistent settings (e.g., varying time limits) across literature and often mask raw model performance, i.e., leading distinct models to level off w.r.t the optimality gap and possibly skewing their conclusions. **iii) Uncontrolled randomness.** Many sequential methods inherently exploit multi-sampling to improve diversity of solution [10, 30, 31] or selectively reports "the best trial among 8 seeds" [20]. Scant attention has been paid to these subtle tricks in literature, leaving ambiguity and unfairness in comparing time and solving quality. **iv) Pipeline discrepancy.** Inconsistent evaluation protocols, e.g., storing heatmaps offline for subsequent searching [18] v.s. generating heatmaps and decoding for each instance sequentially (widely employed), further undermine comparability. More delicately, different pre-processing techniques can have been operated (often without explicit introduction), e.g. generating Dirac delta distributions for graph tasks [20], re-permuting vertex indices [32, 33] and instance-level re-normalization [34, 35] of distance matrices for TSP, etc.

To address the aforementioned shortcomings, we have provided a systematic review of existing ML4CO work, and then developed a modular code framework with a new benchmark in the hope of advocating for a standardized assessment that embraces methodological diversity while ensuring evaluation transparency. Our contributions in this work can be summarized as follows:

- We provide a comprehensive retrospective of existing ML4CO literature, and introduce a benchmark suite, which encompasses standard datasets as well as our completed, refined and concentrated implementation of neural solvers for COPs, on the basis of existing methods under our categorization.

- Extensive experiments upon the benchmarking toolkit provide insights into critical research questions: 1) For edge-oriented tasks (TSP, ATSP and CVRP), SL usually outperforms UL and MAML, especially as the scale increases. 2) For node-oriented tasks (MCl, MCut, MIS and MVC), which can be reformulated as energy minimization problems [19, 26], making them particularly suitable for UL. Furthermore, MAML exhibits strong generalization capabilities on these problems.

- We reproduce a wide range of decoding strategies as well as post-processing methods, and we report the best-performing ML4CO approaches for each benchmark dataset in Table 8. Notably,

ML4CO methods outperform traditional solvers on several benchmark datasets, with the most neural-based approaches demonstrating significant advantages in speed. To support future research in the community, we also release, for the first time, a comprehensive ML4CO dataset with full solution results covering all benchmark datasets (provided in the data link in the abstract).

## 2 Benchmarking Existing ML4CO Methods

**Problem Definition.** The combinatorial optimization problems (COPs) studied in this paper can be consistently represented using the graph structure, i.e., $G = (\mathcal{V}, \mathcal{E})$, where $\mathcal{V} = \{1, \cdots, N\}$ denotes the node set and $\mathcal{E}$ the edge set. Following previous works [16, 36, 37], we define the decision variables $\mathbf{x} = \{0, 1\}^M$ to represent the solution of a given COP. For MCl, MIS, and MVC, we have $M = N$ and $\mathbf{x}_i$ indicates whether the node $i$ is selected. For MCut, $M = N$ and $\mathbf{x}_i$ indicates which subset the node $i$ belongs to. For TSP, ATSP and CVRP, $M = N^2$ and $\mathbf{x}_{i \cdot N + j}$ indicates whether the edge $(i, j)$ is selected. Mathematically, the goal is to find optimal solution $\mathbf{x}^* = \underset{\mathbf{x} \in \Omega}{\operatorname{argmin}}\, c(\mathbf{x}, G)$, where $\Omega$ denotes the feasible set of $\mathbf{x}$ that satisfy the constraints and $c(\cdot, \cdot)$ is the objective cost function of corresponding problems. For better readability of our principal line, we defer the formal definitions of the seven COPs (TSP, ATSP, CVRP, MIS, MCl, MVC, and MCut) to Appendix A.

In this section, we provide a detailed analysis of existing methods of ML4CO from three perspectives: a) *algorithm design*, b) *decoding strategy*, and c) *post-processing optimization*, as depicted in Fig. 1.

### 2.1 Algorithm Designs

Over the past decade, substantial efforts have been dedicated to exploring the potential of data-driven learning for CO, resulting in a rich yet slightly disarrayed body of literature. Hence, we first draw a categorized summary of existing solvers along with three dimensions: *paradigm*, *model*, and *learning*.

**Paradigm.** Following COExpander [37], we regard the solving of COPs as *Variable Determination Process* VDP$(N, k)$, where $N$ denotes the number of decision variables and $k$ denotes the number of iterations required to obtain the complete solution. Based on $N$ and $k$, the existing NCO solvers can be categorized into three paradigms: 1) $k = 1$: **Global Prediction (GP)** solvers [16, 36, 38, 39, 29, 18, 20, 28, 40, 33, 19, 21] leverage neural networks to globally predict the likelihood of each node or edge being selected and then decode the probability heatmaps using heuristic methods (such as simple greedy algorithms) to obtain solutions. 2) $k = N$: **Local Construction (LC)** solvers [41, 31, 33, 42] use neural networks to iteratively predict the best next-step action (e.g., the next node to select) based on the current state until a complete solution is constructed. 3) $k \in (1, N)$: **Adaptive Expansion (AE)** solvers [43, 37, 34] lie in the between of GP and LC. In each iteration, AE solvers utilize neural networks to globally predict probability heatmaps based on the current state, and then determine an adaptive set of decision variables (e.g., until violation of constraints) according to the heatmaps.

**Model.** Whereas the *paradigm* is a higher-level concept agnostic to model implementation, the *model* denotes the specific neural approach employed to undertake the data-driven learning. We categorize existing neural models into two primary classes: 1) **One-Shot (OS)** models achieve results via a single inference pass. This category encompasses models such as Graph Convolution Network (GCN) models [38, 39] and Graph Attention (GAT) models [41, 31, 33]. GCN employs convolutional operations on graphs to aggregate information from neighboring nodes and utilizes node embeddings to capture structural dependencies, while GAT leverages attention mechanisms to dynamically weight node interactions and enable adaptive feature aggregation. These models enhance graph-based learning by capturing complex dependencies and are highly effective for combinatorial optimization and structured decision-making tasks. 2) **Generative (Gen)** models [47, 48, 49] generate results through a progressive denoising process starting from random noise. This category includes Diffusion models [16, 36, 19, 50] and Consistency models [51, 40, 37]. Diffusion models consist of a sequential noising process that transforms feasible solutions into noised vectors and a learnable reverse denoising process, while consistency models converge different noise trajectories of the same graph to the same initial solution, thus significantly reducing the number of denoising steps required. These models leverage the strong expressive power of generative models with the iterative noise addition and removal processes, which significantly improve the prediction quality compared to OS models.

**Learning.** The learning methods of models have long been a central focus of research. A substantial body of works [16, 39, 40] supervised learning (SL), where models are trained on large collections

Table 1: Categorization of methods: Global Prediction (GP), Local Construction (LC), Adaptive Expansion (AE), Unsupervised/Supervised Learning (U/SL). Model-Agnostic Meta-Learning (MAML).

| ID | METHOD | ALGORITHM DESIGN | | | INVOLVED PROBLEMS | | | | | | |
|---|---|---|---|---|---|---|---|---|---|---|---|
| | | PARADIGM | MODEL | LEARNING | TSP | ATSP | CVRP | MIS | MCL | MVC | MCUT |
| 1 | Intel [38] | GP | One-Shot | SL | – | – | – | ✔ | – | – | – |
| 2 | GCN [39] | GP | One-Shot | SL | ✔ | – | – | – | – | – | – |
| 3 | Att-GCN [29] | GP | One-Shot | SL | ✔ | – | – | – | – | – | – |
| 4 | GNNGLS [44] | GP | One-Shot | SL | ✔ | – | – | – | – | – | – |
| * | GP4CO (Ours) | GP | One-Shot | SL | ✔ | ✔ | – | ✔ | ✔ | ✔ | ✔ |
| 5 | DIMES [18] | GP | One-Shot | MAML (RL) | ✔ | – | – | ✔ | – | – | – |
| 6 | Meta-EGN [20] | GP | One-Shot | MAML (UL) | – | – | – | – | ✔ | ✔ | – |
| * | GP4CO (Ours) | GP | One-Shot | MAML | ✔ | – | – | ✔ | ✔ | ✔ | ✔ |
| 7 | UTSP [28] | GP | One-Shot | UL | ✔ | – | – | – | – | – | – |
| 8 | VAG-CO [26] | GP | One-Shot | UL | – | – | – | ✔ | ✔ | ✔ | ✔ |
| * | GP4CO (Ours) | GP | One-Shot | UL | ✔ | – | – | ✔ | ✔ | ✔ | ✔ |
| 9 | DIFUSCO [16] | GP | Generative | SL | ✔ | – | – | ✔ | – | – | – |
| 10 | T2T [36] & Fast-T2T [40] | GP | Generative | SL | ✔ | – | – | ✔ | – | – | – |
| 11 | UniCO-MatDIFFNet [33] | GP | Generative | SL | ✔ | ✔ | – | – | – | ✔ | – |
| * | GP4CO (Ours) | GP | Generative | SL | ✔ | ✔ | ✔ | ✔ | ✔ | ✔ | ✔ |
| 12 | DiffUCO [19] | GP | Generative | UL | – | – | – | ✔ | ✔ | – | ✔ |
| 13 | SDDS [21] | GP | Generative | UL | – | – | – | ✔ | ✔ | – | ✔ |
| * | GP4CO (Ours) | GP | Generative | UL | – | – | – | ✔ | ✔ | – | ✔ |
| 14 | RL4CO [41, 45, 30, 42] | LC | One-Shot | RL | ✔ | – | ✔ | – | – | – | – |
| 15 | MatNet [31] | LC | One-Shot | RL | – | ✔ | – | – | – | – | – |
| 16 | UniCO-MatPOENet [33] | LC | One-Shot | RL | ✔ | ✔ | – | – | – | – | – |
| * | LC4CO (Ours) | LC | One-Shot | RL | ✔ | ✔ | ✔ | – | – | – | – |
| 17 | BQ-NCO [46] | LC | One-Shot | SL | ✔ | ✔ | ✔ | – | – | – | – |
| 18 | GOAL [32] | LC | One-Shot | SL | – | ✔ | ✔ | ✔ | – | ✔ | – |
| * | LC4CO (Ours) | LC | One-Shot | SL | ✔ | ✔ | ✔ | ✔ | ✔ | ✔ | – |
| 19 | LwD [43] | AE | One-Shot | RL | – | – | – | ✔ | – | – | – |
| * | AE4CO (Ours) | AE | One-Shot | RL | – | – | – | ✔ | – | – | – |
| 20 | COExpander [37] | AE | Generative | SL | ✔ | ✔ | ✔ | ✔ | ✔ | ✔ | ✔ |
| * | AE4CO (Ours) | AE | Generative | SL | ✔ | ✔ | ✔ | ✔ | ✔ | ✔ | ✔ |

of labeled data to learn underlying patterns and relationships. However, due to the high cost of obtaining supervised labels for combinatorial optimization problems, several studies [31, 20, 28] explore alternative approaches such as meta-learning (e.g., Model-Agnostic Meta-Learning, MAML), unsupervised learning (UL), and reinforcement learning (RL) to reduce reliance on labeled data.

**Benchmark.** To establish a comprehensive benchmark, we first compile a collection of representative papers from the past decade that apply machine learning techniques to the seven related COPs. Each paper is then carefully classified and annotated based on both the underlying algorithmic design and the specific COPs it addresses. To better structure and communicate the diversity of methodologies, we further group these approaches into three overarching categories according to their solving paradigms, namely *GP4CO*, *LC4CO* and *AE4CO* (**these are categories, not new methods**). During our reproduction and evaluation process, we discover that the applicability of some methods can be extended beyond the problem domains originally claimed in the primary publications. While a full exploration of such extension is beyond the scope of this work, we selectively broaden the evaluation of some promising methods whose cross-domain potential merits further investigation, as delineated in Table 1 for quick references.

## 2.2 Decoding Strategies

We define the *decoding strategy* as the way to determine the decision variables, which mainly includes *greedy decoding*, *sampling decoding*, *beam search* and *parallel mechanisms*.

**Greedy Decoding.** We begin by evaluating the scores $s$ for each decision variable $\mathbf{x}$, based on the prediction output $y$ (probablized) from the neural model. In most cases discussed in this paper, we set $s = y$. However, for MVC, we define $s = -y$, and for MCut, we use $s = |y - 0.5|$. Variables with higher scores are then prioritized (as long as feasible) to constitute the final solution.

**Sampling Decoding.** Instead of always selecting the highest-scoring variable, it samples decision variables according to a probability distribution. Specifically, the probability of selecting variable $\mathbf{x}_i$ is $P(\mathbf{x}_i) = s_i / \sum_{i=1}^{N} s_i$. This approach promotes diversity and may yield better overall solutions.

**Beam Search.** This strategy strikes a balance between efficiency and solution quality. It maintains a set of candidate sequences, with a fixed size $k$ (16 in this paper). At each step, the algorithm expands these candidate sequences by considering all possible extensions and then retains only the top-$k$ sequences with the highest scores. This method allows for a more comprehensive exploration of the solution space compared to greedy decoding, while still keeping the computational cost manageable.

**Parallel Mechanism.** It involves solving the same problem multiple times in parallel and selecting the best answer from the results. This approach is particularly relevant in generative models, where the process of Bernoulli sampling is inherently stochastic. By running the model multiple times, parallel mechanisms can leverage the randomness to generate a variety of potential solutions and ultimately choose the most optimal one. This method can improve the robustness and reliability of the final solution, although it requires more computational resources.

## 2.3 Post-Processing Techniques

Beyond the aforementioned methods of obtaining solutions, there has been a substantial amount of research that predominantly concentrates on leveraging diverse euristic methods (without offline training) to optimize the initial solutions generated by neural networks. We mainly categorize these methods into two types: *K-Opt* for edge-oriented problems and *Sampling* for node-oriented problems. See Appendix B for visualizations, algorithmic details and parameters of the post-processing means.

**K-Opt Optimization.** The core idea is to reduce the total cost by repeatedly swapping certain edges. Following COExpander [37] and ML4TSPBench [17], we adopt **Two-Opt** for TSP and ATSP, **Monte Carlo Tree Search (MCTS)** for TSP, and **Classic-LS** for CVRP as the post-processing operations.

**Sampling Optimization.** Sampling methods [52, 53] based on energy functions and gradient descent have demonstrate strong performance and generalization in solving COPs with simple-constraint. We adopt **RLSA** [53] as a representative post-processor for the node-oriented problems.

## 3 Related Work and Further Discussion

**Related Works.** RL4CO [41] is an early attempt towards providing a unified implementation and evaluation of CO problems, yet only RL-based models are incorporated without delving into deeper analyses of learning paradigms or proposing new methods. PredictiveCO [54] proposes a modular framework to benchmark predictive CO approaches. Besides, there are also some benchmarking works dedicated to specific problems: 1) [55] proposes to benchmark the maximum independent set problem. 2) the recent effort namely ML4TSPBench [17] endeavors to rethink existing methodologies in ML4TSP while concurrently advancing the development of new techniques, through a highly modularized implementation framework. However, existing benchmarking efforts often focus merely on a single problem type or a specific learning paradigm. The lack of a systematic categorization and transparent evaluation of existing methods impedes the possibility of generating generic insights from a more macroscopic perspective. In this work, we primarily focus on three classic edge-oriented (i.e. routing) problems and four types of node-oriented problems.

**Statement on Topics *not* Discussed.** We primarily focus on the methods which obtain solutions to COPs using purely machine learning approach, along with relevant post-processing optimization techniques. Therefore, the following three will not be discussed: **1) Divide-and-Conquer Solving and Optimization.** GLOP [56] and UDC [57] propose a hierarchical framework that uses a non-autoregressive model to perform coarse-grained decomposition of the problem, and employs corresponding solvers to address the decomposed subproblems. [58] proposes a learning-augmented local search framework for solving large-scale Vehicle Routing Problems (VRPs), which iteratively improves solutions by identifying promising subproblems using a transformer-based subproblem selector and delegating their improvement to a black-box subsolver. **2) Learning to Search.** These methods [59, 60, 61, 62] start from an initial solution and use neural networks to guide how to search for appropriate actions (such as edges swap) to optimize the solution. NeuOpt [63] incorporates the idea of tabu search and proposes Guided Infeasible Region Exploration (GIRE), which significantly enhances the solving effect. **3) Machine Learning-assisted Traditional Solvers**. VSRLKH [64] combines reinforcement learning with the Lin-Kernighan-Helsgaun (LKH) algorithm, which uses Q-learning, Sarsa, and Monte Carlo methods to dynamically select optimal moves. NeuroLKH [65] utilizes a sparse graph neural network to learn edge scores and node penalties to guide LKH. **4)**

**Hypergraphs.** In this paper, the graphs involved do not include hypergraphs [66], that is, an edge connects exactly two nodes. In addition, the weights of nodes and edges are not considered.

# 4 Experiments

In this section, we continue the empirical analysis of existing ML4CO methods along the same three dimensions as specified in Sec. 2. Due to space limitations, the experimental analysis below only presents results from a selection of benchmark datasets. The complete experimental results with standard deviations of the performance drop reported are presented in Appendix D.

## 4.1 Experimental Setup

### 4.1.1 Datasets and Metrics

**Synthetic Datasets.** Following the SOTA COExpander [37] and its protocol, we adopt RB [27], ER [67], and BA [23] graphs for node-oriented problems, and use uniform graphs for the edge-oriented routing problem. Specifically, we set the edge connection probability of ER graph to 0.15, the number of connecting nodes for BA graphs to 4. Besides, for RB and BA graphs, we categorize them into three scales: *SMALL* (200-300), *LARGE* (800-1200), and *GIANT* (2000-3000), based on the number of nodes. For CVRP, the demands of clients are randomly generated from the interval [1, 10], and the vehicle capacity is set to 40, 50, 80, 100 for *CVRP-50*, *CVRP-100*, *CVRP-200* and *CVRP-500*. We also conduct experiments on SATLIB[2] for MIS by transforming SAT into MIS.

**Real-World Datasets.** Following previous work [37, 63], we use TSPLIB[3] for TSP, CVRPLIB[4] for CVRP. Specifically, for TSPLIB, we select 49 2D-Euclidean distances with the number of nodes ranging from 51 to 1002; for CVRPLIB, we adopt 70 instances from the benchmark [68] (corresponding to folder X), with the number of nodes ranging from 101 to 513. As for the node-oriented problem, we adopt two social network datasets: Twitter [69] and COLLAB [70].

**Cross-distribution Generalization Datasets.** In addition to the real-world dataset, we have referred to previous work [55, 33] and reorganized the cross-distribution datasets to conduct further generalization tests. Specifically, for ATSP, we reduce both Hamiltonian Cycle Problem (HCP) and 3-SAT to ATSP and treat them as two distinct distribution classes. For TSP, we adopt Cluster and Gaussian distributions. For the node-oriented tasks, we employ the Holme-Kim (HK) [71] and Watts-Strogatz (WS) [72]. For more details on instance generation, please refer to Appendix C.3.

**Metrics.** Following COExpander [37], we use three metrics to evaluate the performance of the model: **1) Objective.** The average objective of the solutions w.r.t. the corresponding instances. **2) Drop.** The relative performance drop w.r.t. the objective compared to the solutions obtained by the baseline exact solver or heuristic solver. **3) Time.** The average computational time per instance. Unless otherwise specified, all experiments were conducted with a batch size set to 1 or in single-thread mode.

### 4.1.2 Evaluated Methods and Model Settings

**Learning-free Baseline Solvers.** We adopt six baseline traditional solvers without learning: Gurobi-11.0.3 [5] for node-oriented problems; While for edge-oriented routing tasks, we turn to the tailored strong solvers: LKH-3.0.7 [6] for TSP and ATSP; Concorde [73] for TSP; GA-EAX [74] for TSP; HGS [75] for CVRP; KaMIS [76] for MIS.

**Generative Model Settings.** 1) Sampling Number $\mathbf{S}$: the number of heatmaps sampled from $p_\theta(x_0|s)$ with different random seeds. 2) Determination Step $\mathbf{D_s}$: the maximum number of determination in AE paradigm. 3) Inference Step $\mathbf{I_s}$: the number of time-steps in the inference phase.

**Training Settings.** Typically, the learning rate is set to 0.002. Regarding the network backbone and number of layers, we prioritize consistency with the original paper. The detailed experimental environment and training parameters are presented in Appendix C.

---

[2]https://www.cs.ubc.ca/~hoos/SATLIB/Benchmarks/SAT/CBS/descr_CBS.html
[3]http://comopt.ifi.uni-heidelberg.de/software/TSPLIB95/
[4]http://vrp.atd-lab.inf.puc-rio.br/media/com_vrp/instances

Table 2: Comparison of different *paradigms* on performance and efficiency (GP vs LC vs AE).

| METHOD | TYPE | TSP-500 | | | TSP-1K | | | ATSP-500 | | |
|---|---|---|---|---|---|---|---|---|---|---|
| | | OBJ.↓ | DROP↓ | TIME↓ | OBJ.↓ | DROP↓ | TIME↓ | OBJ.↓ | DROP↓ | TIME↓ |
| GP4CO | GP-OS-SL | 18.148 | 9.683% | **0.047s** | 26.124 | 13.001% | **0.195s** | 1.729 | 9.880% | **0.157s** |
| LC4CO | LC-OS-SL | **18.006** | **8.825%** | 3.964s | **25.380** | **9.789%** | 7.995s | **1.655** | **5.201%** | 6.328s |
| GP4CO ($I_s$=5) | GP-Gen-SL | 17.473 | 5.603% | **0.099s** | 25.021 | 8.229% | **0.289s** | 1.641 | 4.294% | **0.558s** |
| AE4CO ($D_s$=3, $I_s$=5) | AE-Gen-SL | **17.119** | **3.466%** | 0.231s | **24.698** | **6.834%** | 0.691s | **1.623** | **3.128%** | 1.669s |

| METHOD | TYPE | MIS-RB-LARGE | | | MCL-RB-LARGE | | | MVC-RB-LARGE | | |
|---|---|---|---|---|---|---|---|---|---|---|
| | | OBJ.↑ | DROP↓ | TIME↓ | OBJ.↑ | DROP↓ | TIME↓ | OBJ.↓ | DROP↓ | TIME↓ |
| GP4CO | GP-OS-SL | 36.060 | 16.036% | **0.048s** | **38.622** | **4.331%** | **0.042s** | 976.112 | 0.817% | **0.068s** |
| LC4CO | LC-OS-SL | **37.596** | **12.476%** | 3.962s | 29.382 | 26.156% | 3.148s | **972.532** | **0.447%** | 3.950s |
| GP4CO ($I_s$=1) | GP-Gen-SL | 36.394 | 15.219% | **0.044s** | 36.402 | 9.809% | **0.042s** | 974.950 | 0.696% | **0.044s** |
| AE4CO ($D_s$=20, $I_s$=1) | AE-Gen-SL | **41.234** | **4.060%** | 0.624s | **36.752** | **8.817%** | 0.150s | **969.922** | **0.176%** | 0.522s |

Table 3: Comparison between one-shot (OS) and generative (Gen) *models*.

| METHOD | TYPE | TSP-100 | | | ATSP-100 | | | ATSP-200 | | |
|---|---|---|---|---|---|---|---|---|---|---|
| | | OBJ.↓ | DROP↓ | TIME↓ | OBJ.↓ | DROP↓ | TIME↓ | OBJ.↓ | DROP↓ | TIME↓ |
| GP4CO | GP-OS-SL | 7.937 | 2.334% | 0.007s | 1.666 | 6.408% | 0.008s | 1.671 | 6.777% | 0.053s |
| GP4CO ($I_s$=1) | GP-Gen-SL | 7.941 | 2.381% | 0.009s | 1.665 | 6.329% | 0.008s | 1.678 | 7.280% | 0.037s |
| GP4CO ($I_s$=5) | GP-Gen-SL | **7.774** | **0.230%** | 0.031s | **1.628** | **3.935%** | 0.033s | **1.638** | **4.677%** | 0.162s |

| METHOD | TYPE | MIS-RB-LARGE | | | MIS-ER-700-800 | | | MIS-SATLIB | | |
|---|---|---|---|---|---|---|---|---|---|---|
| | | OBJ.↑ | DROP↓ | TIME↓ | OBJ.↑ | DROP↓ | TIME↓ | OBJ.↑ | DROP↓ | TIME↓ |
| GP4CO | GP-OS-SL | 36.060 | 16.036% | 0.048s | 35.852 | 20.267% | 0.039s | 421.056 | 1.154% | 0.019s |
| GP4CO ($I_s$=1) | GP-Gen-SL | 36.394 | 15.219% | 0.044s | 35.711 | 20.580% | 0.039s | 421.578 | 1.031% | 0.016s |
| GP4CO ($I_s$=20) | GP-Gen-SL | **38.936** | **9.388%** | 0.714s | **40.195** | **10.618%** | 0.641s | **424.556** | **0.329%** | 0.190s |

| METHOD | TYPE | MCL-RB-LARGE | | | MVC-RB-LARGE | | | MCUT-BA-LARGE | | |
|---|---|---|---|---|---|---|---|---|---|---|
| | | OBJ.↑ | DROP↓ | TIME↓ | OBJ.↓ | DROP↓ | TIME↓ | OBJ.↑ | DROP↓ | TIME↓ |
| GP4CO | GP-OS-SL | **38.622** | **4.331%** | 0.042s | 976.112 | 0.817% | 0.068s | 2827.480 | 3.744% | 0.016s |
| GP4CO ($I_s$=1) | GP-Gen-SL | 36.402 | 9.809% | 0.042s | 974.950 | 0.696% | 0.044s | 2783.834 | 5.231% | 0.016s |
| GP4CO ($I_s$=20) | GP-Gen-SL | 35.384 | 12.672% | 0.708s | **972.608** | **0.453%** | 0.718s | **2948.112** | **-0.369%** | 0.194s |

## 4.2 Experimental Results and Analysis

### 4.2.1 Analysis of Different Algorithm Designs

To avoid the influence of decoding strategies and post-processing methods on the results, we adopt *greedy decoding*, the most fundamental strategy, to analyze the differences in performance and efficiency resulting from various algorithm designs. Corresponding to Sec. 2, we conduct analyses from the three dimensions of algorithm design. For **paradigm**, we use *GP-OS-SL* vs *LC-OS-SL* and *GP-Gen-SL* vs *AE-Gen-SL* to perform comparative experiments. For **model**, we conduct a comparison between *GP-OS-SL* vs *GP-Gen-SL*. As for **learning**, we use *GP-OS-UL* vs *GP-OS-MAML* vs *GP-OS-SL*; *GP-Gen-UL* vs *GP-Gen-SL*; as well as *LC-OS-RL* vs *LC-Gen-SL*. We have conducted these comparative experiments across various COPs and have drawn the following analytical conclusions.

- **Paradigm Efficiency Analysis.** Due to the different numbers of iterations $k$ required by the three paradigms to construct a complete solution, there is an inherent order in their solving speeds under the same settings, with GP > AE > LC, as proven in Table 2.

- **Paradigm Performance.** The performance is mainly influenced by the model and the training method. However, as shown in the data from Table 2, the performance of LC and AE is slightly better than that of GP when other settings are the same, which may be related to the *prediction conflicts* [37] existing in the GP paradigm.

- **Model Efficiency.** The single-step inference time of generative models and one-shot models is comparable. Therefore, when $I_s$ is set to 1, the two models are consistent in terms of efficiency - see Table 3. Besides, as $I_s$ increases, the solution time of the generative model increases accordingly.

- **Model Performance.** When $I_s = 1$, the performance of the two models is similar. As $I_s$ increases, the more fine-grained noise addition and reduction process typically allows the generative model to demonstrate superior performance. Nevertheless, it can be observed that the result of MCl in Table 3 does not follow this pattern.

- **Learning Analysis for Edge Problems.** For the global prediction paradigm, increasing the amount of training data can enhance the solution performance, i.e., SL > MAML > UL. This is because the constraints of the edge-oriented routing problem are relatively complex, making it difficult to

Table 4: Comparison of different *training* methods for edge problems (UL vs SL vs RL vs MAML).

| METHOD | TYPE | TSP-50 | | | TSP-100 | | | TSP-500 | | |
|---|---|---|---|---|---|---|---|---|---|---|
| | | OBJ.↓ | DROP↓ | TIME↓ | OBJ.↓ | DROP↓ | TIME↓ | OBJ.↓ | DROP↓ | TIME↓ |
| GP4CO | GP-OS-UL | 7.097 | 24.808% | 0.007s | 9.377 | 20.909% | 0.009s | 19.761 | 19.435 | 0.059s |
| GP4CO | GP-OS-MAML | 6.325 | 11.233% | 0.003s | 8.757 | 12.918% | 0.005s | 18.888 | 14.155% | 0.273s |
| GP4CO | GP-OS-SL | **5.718** | **0.520%** | 0.006s | **7.937** | **2.334%** | 0.007s | **18.148** | **9.683%** | 0.047s |
| LC4CO | LC-OS-RL | **5.702** | **0.258%** | 0.050s | **7.872** | **1.491%** | 0.767s | 21.105 | 27.561% | 0.460s |
| LC4CO | LC-OS-SL | 5.742 | 0.965% | 0.393s | 7.905 | 1.920% | 0.762s | **18.006** | **8.825%** | 3.964s |

| METHOD | TYPE | ATSP-50 | | | ATSP-100 | | | CVRP-50 | | |
|---|---|---|---|---|---|---|---|---|---|---|
| | | OBJ.↓ | DROP↓ | TIME↓ | OBJ.↓ | DROP↓ | TIME↓ | OBJ.↓ | DROP↓ | TIME↓ |
| LC4CO | LC-OS-RL | **1.576** | **1.373%** | 0.032s | **1.620** | **3.456%** | 0.053s | **10.769** | **3.891%** | 0.087s |
| LC4CO | LC-OS-SL | 1.595 | 2.567% | 0.395s | 1.644 | 5.010% | 0.723s | 10.906 | 5.193% | 0.504s |

| METHOD | TYPE | CVRP-100 | | | CVRP-200 | | | CVRP-500 | | |
|---|---|---|---|---|---|---|---|---|---|---|
| | | OBJ.↓ | DROP↓ | TIME↓ | OBJ.↓ | DROP↓ | TIME↓ | OBJ.↓ | DROP↓ | TIME↓ |
| LC4CO | LC-OS-RL | **16.220** | **4.241%** | 0.166s | **20.662** | **5.274%** | 0.320s | 40.382 | 8.723% | 0.769s |
| LC4CO | LC-OS-SL | 16.342 | 5.005% | 0.962s | 21.439 | 9.263% | 1.978s | **38.846** | **4.587%** | 7.095s |

Table 5: Comparison of different *training* methods for node problems (UL vs SL vs MAML).

| METHOD | TYPE | MIS-RB-SMALL | | | MIS-RB-LARGE | | | MIS-RB-GIANT | | |
|---|---|---|---|---|---|---|---|---|---|---|
| | | OBJ.↑ | DROP↓ | TIME↓ | OBJ.↑ | DROP↓ | TIME↓ | OBJ.↑ | DROP↓ | TIME↓ |
| GP4CO | GP-OS-UL | 17.308 | 13.709% | 0.016s | 34.002 | 20.779% | 0.029s | 37.260 | 24.195% | 0.151s |
| GP4CO | GP-OS-MAML | 17.692 | 11.778% | 0.084s | 34.670 | 19.259% | 0.127s | **38.840** | **21.039%** | 0.739s |
| GP4CO | GP-OS-SL | **18.170** | **9.413%** | 0.014s | **36.060** | **16.036%** | 0.048s | 38.760 | 21.224% | 0.240s |
| GP4CO | GP-Gen-UL | 19.200 | 4.369% | 0.470s | 38.490 | 10.428% | 4.712s | **40.480** | **17.722%** | 45.680s |
| GP4CO | GP-Gen-SL | **19.330** | **3.710%** | 0.192s | **38.936** | **9.388%** | 0.714s | 38.380 | 21.995% | 8.980s |

| METHOD | TYPE | MCL-RB-SMALL | | | MCL-RB-LARGE | | | MCL-RB-GIANT | | |
|---|---|---|---|---|---|---|---|---|---|---|
| | | OBJ.↑ | DROP↓ | TIME↓ | OBJ.↑ | DROP↓ | TIME↓ | OBJ.↑ | DROP↓ | TIME↓ |
| GP4CO | GP-OS-UL | 15.084 | 20.720% | 0.016s | 29.052 | 26.604% | 0.029s | 55.580 | 29.171% | 0.148s |
| GP4CO | GP-OS-MAML | 17.856 | 6.400% | 0.065s | 34.802 | 13.180% | 0.103s | **77.160** | **1.613%** | 0.683s |
| GP4CO | GP-OS-SL | **18.084** | **5.782%** | 0.014s | **38.622** | **4.331%** | 0.042s | 58.440 | 26.568% | 0.220s |

| METHOD | TYPE | MVC-RB-SMALL | | | MVC-RB-LARGE | | | MVC-RB-GIANT | | |
|---|---|---|---|---|---|---|---|---|---|---|
| | | OBJ.↓ | DROP↓ | TIME↓ | OBJ.↓ | DROP↓ | TIME↓ | OBJ.↓ | DROP↓ | TIME↓ |
| GP4CO | GP-OS-SL | **207.554** | **0.871%** | 0.014s | **976.112** | **0.817%** | 0.068s | 2407.940 | 0.465% | 0.365s |
| GP4CO | GP-OS-UL | 208.982 | 1.569% | 0.013s | 977.186 | 0.928% | 0.037s | 2408.320 | 0.482% | 0.221s |
| GP4CO | GP-OS-MAML | 208.814 | 1.486% | 0.065s | 976.672 | 0.876% | 0.097s | **2406.360** | **0.402%** | 0.555s |

| METHOD | TYPE | MCUT-BA-SMALL | | | MCUT-BA-LARGE | | | MCUT-BA-GIANT | | |
|---|---|---|---|---|---|---|---|---|---|---|
| | | OBJ.↑ | DROP↓ | TIME↓ | OBJ.↑ | DROP↓ | TIME↓ | OBJ.↑ | DROP↓ | TIME↓ |
| GP4CO | GP-OS-MAML | 672.748 | 7.563% | 0.063s | 2748.310 | 6.425% | 0.090s | 6712.120 | 6.994% | 0.535s |
| GP4CO | GP-OS-SL | **705.330** | **3.102%** | 0.014s | 2827.480 | 3.744% | 0.016s | 6979.260 | 3.291% | 0.060s |
| GP4CO | GP-OS-UL | 700.972 | 3.706% | 0.017s | **2884.086** | **1.815%** | 0.019s | **7124.880** | **1.287%** | 0.059s |
| GP4CO | GP-Gen-SL | 725.624 | 0.319% | 0.172s | 2948.112 | -0.369% | 0.194s | 7308.260 | -1.258% | 0.700s |
| GP4CO-FT | GP-Gen-SL | 726.538 | 0.195% | 0.172s | 2980.508 | -1.467% | 0.196s | 7372.100 | -2.134% | 0.720s |
| GP4CO | GP-Gen-UL | **726.900** | **0.146%** | 0.197s | **2986.932** | **-1.688%** | 0.654s | **7384.020** | **-2.306%** | 2.480s |

obtain high-quality global heatmaps through unsupervised methods. For the local construction paradigm, when the scale is small, RL outperforms SL, as shown in Table 4. However, as the scale increases, the performance of RL methods falls behind that of SL methods due to the rapid expansion of the state space and the sparsity of rewards.

- *Learning Analysis for Node Problems.* Compared with the edge-oriented problems, UL performs relatively better on the node-oriented problems, and even outperforms SL on MCut, as shown in Table 5. These may because: **1)** *the quality of supervised data.* When obtaining training labels, the solvers (Gurobi, KaMIS) used for the node-oriented problems has relatively weaker performance than the solvers (Concorde, LKH, HGS) used for the edge-oriented problems. **2)** *the uniqueness of the solution.* The solutions to edge-oriented problems are often unique, whereas solutions to node-oriented problems typically have many possibilities. **3)** *the design of the loss function.* The node-oriented problems can be readily transformed into their corresponding energy equations [19], as detailed in Appendix A. This transformation makes it natural to leverage these energy equations as the loss functions for unsupervised training. Particularly in the case of MCut, there are even no constraint terms, allowing the loss function to consistently decrease in a single direction. However, in the edge-oriented problems, the number of decision variables is quadratically related to the problem size, which significantly increases the complexity. Moreover, due to more intricate constraint relationships, designing the loss function for edge-oriented problems often requires optimizing multiple objectives, which may also lead to poor training outcomes. Additionally, our experiments also validate the generalization ability of MAML as proposed in [18, 20].

Table 6: Research on the performance improvement of beam search.

| METHOD | TYPE | BEAM-16 | TSP-500 | | | ATSP-500 | | | CVRP-500 | | |
|---|---|---|---|---|---|---|---|---|---|---|---|
| | | | OBJ.↓ | DROP↓ | TIME↓ | OBJ.↓ | DROP↓ | TIME↓ | OBJ.↓ | DROP↓ | TIME↓ |
| LC4CO | LC-OS-SL | ✗ | 18.006 | 8.825% | 3.964s | 1.655 | 5.201% | 6.328s | 38.846 | 4.587% | 7.095s |
| LC4CO | LC-OS-SL | ✔ | **17.279** | **4.433%** | 4.940s | **1.639** | **4.190%** | 63.388s | **38.279** | **3.051%** | 64.431s |

| METHOD | TYPE | BEAM-16 | MCL-RB-SMALL | | | MCL-RB-LARGE | | | MCL-RB-GIANT | | |
|---|---|---|---|---|---|---|---|---|---|---|---|
| | | | OBJ.↑ | DROP↓ | TIME↓ | OBJ.↑ | DROP↓ | TIME↓ | OBJ.↑ | DROP↓ | TIME↓ |
| GP4CO | GP-OS-UL | ✗ | 15.084 | 20.720% | 0.016s | 29.052 | 26.604% | 0.029s | 55.580 | 29.171% | 0.148s |
| GP4CO | GP-OS-UL | ✔ | **17.086** | **10.572%** | 0.017s | **35.098** | **12.318%** | 0.091s | **76.360** | **4.433%** | 0.333s |
| GP4CO | GP-OS-SL | ✗ | 18.084 | 5.782% | 0.014s | 38.622 | 4.331% | 0.042s | 58.440 | 26.568% | 0.220s |
| GP4CO | GP-OS-SL | ✔ | **18.592** | **2.950%** | 0.023s | **39.540** | **1.857%** | 0.104s | **77.460** | **3.119%** | 0.406s |
| GP4CO (I$_s$=1) | GP-Gen-SL | ✗ | 18.258 | 4.870% | 0.016s | 36.402 | 9.809% | 0.042s | 60.860 | 22.902% | 0.240s |
| GP4CO (I$_s$=1) | GP-Gen-SL | ✔ | **18.664** | **2.510%** | 0.022s | **38.754** | **3.798%** | 0.104s | **77.340** | **2.732%** | 0.420s |

Table 7: Research on the performance improvement of parallel mechanism.

| METHOD | TYPE | PARALLEL | TSP-500 | | | TSP-1K | | | ATSP-500 | | |
|---|---|---|---|---|---|---|---|---|---|---|---|
| | | | OBJ.↓ | DROP↓ | TIME↓ | OBJ.↓ | DROP↓ | TIME↓ | OBJ.↓ | DROP↓ | TIME↓ |
| GP4CO (I$_s$=5) | GP-Gen-SL | ✗ | 17.473 | 5.603% | 0.099s | 25.021 | 8.229% | 0.289s | 1.641 | 4.294% | 0.558s |
| GP4CO (I$_s$=5) | GP-Gen-SL | ✔ | **17.012** | **2.817%** | 0.289s | **24.371** | **5.419%** | 1.188s | **1.614** | **2.543%** | 2.144s |
| AE4CO (D$_s$=3, I$_s$=5) | AE-Gen-SL | ✗ | 17.119 | 3.466% | 0.231s | 24.698 | 6.834% | 0.691s | 1.623 | 3.128% | 1.669s |
| AE4CO (D$_s$=3, I$_s$=5) | AE-Gen-SL | ✔ | **16.776** | **1.395%** | 0.289s | **24.081** | **4.165%** | 2.412s | **1.604** | **1.965%** | 6.267s |

| METHOD | TYPE | PARALLEL | MIS-RB-SMALL | | | MIS-RB-LARGE | | | MIS-ER-700-800 | | |
|---|---|---|---|---|---|---|---|---|---|---|---|
| | | | OBJ.↑ | DROP↓ | TIME↓ | OBJ.↑ | DROP↓ | TIME↓ | OBJ.↑ | DROP↓ | TIME↓ |
| GP4CO (I$_s$=20) | GP-Gen-SL | ✗ | 19.330 | 3.710% | 0.192s | 38.936 | 9.388% | 0.714s | 40.195 | 10.618% | 0.641s |
| GP4CO (I$_s$=20) | GP-Gen-SL | ✔ | **19.742** | **1.690%** | 0.332s | **40.066** | **6.742%** | 2.398s | **41.430** | **7.867%** | 2.094s |
| AE4CO (D$_s$=20, I$_s$=1) | AE-Gen-SL | ✗ | 19.662 | 2.088% | 0.112s | 41.234 | 4.060% | 0.624s | 42.383 | 5.746% | 0.469s |
| AE4CO (D$_s$=20, I$_s$=1) | AE-Gen-SL | ✔ | **19.706** | **1.880%** | 0.120s | **41.438** | **3.582%** | 1.298s | **42.563** | **5.343%** | 0.703s |

| METHOD | TYPE | PARALLEL | MCL-RB-SMALL | | | MCL-RB-LARGE | | | MVC-RB-LARGE | | |
|---|---|---|---|---|---|---|---|---|---|---|---|
| | | | OBJ.↑ | DROP↓ | TIME↓ | OBJ.↑ | DROP↓ | TIME↓ | OBJ.↓ | DROP↓ | TIME↓ |
| GP4CO (I$_s$=20) | GP-Gen-SL | ✗ | 18.608 | 2.850% | 0.200s | 35.384 | 12.672% | 0.708s | 972.608 | 0.453% | 0.718s |
| GP4CO (I$_s$=20) | GP-Gen-SL | ✔ | **18.982** | **0.607%** | 0.328s | **38.706** | **4.197%** | 2.386s | **971.420** | **0.330%** | 2.412s |
| AE4CO (D$_s$=20, I$_s$=1) | AE-Gen-SL | ✗ | 18.766 | 1.892% | 0.046s | 36.752 | 8.817% | 0.150s | 969.922 | 0.176% | 0.522s |
| AE4CO (D$_s$=20, I$_s$=1) | AE-Gen-SL | ✔ | **18.922** | **1.018%** | 0.056s | **39.170** | **2.722%** | 0.414s | **969.806** | **0.163%** | 0.626s |

Table 8: ML4CO-101 vs baseline solvers: a summarized comparaison. Rd denotes random.

| PROBLEM | DATASET | LEARNING-FREE BASELINE | | | LEARNING-BASED METHODS | | | | |
|---|---|---|---|---|---|---|---|---|---|
| | | METHOD | OBJ. | TIME↓ | METHOD TYPE | SETTINGS | OBJ. | DROP↓ | TIME↓ |
| TSP | TSP-50 | Concorde [73] | 5.688 | 0.059s | GP-OS-SL | Rd-16 + MCTS (0.05s) | 5.688 | 0.001% | 0.031s |
| TSP | TSP-100 | Concorde [73] | 7.756 | 0.238s | GP-OS-SL | Rd-16 + MCTS (0.05s) | 7.756 | 0.005% | 0.119s |
| TSP | TSP-500 | Concorde [73] | 16.546 | 18.672s | AE-Gen-SL | 4×Greedy + Two-Opt | 16.588 | 0.257% | 0.695s |
| TSP | TSP-1K | Concorde [73] | 23.118 | 84.413s | AE-Gen-SL | 4×Greedy + Two-Opt | 23.271 | 0.662% | 2.609s |
| TSP | TSP-10K | LKH(500) [6] | 71.755 | 332.758s | AE-Gen-SL | Greedy + Two-Opt | 72.832 | 1.450% | 33.485s |
| TSP | TSPLIB | Concorde [73] | 8.062 | – | – | – | 8.095 | 0.356% | – |
| ATSP | ATSP-50 | LKH (1K) [6] | 1.554 | 0.097s | AE-Gen-SL | 4×Greedy + Two-Opt | 1.557 | 0.171% | 0.103s |
| ATSP | ATSP-100 | LKH (1K) [6] | 1.566 | 0.238s | AE-Gen-SL | 4×Greedy + Two-Opt | 1.581 | 0.946% | 0.516s |
| ATSP | ATSP-200 | LKH (1K) [6] | 1.565 | 0.724s | AE-Gen-SL | 4×Greedy + Two-Opt | 1.588 | 1.501% | 1.097s |
| ATSP | ATSP-500 | LKH (1K) [6] | 1.573 | 4.376s | AE-Gen-SL | 4×Greedy + Two-Opt | 1.598 | 1.568% | 6.597s |
| CVRP | CVRP-50 | HGS [75] | 10.366 | 1.005s | LC-OS-RL | N×Sampling + Classic-LS | 10.489 | 1.179% | 0.154s |
| CVRP | CVRP-100 | HGS [75] | 15.563 | 20.027s | LC-OS-RL | N×Sampling + Classic-LS | 15.822 | 1.663% | 0.202s |
| CVRP | CVRP-200 | HGS [75] | 19.630 | 60.024s | LC-OS-RL | N×Sampling + Classic-LS | 20.091 | 2.359% | 0.439s |
| CVRP | CVRP-500 | HGS [75] | 37.154 | 360.376s | LC-OS-RL | Beam-16 + Classic-LS | 37.901 | 2.031% | 64.589s |
| CVRP | CVRPLIB | HGS [75] | 45.183 | – | LC-OS-RL | N×Sampling + Classic-LS | 48.263 | 5.469% | – |
| MIS | RB-SMALL | Gurobi [5] | 20.090 | 0.538s | AE-Gen-SL | Greedy + RLSA | 20.070 | 0.093% | 0.471s |
| MIS | RB-LARGE | KaMIS [76] | 43.004 | 56.974s | AE-Gen-SL | Greedy + RLSA | 42.400 | 1.366% | 1.816s |
| MIS | ER-700-800 | KaMIS [76] | 44.969 | 60.753s | AE-Gen-SL | Greedy + RLSA | 44.984 | -0.041% | 1.390s |
| MIS | SATLIB | KaMIS [76] | 425.954 | 24.368s | AE-Gen-SL | Greedy + RLSA | 425.316 | 0.151% | 1.775s |
| MIS | ER-1400-1600 | KaMIS [76] | 50.938 | 60.824s | AE-Gen-SL | Greedy + RLSA | 50.719 | 0.418% | 4.102s |
| MIS | RB-GIANT | KaMIS [76] | 49.260 | 60.960s | AE-Gen-SL | Greedy + RLSA | 47.880 | 2.741% | 9.490s |
| MCl | RB-SMALL | Gurobi [5] | 19.082 | 0.900s | GP-OS-SL | Beam-16 + RLSA | **19.082** | **0.000%** | **0.041s** |
| MCl | RB-LARGE | Gurobi [5] | 40.182 | 276.657s | GP-OS-SL | Beam-16 + RLSA | **40.256** | **-0.275%** | **0.171s** |
| MCl | TWITTER | Gurobi [5] | 14.210 | 0.276s | GP-OS-SL | Beam-16 + RLSA | **14.210** | **0.000%** | **0.044s** |
| MCl | COLLAB | Gurobi [5] | 42.113 | 0.063s | GP-OS-SL | Beam-16 + RLSA | **42.113** | **0.000%** | **0.024s** |
| MCl | RB-GIANT | Gurobi [5] | 81.520 | 3606.201s | GP-OS-SL | Beam-16 + RLSA | **85.380** | **-7.912%** | **4.342s** |
| MVC | RB-SMALL | Gurobi [5] | 205.764 | 3.340s | AE-Gen-SL | Greedy + RLSA | 205.772 | 0.004% | 0.612s |
| MVC | RB-LARGE | Gurobi [5] | 968.228 | 290.227s | AE-Gen-SL | Greedy + RLSA | 968.398 | 0.018% | 1.592s |
| MVC | TWITTER | Gurobi [5] | 85.251 | 0.133s | AE-Gen-SL | Greedy + RLSA | **85.251** | **0.000%** | **0.115s** |
| MVC | COLLAB | Gurobi [5] | 65.086 | 0.058s | AE-Gen-SL | Greedy + RLSA | 65.086 | 0.000% | 0.158s |
| MVC | RB-GIANT | Gurobi [5] | 2396.780 | 60.612s | AE-Gen-SL | Greedy + RLSA | 2397.360 | 0.026% | 8.590s |
| MCut | BA-SMALL | Gurobi [5] | 727.844 | 60.612s | AE-Gen-SL | Greedy + RLSA | **729.706** | **-0.240%** | **0.727s** |
| MCut | BA-LARGE | Gurobi [5] | 2936.886 | 300.214s | GP-OS-SL | Greedy + RLSA | **2994.118** | **-1.932%** | **0.999s** |
| MCut | BA-GIANT | Gurobi [5] | 7217.900 | 3601.342s | GP-Gen-SL | Greedy + RLSA | **7389.300** | **-2.383%** | **2.228s** |

Table 9: Generalization Study on Cross-distribution Datasets.

| PROBLEM | DATASET | LEARNING-FREE BASELINE | | | LEARNING-BASED METHODS | | | | |
|---|---|---|---|---|---|---|---|---|---|
| | | METHOD | OBJ. | TIME↓ | METHOD TYPE | SETTINGS | OBJ. | DROP↓ | TIME↓ |
| TSP | Cluster-50 | Concorde [73] | 3.730 | 0.140s | GP-OS-SL | Rd-16 + MCTS (0.05s) | 3.730 | 0.001% | 0.098s |
| TSP | Cluster-100 | Concorde [73] | 5.526 | 0.290s | GP-OS-SL | Rd-16 + MCTS (0.05s) | 5.527 | 0.011% | 0.604s |
| TSP | Cluster-500 | Concorde [73] | 10.723 | 5.073s | AE-Gen-SL | 4×Greedy + Two-Opt | 10.937 | 1.998% | 1.250s |
| TSP | Gaussian-50 | Concorde [73] | 23.840 | 0.166s | GP-OS-SL | Rd-16 + MCTS (0.05s) | 23.841 | 0.005% | 0.256s |
| TSP | Gaussian-100 | Concorde [73] | 34.031 | 0.438s | GP-OS-SL | Rd-16 + MCTS (0.05s) | 34.044 | 0.037% | 0.604s |
| TSP | Gaussian-500 | Concorde [73] | 77.521 | 19.952s | AE-Gen-SL | 4×Greedy + Two-Opt | 78.122 | 0.774% | 1.169s |
| ATSP | HCP-50 | LKH (1K) [6] | 0.000 | 0.107s | AE-Gen-SL | Greedy + Two-Opt | 1.468 | – | 0.075s |
| ATSP | HCP-100 | LKH (1K) [6] | 0.000 | 0.211s | AE-Gen-SL | Greedy + Two-Opt | 1.380 | – | 0.109s |
| ATSP | HCP-200 | LKH (1K) [6] | 0.000 | 0.355s | AE-Gen-SL | Greedy + Two-Opt | 1.050 | – | 2.508s |
| ATSP | HCP-500 | LKH (1K) [6] | 0.000 | 1.410s | AE-Gen-SL | Greedy + Two-Opt | 0.880 | – | 3.219s |
| ATSP | SAT-50 | LKH (1K) [6] | 0.151 | 0.079s | AE-Gen-SL | Greedy + Two-Opt | 5.018 | – | 0.076s |
| ATSP | SAT-100 | LKH (1K) [6] | 0.079 | 0.128s | AE-Gen-SL | Greedy + Two-Opt | 12.717 | – | 0.107s |
| ATSP | SAT-200 | LKH (1K) [6] | 0.130 | 0.192s | AE-Gen-SL | Greedy + Two-Opt | 21.470 | – | 1.768s |
| ATSP | SAT-500 | LKH (1K) [6] | 0.430 | 0.781s | AE-Gen-SL | Greedy + Two-Opt | 2.340 | – | 3.222s |
| MIS | HK-SMALL | KaMIS [76] | 79.372 | 54.174s | AE-Gen-SL | Greedy + RLSA | 79.358 | 0.017% | 0.353s |
| MIS | HK-LARGE | KaMIS [76] | 330.946 | 67.272s | AE-Gen-SL | Greedy + RLSA | 330.422 | 0.154% | 1.600s |
| MIS | WS-SMALL | KaMIS [76] | 76.904 | 51.490s | AE-Gen-SL | Greedy + RLSA | 76.894 | 0.013% | 0.357s |
| MIS | WS-LARGE | KaMIS [76] | 262.570 | 37.792s | AE-Gen-SL | Greedy + RLSA | 260.114 | 0.926% | 1.502s |
| MCl | HK-SMALL | Gurobi [5] | 6.792 | 1.838s | GP-OS-SL | Beam-16 + RLSA | **6.792** | **0.000%** | **0.039s** |
| MCl | HK-LARGE | Gurobi [5] | 6.774 | 46.502s | GP-OS-SL | Beam-16 + RLSA | **6.774** | **0.000%** | **0.169s** |
| MCl | WS-SMALL | Gurobi [5] | 7.164 | 1.589s | GP-OS-SL | Beam-16 + RLSA | **7.164** | **0.000%** | **0.040s** |
| MCl | WS-LARGE | Gurobi [5] | 5.978 | 27.051s | GP-OS-SL | Beam-16 + RLSA | 5.976 | 0.033% | 0.172s |
| MVC | HK-SMALL | Gurobi [5] | 142.506 | 0.382s | AE-Gen-SL | Greedy + RLSA | **142.506** | **0.000%** | **0.353s** |
| MVC | WS-SMALL | Gurobi [5] | 154.618 | 1.596s | AE-Gen-SL | Greedy + RLSA | **154.610** | **-0.005%** | **0.317s** |
| MCut | HK-SMALL | Gurobi [5] | 1540.608 | 60.089s | AE-Gen-SL | Greedy + RLSA | **1542.810** | **-0.141%** | **0.382s** |
| MCut | HK-LARGE | Gurobi [5] | 6401.320 | 300.357s | GP-Gen-SL | Greedy + RLSA | **6454.100** | **-0.766%** | **0.991s** |
| MCut | WS-SMALL | Gurobi [5] | 872.116 | 60.357s | AE-Gen-SL | Greedy + RLSA | **874.372** | **-0.250%** | **0.338s** |
| MCut | WS-LARGE | Gurobi [5] | 3454.176 | 300.247s | GP-Gen-SL | Greedy + RLSA | **3489.304** | **-1.007%** | **0.668s** |

#### 4.2.2 Enhancement from Decoding Strategies and Post-Processing Methods

**Decoding Strategy.** As shown in Table 6 and Table 7, compared with greedy decoding, beam search and parallel mechanisms (×4) bring about performance improvements of approximately 67% and 42%, respectively. A larger number of candidates $k$ in beam search and a bigger number of parallelism in parallel mechanisms can lead to better results. However, this comes at the cost of increased time consumption. A trade-off between performance and time needs to be made in practice.

**Post-Processing Methods.** Intuitively, we find the incorporation of post-processing greatly enhances the quality of the solutions. For a clearer principal line, we defer the complete results of various integration of post-processing methods with the neural counterparts to Appendix D, while leaving the best method composition for each benchmark (both time and performance considered) in Table 8.

#### 4.2.3 Generalization Study on Cross-distribution Datasets

We further conducted a generalization study on cross-distribution datasets, with the results presented in Table 9. The model selection and solving parameters remain consistent with those used in Table 8. For the ATSP, several clarifications are necessary: 1) Owing to the inherent properties of the HCP and SAT problems, the derived ATSP instances have an optimal solution of zero by construction, making it infeasible to compute performance drop values. 2) Since the solution is determined during instance generation, the result from the LKH(1K) solver merely serves as a benchmark to demonstrate the performance of a traditional solver, rather than a reference solution. 3) To accelerate the solving process, we employ a *Greedy + Two-Opt* approach for the ATSP, instead of the 4×*Greedy + Two-Opt* used elsewhere. As shown by the generalization results, the proposed model achieves strong generalization on most problem types, with the exception of the SAT distribution in ATSP.

## 5 Conclusion and Outlook

We have developed a benchmark for 7 classic CO problems on graphs. We categorize existing works under a *paradigm-model-learning* criteria with three major schemes: *GP4CO*, *LC4CO*, and *AE4CO*. We also provide 34 benchmark datasets across the COPs with extensive experiments on representative methods. In the future, we will extend our benchmark to encompass more real-world CO problems, such as TSPTW, CVRPTW, and JSSP. We hope our work can ultimately advance the research on more practical CO problems, in analogy to the role of ImageNet [77] for vision, i.e., from classic object detection in static images to more open-environment applications like autonomous driving.

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

# Appendix

## A Formal Problem Definitions

**Traveling Salesman Problem (TSP).** Given $G$ and a cost matrix $\mathbf{C} \in \mathbb{R}^{N \times N}$, where $\mathbf{C}_{ij}$ denotes the cost of edge $(i, j)$, the objective is to find a tour $\tau = (i_1, \cdots, i_N)$ that minimizes the total cost: $\sum_{k=1}^{N-1} \mathbf{C}_{i_k i_{k+1}} + \mathbf{C}_{i_N i_1}$.

**Asymmetric Traveling Salesman Problem (ATSP).** ATSP is a variant of TSP where the cost matrix $\mathbf{C}$ is not necessarily symmetric, i.e., $\mathbf{C}_{ij} = \mathbf{C}_{ji}$ does not always hold for all $i, j \in \mathcal{V}$. In this paper, we follow [32, 31, 33, 37] to study the metric ATSP, where the triangle inequality holds: $\mathbf{C}_{ij} + \mathbf{C}_{jk} \geq \mathbf{C}_{ik}$ for any distinct nodes $i$, $j$, and $k$.

**Capacitated Vehicle Routing Problem (CVRP).** Given $G$, a depot node $v_0 \in \mathcal{V}$, a cost matrix $\mathbf{C} \in \mathbb{R}^{N \times N}$, a demand vector $\mathbf{d} \in \mathbb{R}_+^N$, and a vehicle capacity $Q > 0$, the goal is to plan a set of routes $\mathcal{R}$, each route $r \in \mathcal{R}$ starting and ending at the depot $v_0$, such that each customer node is visited exactly once and the total demand on each route does not exceed $Q$, i.e., $\sum_{i \in r} \mathbf{d}_i \leq Q$. The objective is to minimize the total cost of all routes: $\min_{\mathcal{R}} \sum_{r \in \mathcal{R}} \sum_{(i,j) \in r} \mathbf{C}_{ij}$.

**Maximum Independent Set (MIS).** Given $G$, an *independent set* $S \subseteq \mathcal{V}$ is a subset of nodes such that no two nodes in $S$ are adjacent. The goal is to maximize $|S|$ s.t. $\forall i, j \in S, (i, j) \notin \mathcal{E}$.

**Maximum Clique (MCl).** Given $G$, a *clique* $K \subseteq \mathcal{V}$ is a subset of nodes in which every pair is connected by an edge. It aims to maximize $|K|$ s.t. $\forall i, j \in K, (i, j) \in \mathcal{E}$.

**Minimum Vertex Cover (MVC).** Given $G$, a *vertex cover* $C \subseteq \mathcal{V}$ is a subset of nodes such that every edge $(i, j) \in \mathcal{E}$, at least one of $i$ or $j$ is in $C$. Mathematically, the goal is to minimize $|C|$ s.t. $\forall (i, j) \in \mathcal{E}, i \in C$ or $j \in C$.

**Maximum Cut (MCut).** Given $G$, a *cut* $C = (S, \overline{S})$ partitions the node set $\mathcal{V}$ into two disjoint subsets. The objective is to maximize $\sum_{i \in S, j \in \overline{S}} \mathbf{C}_{ij}$, where $\mathbf{C}$ is the adjacency matrix of $G$.

**Energy functions.** Following previous works [19, 26, 78], the energy function formulations for the four node-oriented problems are defined as follows, where $\beta$ is the constraint penalty coefficient.

- *MIS.* $H(x) = -\sum_{i=1}^{N} x_i + \beta \cdot \sum_{(i,j) \in \mathcal{E}} x_i x_j$
- *MCl.* $H(x) = -\sum_{i=1}^{N} x_i + \beta \cdot \sum_{(i,j) \notin \mathcal{E}} x_i x_j$
- *MVC.* $H(x) = \sum_{i=1}^{N} x_i + \beta \cdot \sum_{(i,j) \in \mathcal{E}} (1 - x_i) \cdot (1 - x_j)$
- *MCut.* $H(x) = \sum_{(i,j) \in \mathcal{E}} (2x_i - 1) \cdot (2x_j - 1)$

## B Discussion of Post-Processing Methods

### B.1 K-Opt Optimization

**Overview.** K-Opt is a classic post-processing method for routing problems, which includes an iterative process. In each iteration, the algorithm attempts to select a subset of nodes and swap the edges between them. If such an operation is found to yield benefits, such as reducing the total distance of the route, it will be executed. The iteration terminates when either no further improvements can be made or the maximum number of iterations (default: 5000) is reached.

**Two-Opt for TSP.** Given an initial tour $\mathbf{x}_1, ..., \mathbf{x}_p, \mathbf{x}_{p+1}, ..., \mathbf{x}_{q-1}, \mathbf{x}_q, \mathbf{x}_{q+1}, ...$ and distance matrix D, two nodes $\mathbf{x}_p$ and $\mathbf{x}_q$ are selected to perform the Two-Opt operation, i.e., connecting the two nodes and swapping the subsequence between them and resulting a new path $\mathbf{x}_1, ..., \mathbf{x}_p, \mathbf{x}_q, \mathbf{x}_{q-1}..., \mathbf{x}_{p+1}, \mathbf{x}_{q+1}$. The reward $r = \mathrm{D}_{p,p+1} + \mathrm{D}_{q,q+1} - \mathrm{D}_{p,q} - \mathrm{D}_{p+1,q+1}$ denotes the improvement after the swap.

**Two-Opt for ATSP.** Given an initial tour $\mathbf{x}_1, ..., \mathbf{x}_p, \mathbf{x}_{p+1}, ..., \mathbf{x}_{q-1}, \mathbf{x}_q, \mathbf{x}_{q+1}, ...,$ Two-Opt performing on nodes $\mathbf{x}_p$ and $\mathbf{x}_q$ results in a new path $\mathbf{x}_1, ..., \mathbf{x}_p, \mathbf{x}_q, \mathbf{x}_{p+1}..., \mathbf{x}_{q-1}, \mathbf{x}_{q+1}$ with reward $r = \mathrm{D}_{p,p+1} + \mathrm{D}_{q-1,q} + \mathrm{D}_{q,q+1} - \mathrm{D}_{p,q} - \mathrm{D}_{q,p+1} - \mathrm{D}_{q-1,q+1}$.

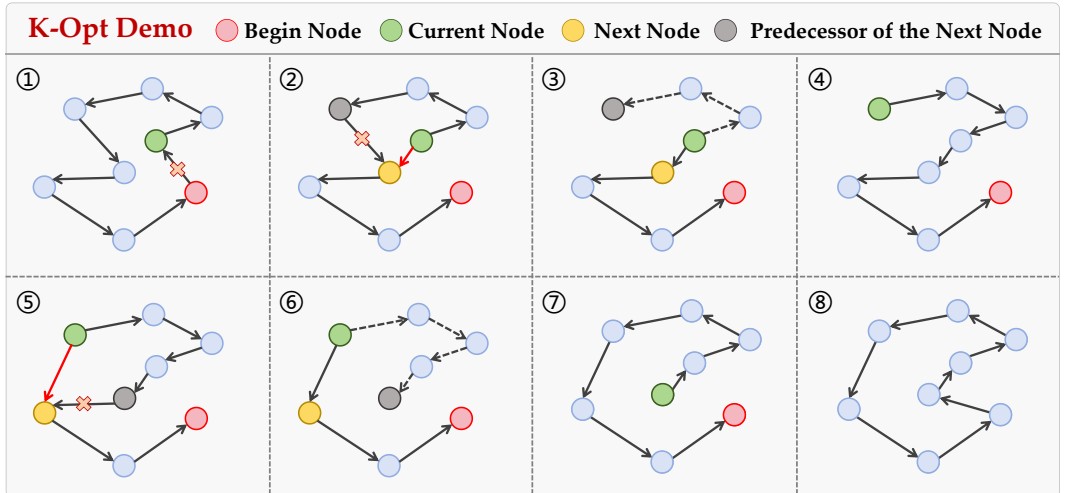

Figure 2: Demonstration of the K-Opt (K=3) post-processing method for TSP.

**K-Opt for TSP.** Given an initial tour $\mathbf{x}_1, ..., \mathbf{x}_p, \mathbf{x}_{p+1}, ..., \mathbf{x}_{q-1}, \mathbf{x}_q, \mathbf{x}_{q+1}, ...$, we start with breaking the edge between $x_p$ (begin node) and $x_{p+1}$ and designate $x_{p+1}$ as the *current node*, which has only successor node but no predecessor node. Then, the K-Opt (we take $K = 3$ as an example, as shown in Fig. 2) process will be carried out. In each iteration, the algorithm needs to first select the next node to connect to the current node $x_{q+1}$, which we refer to as *next node*. Since the new edge is from $x_{p+1}$ to $x_{q+1}$, we need to disconnect the edge between $x_{q+1}$ and its predecessor node $x_q$. After that, the algorithm will reverse all of the edges between $x_{p+1}$ and $x_{q+1}$. At this point, the first iteration is completed, and the current node has been updated from $x_{p+1}$ to $x_{q+1}$. When all the iterations are completed, it is only necessary to connect the *begin node* with the *current node* to re-form a valid path. The most crucial part of K-Opt is the selection of the *next node*, and the most famous method is MCTS [29], using the probability heatmap predicted by a neural network as guidance.

**Parameter Selection for MCTS.** We follow ML4TSP-Bench [17] to set the maximum depth $K$ to 10 and the maximum number of iterations to 5000. As for the time limit, we take into account a comprehensive set of factors, including: 1) recommendations for ML4TSP-Bench [17], 2) our experimental environment (see Appendix C), 3) the solving time of the baseline solver, and 4) the proportion of post-processing time compared to the overall time. We set the time limits for TSP-50, TSP-100, TSP-500, and TSP-1000 to be 0.05s, 0.05s, 1s, and 1s, respectively.

### B.2 Sampling Optimization

Sampling optimization refers to a class of methods based on random sampling, where the core idea is to draw samples from a target distribution to gradually approximate the optimal solution. In recent years, the incorporation of gradient information and parallel computing has significantly enhanced both the efficiency and solution quality of these methods.

iSCO [52] is a sampling-based optimization method built upon enhanced Langevin dynamics. It improves sampling efficiency through parallel neighborhood exploration and efficient gradient computation. Moreover, iSCO dynamically balances exploration and exploitation by adapting temperature parameters and sampling step sizes throughout the optimization process.

RLSA [53] integrates regularized Langevin dynamics with simulated annealing. To effectively escape local optima, RLSA enforces a desired distance between the sampled solution and the current state during the sampling process. Its core mechanism involves using regularization to control the update step size, thereby enabling more efficient exploration of the solution space in discrete domains. The RLSA framework combines the temperature annealing schedule of simulated annealing with the gradient-guided sampling of Langevin dynamics, gradually approximating the target distribution as the temperature decreases. Algorithm 1 illustrates how RLSA works, where $\tau_0$ denotes the initial temperature; $d$ represents the regularization term, where the $d$-th largest gradient value is used to

control the overall flipping probability; $k$ indicates the number of parallel runs, which must be greater than $d$; $t$ represents the total number of iterations.

In this paper, we follow COExpander [37] to use RLSA as a representative sampling method to serve as a post-processor for node-oriented problems, as shown in Algorithm 2, where $\alpha_2$ denotes the noise factor. The RLSA-related parameters used for each problem are presented in Table 10.

Table 10: RLSA Parameters adopted for different problems

| PROBLEM | DATASET | RLSA PARAMETERS | | | | | |
|---|---|---|---|---|---|---|---|
| | | $\tau_0$ | $d$ | $k$ | $t$ | $\beta$ | $\alpha_2$ |
| MIS | RB-SMALL | 0.01 | 5 | 1000 | 500 | 1.02 | 0.3 |
| MIS | RB-LARGE | 0.01 | 5 | 1000 | 1000 | 1.02 | 0.3 |
| MIS | ER-700-800 | 0.2 | 10 | 1000 | 1000 | 1.001 | 0.3 |
| MIS | SATLIB | 0.01 | 2 | 1000 | 1000 | 1.02 | 0.3 |
| MIS | ER-1400-1600 | 0.2 | 10 | 1000 | 1000 | 1.001 | 0.3 |
| MIS | RB-GIANT | 0.01 | 5 | 1000 | 1000 | 1.02 | 0.3 |
| MCl | RB-SMAL | 0.01 | 2 | 200 | 50 | 1.02 | 0.3 |
| MCl | RB-LARGE | 0.01 | 2 | 200 | 200 | 1.02 | 0.3 |
| MCl | TWITTER | 0.01 | 2 | 200 | 50 | 4.0 | 0.3 |
| MCl | COLLAB | 0.01 | 2 | 200 | 20 | 1.001 | 0.3 |
| MCl | RB-GIANT | 0.01 | 2 | 1000 | 1000 | 1.02 | 0.3 |
| MVC | RB-SMALL | 0.01 | 2 | 1000 | 1000 | 1.02 | 0.3 |
| MVC | RB-LARGE | 0.01 | 2 | 1000 | 1000 | 1.02 | 0.3 |
| MVC | TWITTER | 0.01 | 2 | 1000 | 200 | 4.0 | 0.3 |
| MVC | COLLAB | 0.01 | 2 | 1000 | 300 | 1.02 | 0.3 |
| MVC | RB-GIANT | 0.01 | 2 | 1000 | 1000 | 1.02 | 0.3 |
| MCut | BA-SMALL | 1 | 20 | 1000 | 1000 | – | 0.3 |
| MCut | BA-LARGE | 1 | 50 | 1000 | 1000 | – | 0.3 |
| MCut | BA-GIANT | 1 | 100 | 1000 | 1000 | – | 0.3 |

---

**Algorithm 1** Original RLSA.

**Input:** Graph $G$, Energy Function $H(\cdot)$, hyperparameter $(\tau_0, d, k, t)$.
Parallel $k$:
$\mathbf{x} \leftarrow \text{Bern}(p = 0.5)$
**for** $s = 1$ to $t$ **do**
    $\tau \leftarrow \tau_0(1 - \frac{s-1}{t})$
    $\Delta \leftarrow (2\mathbf{x} - 1) \odot \nabla H(\mathbf{x})$
    **for** $i = 1$ to $N$ **do**
        $p \leftarrow \text{Sigmoid}\left((\Delta_i - \Delta_{(d)})/(2\tau)\right)$
        $c \sim \text{Bern}(p)$
        $\mathbf{x}_i \leftarrow \mathbf{x}_i \cdot (1 - c) + (1 - \mathbf{x}_i) \cdot c$
    **end for**
    **if** $H(\mathbf{x}) < H(\mathbf{x}^*)$ **then**
        $\mathbf{x}^* \leftarrow \mathbf{x}$
    **end if**
**end for**
**Output: Solution** $\mathbf{x}$

---

**Algorithm 2** RLSA as Local Search.

**Input:** Graph $G$, Energy Function $H(\cdot)$, hyperparameter $(\tau_0, d, k, t, \alpha_2)$, Initial Solution $\mathbf{x}_0$.
Parallel $k$:
$\mathbf{x} \leftarrow \text{Bern}((1 - \alpha_2) \cdot \mathbf{x}_0 + \alpha_2 \cdot \text{Bern}(p = 0.5))$
**for** $s = 1$ to $t$ **do**
    $\tau \leftarrow \tau_0(1 - \frac{s-1}{t})$
    $\Delta \leftarrow (2\mathbf{x} - 1) \odot \nabla H(\mathbf{x})$
    **for** $i = 1$ to $N$ **do**
        $p \leftarrow \text{Sigmoid}\left((\Delta_i - \Delta_{(d)})/(2\tau)\right)$
        $c \sim \text{Bern}(p)$
        $\mathbf{x}_i \leftarrow \mathbf{x}_i \cdot (1 - c) + (1 - \mathbf{x}_i) \cdot c$
    **end for**
    **if** $H(\mathbf{x}) < H(\mathbf{x}^*)$ **then**
        $\mathbf{x}^* \leftarrow \mathbf{x}$
    **end if**
**end for**
**Output: Solution** $\mathbf{x}$

---

## C  Model Settings and Benchmark Datasets

### C.1  Hardware.

All models are trained and tested using NVIDIA H800 (80G) GPUs and an Intel(R) Xeon(R) Platinum 8558 96-Core Processor CPU. For all test evaluations, a single GPU is utilized, and the batch size is set to 1 to ensure a fair comparison of the solving time across different models.

## C.2 Training Settings and Hyper-parameters.

In principle, the network hyperparameters for each method are adopted according to their original publications, and efforts are made to keep the training hyperparameters and dataset sizes as consistent as possible. Typically, for neural solvers such as **GP-OS-SL**, **GP-OS-UL**, **GP-Gen-SL**, **AE-Gen-SL**, their model architecture and training settings (such as learning rate and training epochs) are strictly aligned with those of COExpander [37], and below are other cases that deviate from the general setting.

- *GP-OS-UL* solver for TSP. Following UTSP [28], we adopt SAG as the backbone, consisting of three layers with a hidden dimension of 64.
- *GP-OS-MAML* solver for TSP. Following DIMES [18], a three-layer network is employed. The hidden dimension is set to 32 for sparse problems and 64 for dense problems.
- *GP-OS-MAML* solver for node-oriented problems. Following Meta-EGN [20], We adopt a 4-layer convolutional network (specifically, 6 layers for MIS), with a hidden dimension of 64.
- *GP-Gen-UL* solver. We use the open-source code [5] and pre-trained checkpoints of DiffUCO [19].
- *LC-OS-RL* solver for TSP and CVRP. Following RL4CO [41] and ML4TSP-Bench [17], we adopt a 3-layer GAT with a hidden dimension of 128 and 8 attention heads.
- *LC-OS-RL* solver for ATSP. Following MatNet [31], we adopt a 5-layer GAT with a hidden dimension of 256 and 16 attention heads.
- *LC-OS-SL* solver. Following GOAL [32], we adopt a 9-layer GAT with a hidden dimension of 128 and 8 attention heads.
- *AE-OS-RL* solver for MIS. Following LwD [43], We adopt a 4-layer convolutional network, with a hidden dimension of 128.

## C.3 Cross-distribution Generalization Datasets.

**ATSP.** We reduce both the Hamiltonian Cycle Problem (HCP) and 3-SAT to ATSP, treating them as two distinct distributions. For HCP, we consider four scales: *HCP-50*, *HCP-100*, *HCP-200*, and *HCP-500*. For 3-SAT, given a formula with $N_v$ variables and $N_c$ clauses, we transform it into an ATSP instance containing $(2N_vN_c + N_c)$ nodes. Specifically, we set $N_v \in \{4, 8, 12, 19\}$ and $N_c \in \{6, 6, 8, 13\}$, resulting in *SAT-54*, *SAT-102*, *SAT-200*, *SAT-507*, respectively.

**TSP.** We adopt Cluster and Gaussian distributions as generalization datasets. The Cluster TSP simulates scenarios where nodes are grouped into distinct spatial regions. Specifically, $C$ cluster centers are uniformly sampled within the unit square $[0,1]^2$, and nodes are then drawn from Gaussian distributions centered at these locations, with a cluster-level standard deviation $\sigma_c$ (we use 0.03) as a hyperparameter. We generate *Cluster-50*, *Cluster-100*, *Cluster-200*, *Cluster-500* with corresponding $C$ values of 10, 20, 20, and 25. The Gaussian TSP models real-world settings where locations concentrate around a central hub, such as delivery points near a warehouse or demand centers in dense urban areas. Node coordinates are sampled from a 2D Gaussian distribution $\mathcal{N}\big((\mu_x, \mu_y), \sigma^2 I\big)$, where $(\mu_x, \mu_y)$ is the mean and $\sigma$ controls the spatial spread, with $I$ denoting the identity matrix. We set $\mu_x = 0, \mu_y = 0, \sigma = 1$ for *Gaussian-50*, *Gaussian-100*, *Gaussian-200*, and *Gaussian-500*.

**Node-oriented Tasks.** We adopt the HK [71] and WS [72] graph models. The *HK graph* starts from a small fully connected core, where each new node attaches to $N_d$ existing nodes via preferential attachment and, with probability $p$, additionally connects to one of their neighbors, forming triangles. This mechanism produces networks with power-law degree distributions and high clustering, suitable for representing social or biological systems. The *WS graph* captures small-world properties by balancing high clustering and short path lengths. It begins with a regular ring lattice where each node is connected to its $N_k$ nearest neighbors, and with probability $p$, each edge is rewired to a random node while maintaining connectivity. The resulting structure combines local cohesion with global efficiency, mirroring many real-world networks. We set $N_d = 10$, $N_k = 10$, and $p = 0.3$ for both graph types, and generate four datasets: *HK-SMALL*, *HK-LARGE*, *WS-SMALL*, and *WS-LARGE*. Consistent with RB graphs, *SMALL* denotes instances with 200 to 300 nodes, and *LARGE* denotes instances with 800 to 1200 nodes.

---

[5] https://github.com/ml-jku/DIffUCO

## C.4 Benchmark Datasets.

We provide open-source datasets for each benchmark, including 24 training datasets and 65 test datasets, as shown in Table 11.

# D Supplementary Experiments

We present the complete results of TSP in Table 12, ATSP in Table 13, CVRP in Table 14, MIS in Tabel 15, MCl in Table 16, MVC in Table 17, MCut in Table 18 respectively.

# E Limitations and Broader Impacts

## E.1 Limitations

**Solving methods.** As stated in the "Statement on Topics *not* Discussed" part of Section. 3, our benchmarking work is currently limited to the solution approach that first obtains solutions using machine learning methods and then optimize them with heuristic algorithms. We have not yet included other solution approaches, such as *D&C* and *learning to search*.

**Problem types.** Currently, more than 30 types of COPs have been systematically studied and organized[6]. In this paper, we only discuss seven of them (TSP, ATSP, CVRP, MIS, MCl, MVC, MCut), and we plan to extend our benchmarking work to include other problems (such as TSPTW, CVRPTW, JSSP) in the future.

## E.2 Broader Impacts

Our work systematically organizes ML4CO using an *"paradigm-model-learning"* framework, covering seven major problem types. To our best knowledge, it is the first in the ML4CO field to openly release as many as 34 datasets (over 100GB) in a structured and comprehensive manner. We believe that our work can bring standardized evaluation and facilitate future development for ML4CO.

---

[6]https://github.com/Thinklab-SJTU/awesome-ml4co

Table 11: Overview of our benchmark datasets for the seven COPs. The "MODEL" column indicates the dataset on which the model is trained. I.e., datasets with the same "TYPE" and "MODEL" is suggested for i.i.d. training-testing, whereas datasets with different "TYPE" and "MODEL" (with "testing" part only) are designed for evaluating the o.o.d. generalization performance of models trained on the corresponding (smaller-scaled, differently distributed, etc.) data. The parentheses indicate the solver parameters used: the maximum solution time for Gurobi, KaMIS and HGS; the maximum trials for LKH. Blue: Cross-distribution Generalization Dataset.

| ID | PROBLEM | TYPE | MODEL | Training | | | Testing | | |
|---|---|---|---|---|---|---|---|---|---|
| | | | | DATA SIZE | SOLVER | STORAGE | DATA SIZE | SOLVER | OBJ. |
| 1 | TSP | Uniform-50 | Uniform-50 | 1,280,000 | Concorde | 2.48GB | 1280 | Concorde | 5.688 |
| 2 | TSP | Uniform-100 | Uniform-100 | 1,280,000 | Concorde | 4.95GB | 1280 | Concorde | 7.756 |
| 3 | TSP | Uniform-200 | Uniform-200 | 128,000 | Concorde | 1.05GB | 128 | Concorde | 10.719 |
| 4 | TSP | Uniform-500 | Uniform-500 | 64,000 | Concorde | 1.26GB | 128 | Concorde | 16.546 |
| 5 | TSP | Uniform-1K | Uniform-1K | 64,000 | LKH(1000) | 2.53GB | 128 | Concorde | 23.118 |
| 6 | TSP | Uniform-10K | Uniform-10K | 6,400 | LKH(500) | 2.59GB | 16 | LKH(500) | 71.755 |
| 7 | TSP | TSPLIB | Mixed | – | – | – | 49 | Concorde | 8.062 |
| 8 | TSP | Cluster-50 | Uniform-50 | – | – | – | 1280 | Concorde | 3.730 |
| 9 | TSP | Cluster-100 | Uniform-100 | – | – | – | 1280 | Concorde | 5.526 |
| 10 | TSP | Cluster-200 | Uniform-200 | – | – | – | 128 | Concorde | 6.912 |
| 11 | TSP | Cluster-500 | Uniform-500 | – | – | – | 128 | Concorde | 10.723 |
| 12 | TSP | Gaussian-50 | Uniform-50 | – | – | – | 1280 | Concorde | 23.840 |
| 13 | TSP | Gaussian-100 | Uniform-100 | – | – | – | 1280 | Concorde | 34.031 |
| 14 | TSP | Gaussian-200 | Uniform-200 | – | – | – | 128 | Concorde | 48.127 |
| 15 | TSP | Gaussian-500 | Uniform-500 | – | – | – | 128 | Concorde | 77.521 |
| 16 | ATSP | Uniform-50 | Uniform-50 | 640,000 | LKH(500) | 14.72GB | 2500 | LKH(1000) | 1.5545 |
| 17 | ATSP | Uniform-100 | Uniform-100 | 128,000 | LKH(500) | 11.78GB | 2500 | LKH(1000) | 1.5660 |
| 18 | ATSP | Uniform-200 | Uniform-200 | 32,000 | LKH(1000) | 11.76GB | 100 | LKH(1000) | 1.5647 |
| 19 | ATSP | Uniform-500 | Uniform-500 | 6,400 | LKH(1000) | 14.70GB | 100 | LKH(1000) | 1.5734 |
| 20 | ATSP | HCP-50 | Uniform-50 | – | – | – | 2500 | – | 0.0000 |
| 21 | ATSP | HCP-100 | Uniform-100 | – | – | – | 2500 | – | 0.0000 |
| 22 | ATSP | HCP-200 | Uniform-200 | – | – | – | 100 | – | 0.0000 |
| 23 | ATSP | HCP-500 | Uniform-500 | – | – | – | 100 | – | 0.0000 |
| 24 | ATSP | SAT-54 | Uniform-50 | – | – | – | 2500 | – | 0.0000 |
| 25 | ATSP | SAT-102 | Uniform-100 | – | – | – | 2500 | – | 0.0000 |
| 26 | ATSP | SAT-200 | Uniform-200 | – | – | – | 100 | – | 0.0000 |
| 27 | ATSP | SAY-507 | Uniform-500 | – | – | – | 100 | – | 0.0000 |
| 28 | CVRP | Uniform-50 | Uniform-50 | 1,280,000 | HGS(1s) | 2.83GB | 10,000 | HGS(1s) | 10.366 |
| 29 | CVRP | Uniform-100 | Uniform-100 | 640,000 | HGS(20s) | 1.11GB | 10,000 | HGS(20s) | 15.563 |
| 30 | CVRP | Uniform-200 | Uniform-200 | 128,000 | HGS(60s) | 1.11GB | 100 | HGS(60s) | 19.630 |
| 31 | CVRP | Uniform-500 | Uniform-500 | 12,800 | HGS(360s) | 285MB | 100 | HGS(360s) | 37.154 |
| 32 | CVRP | CVRPLIB | Mixed | – | – | – | 70 | – | 45.183 |
| 33 | MIS | RB-SMALL | RB-SMALL | 64,000 | KaMIS(10s) | 3.52GB | 500 | KaMIS(60s) | 20.090 |
| 34 | MIS | RB-LARGE | RB-LARGE | 6,400 | KaMIS(60s) | 4.74GB | 500 | KaMIS(60s) | 43.004 |
| 35 | MIS | RB-GIANT | RB-LARGE | – | – | – | 50 | KaMIS(60s) | 49.260 |
| 36 | MIS | ER-700-800 | ER-700-800 | 12,800 | KaMIS(60s) | 7.83GB | 128 | KaMIS(60s) | 44.969 |
| 37 | MIS | ER-1400-1600 | ER-700-800 | – | – | – | 128 | KaMIS(60s) | 50.938 |
| 38 | MIS | SATLIB | SATLIB | 39,500 | KaMIS(60s) | 3.75GB | 500 | KaMIS(60s) | 425.954 |
| 39 | MIS | HK-SMALL | ER-700-800 | – | – | – | 500 | KaMIS(60s) | 79.372 |
| 40 | MIS | HK-LARGE | ER-700-800 | – | – | – | 500 | KaMIS(60s) | 330.946 |
| 41 | MIS | WS-SMALL | ER-700-800 | – | – | – | 500 | KaMIS(60s) | 76.904 |
| 42 | MIS | WS-LARGE | ER-700-800 | – | – | – | 500 | KaMIS(60s) | 262.570 |
| 43 | MCl | RB-SMALL | RB-SMALL | 64,000 | Gurobi(60s) | 3.42GB | 500 | Gurobi(60s) | 19.082 |
| 44 | MCl | Twitter | RB-SMALL | – | – | – | 195 | Gurobi(60s) | 14.210 |
| 45 | MCl | COLLAB | RB-SMALL | – | – | – | 1000 | Gurobi(60s) | 42.113 |
| 46 | MCl | RB-LARGE | RB-LARGE | 6,400 | Gurobi(300s) | 4.74GB | 500 | Gurobi(300s) | 40.182 |
| 47 | MCl | RB-GIANT | RB-LARGE | – | – | – | 50 | Gurobi(3600s) | 81.520 |
| 48 | MCl | HK-SMALL | RB-LARGE | – | – | – | 500 | Gurobi(60s) | 6.792 |
| 49 | MCl | HK-LARGE | RB-LARGE | – | – | – | 500 | Gurobi(300s) | 6.774 |
| 50 | MCl | WS-SMALL | RB-LARGE | – | – | – | 500 | Gurobi(60s) | 7.164 |
| 51 | MCl | WS-LARGE | RB-LARGE | – | – | – | 500 | Gurobi(300s) | 5.978 |
| 52 | MVC | RB-SMALL | RB-SMALL | 128,000 | Gurobi(60s) | 7.01GB | 500 | Gurobi(60s) | 205.764 |
| 53 | MVC | Twitter | RB-SMALL | – | – | – | 195 | Gurobi(60s) | 85.251 |
| 54 | MVC | COLLAB | RB-SMALL | – | – | – | 1000 | Gurobi(60s) | 65.086 |
| 55 | MVC | RB-LARGE | RB-LARGE | 6,400 | Gurobi(300s) | 4.74GB | 500 | Gurobi(300s) | 968.228 |
| 56 | MVC | RB-GIANT | RB-LARGE | – | – | – | 50 | Gurobi(300s) | 2398.480 |
| 57 | MVC | HK-SMALL | RB-SMALL | – | – | – | 500 | Gurobi(60s) | 142.506 |
| 58 | MVC | WS-SMALL | RB-SMALL | – | – | – | 500 | Gurobi(60s) | 154.618 |
| 59 | MCut | BA-SMALL | BA-SMALL | 128,000 | Gurobi(60s) | 1.78GB | 500 | Gurobi(60s) | 727.844 |
| 60 | MCut | BA-LARGE | BA-LARGE | 128,000 | Gurobi(300s) | 8.08GB | 500 | Gurobi(300s) | 2936.886 |
| 61 | MCut | BA-GIANT | BA-LARGE | – | – | – | 50 | Gurobi(300s) | 7217.900 |
| 62 | MCut | HK-SMALL | BA-SMALL | – | – | – | 500 | Gurobi(60s) | 1540.608 |
| 63 | MCut | HK-LARGE | BA-LARGE | – | – | – | 500 | Gurobi(300s) | 6401.320 |
| 64 | MCut | WS-SMALL | BA-SMALL | – | – | – | 500 | Gurobi(60s) | 872.116 |
| 65 | MCut | WS-LARGE | BA-LARGE | – | – | – | 500 | Gurobi(300s) | 3454.176 |

Table 12: Complete results on TSP.

| METHOD | TYPE | SOLVING STAGE | | TSP-50 | | | TSP-100 | | |
| --- | --- | --- | --- | --- | --- | --- | --- | --- | --- |
| | | INITIALIZATION | OPTIMIZATION | OBJ.↓ | DROP↓ | TIME↓ | OBJ.↓ | DROP↓ | TIME↓ |
| Concorde [73] | Exact | – | – | 5.688* | 0.000±0.000% | 0.059s | 7.756* | 0.000±0.000% | 0.238s |
| LKH (500) [6] | Heuristics | Random | K-Opt | 5.688 | 0.001±0.014% | 0.058s | 7.756 | 0.001±0.014% | 0.176s |
| GA-EAX [74] | Heuristics | Random | K-Opt + EAX | 5.688 | 0.000±0.001% | 0.101s | 7.756 | 0.000±0.004% | 1.862s |
| GP4CO [39] | GP-OS-SL | Greedy | – | 5.718 | 0.520±1.918% | 0.006s | 7.937 | 2.334±4.140% | 0.007s |
| GP4CO [39] | GP-OS-SL | Greedy | Two-Opt | 5.691 | 0.066±0.316% | 0.007s | 7.775 | 0.243±0.546% | 0.009s |
| GP4CO [39] | GP-OS-SL | Greedy | MCTS (0.05s) | 5.688 | 0.012±0.068% | 0.007s | 7.762 | 0.073±0.224% | 0.013s |
| GP4CO [39] | GP-OS-SL | Sampling | MCTS (0.05s) | 5.688 | 0.004±0.031% | 0.057s | 7.758 | 0.022±0.083% | 0.058s |
| GP4CO [39] | GP-OS-SL | Random-16 | MCTS (0.05s) | 5.688 | 0.001±0.010% | 0.031s | 7.756 | 0.005±0.025% | 0.119s |
| GP4CO [18] | GP-OS-MAML(RL) | Greedy | – | 6.325 | 11.233±4.503% | 0.003s | 8.757 | 12.918±3.761% | 0.005s |
| GP4CO [18] | GP-OS-MAML(RL) | Greedy | Two-Opt | 5.891 | 3.578±2.361% | 0.007s | 8.108 | 4.543±1.838% | 0.012s |
| GP4CO [18] | GP-OS-MAML(RL) | Greedy | MCTS (0.05s) | 5.755 | 1.188±1.420% | 0.008s | 7.897 | 1.819±1.234% | 0.025s |
| GP4CO [18] | GP-OS-MAML(RL) | Sampling | MCTS (0.05s) | 5.696 | 0.153±0.389% | 0.054s | 7.853 | 1.254±0.933% | 0.059s |
| GP4CO [18] | GP-OS-MAML(RL) | Random-16 | MCTS (0.05s) | 5.690 | 0.034±0.145% | 0.111s | 7.783 | 0.348±0.386% | 0.571s |
| GP4CO [28] | GP-OS-UL | Greedy | – | 7.097 | 24.808±7.728% | 0.007s | 9.377 | 20.909±5.049% | 0.009s |
| GP4CO [28] | GP-OS-UL | Greedy | Two-Opt | 5.976 | 5.080±3.065% | 0.012s | 8.161 | 5.215±2.140% | 0.017s |
| GP4CO [28] | GP-OS-UL | Greedy | MCTS (0.05s) | 5.862 | 3.078±2.395% | 0.028s | 8.052 | 3.817±1.852% | 0.065s |
| GP4CO [28] | GP-OS-UL | Sampling | MCTS (0.05s) | 5.818 | 2.295±1.809% | 0.062s | 8.270 | 6.630±2.376% | 0.067s |
| GP4CO ($I_s$=1) [40] | GP-Gen-SL | Greedy | – | 5.723 | 0.612±2.389% | 0.007s | 7.941 | 2.381±4.385% | 0.009s |
| GP4CO ($I_s$=1) [40] | GP-Gen-SL | Greedy | Two-Opt | 5.691 | 0.065±0.344% | 0.007s | 7.773 | 0.223±0.529% | 0.009s |
| GP4CO ($I_s$=5) [40] | GP-Gen-SL | Greedy | – | 5.691 | 0.061±0.119% | 0.030s | 7.774 | 0.230±0.888% | 0.031s |
| GP4CO ($I_s$=5) [40] | GP-Gen-SL | Greedy | Two-Opt | 5.689 | 0.029±0.119% | 0.030s | 7.762 | 0.078±0.197% | 0.031s |
| GP4CO ($I_s$=5) [40] | GP-Gen-SL | 4×Greedy | – | 5.688 | 0.008±0.041% | 0.031s | 7.758 | 0.029±0.085% | 0.064s |
| GP4CO ($I_s$=5) [40] | GP-Gen-SL | 4×Greedy | Two-Opt | 5.688 | 0.006±0.031% | 0.031s | 7.757 | 0.017±0.053% | 0.065s |
| LC4CO [41] | LC-OS-RL | Greedy | – | 5.702 | 0.258±0.374% | 0.050s | 7.872 | 1.491±0.767% | 0.098s |
| LC4CO [41] | LC-OS-RL | Greedy | Two-Opt | 5.699 | 0.197±0.339% | 0.051s | 7.840 | 1.087±0.714% | 0.100s |
| LC4CO [41] | LC-OS-RL | N×Greedy | – | 5.697 | 0.165±0.300% | 0.056s | 7.851 | 1.222±0.625% | 0.110s |
| LC4CO [41] | LC-OS-RL | N×Greedy | Two-Opt | 5.694 | 0.119±0.239% | 0.057s | 7.824 | 0.881±0.604% | 0.111s |
| LC4CO [41] | LC-OS-RL | N×Sampling | – | 5.697 | 0.164±0.281% | 0.061s | 7.853 | 1.252±0.634% | 0.120s |
| LC4CO [41] | LC-OS-RL | N×Sampling | Two-Opt | 5.694 | 0.113±0.231% | 0.061s | 7.823 | 0.860±0.587% | 0.121s |
| LC4CO [32] | LC-OS-SL | Greedy | – | 5.742 | 0.965±1.224% | 0.393s | 7.905 | 1.920±2.182% | 0.762s |
| LC4CO [32] | LC-OS-SL | Greedy | Two-Opt | 5.735 | 0.824±1.022% | 0.394s | 7.862 | 1.369±1.139% | 0.764s |
| LC4CO [32] | LC-OS-SL | Beam-16 | – | 5.701 | 0.233±0.442% | 0.462s | 7.787 | 0.398±0.421% | 0.995s |
| LC4CO [32] | LC-OS-SL | Beam-16 | Two-Opt | 5.697 | 0.171±0.362% | 0.463s | 7.784 | 0.358±0.404% | 0.995s |
| AE4CO ($D_s$=3, $I_s$=1) [37] | AE-Gen-SL | Greedy | – | 5.702 | 0.244±1.205% | 0.018s | 7.846 | 1.164±2.924% | 0.020s |
| AE4CO ($D_s$=3, $I_s$=1) [37] | AE-Gen-SL | Greedy | Two-Opt | 5.689 | 0.029±0.145% | 0.018s | 7.765 | 0.124±0.361% | 0.021s |
| AE4CO ($D_s$=3, $I_s$=5) [37] | AE-Gen-SL | Greedy | – | 5.689 | 0.031±0.111% | 0.096s | 7.761 | 0.073±0.191% | 0.102s |
| AE4CO ($D_s$=3, $I_s$=5) [37] | AE-Gen-SL | Greedy | Two-Opt | 5.689 | 0.025±0.081% | 0.096s | 7.760 | 0.058±0.121% | 0.102s |
| AE4CO ($D_s$=3, $I_s$=5) [37] | AE-Gen-SL | 4×Greedy | – | 5.688 | 0.005±0.029% | 0.101s | 7.757 | 0.015±0.045% | 0.187s |
| AE4CO ($D_s$=3, $I_s$=5) [37] | AE-Gen-SL | 4×Greedy | Two-Opt | 5.688 | 0.004±0.028% | 0.101s | 7.757 | 0.013±0.043% | 0.191s |

| METHOD | TYPE | SOLVING STAGE | | TSP-500 | | | TSP-1K | | |
| --- | --- | --- | --- | --- | --- | --- | --- | --- | --- |
| | | INITIALIZATION | OPTIMIZATION | OBJ.↓ | DROP↓ | TIME↓ | OBJ.↓ | DROP↓ | TIME↓ |
| Concorde [73] | Exact | – | – | 16.546* | 0.000±0.000% | 18.672s | 23.118* | 0.000±0.000% | 84.413s |
| LKH (500) [6] | Heuristics | Random | K-Opt | 16.546 | 0.002±0.010% | 1.848s | 23.119 | 0.005±0.011% | 4.641s |
| GA-EAX [74] | Heuristics | Random | K-Opt + EAX | 16.546 | 0.001±0.004% | 1.857s | 23.118 | 0.000±0.001% | 17.544s |
| GP4CO [39] | GP-OS-SL | Greedy | – | 18.148 | 9.683±3.880% | 0.047s | 26.124 | 13.001±2.698% | 0.195s |
| GP4CO [39] | GP-OS-SL | Greedy | Two-Opt | 16.769 | 1.348±0.589% | 0.063s | 23.527 | 1.769±0.434% | 0.227s |
| GP4CO [39] | GP-OS-SL | Greedy | MCTS (1s) | 16.658 | 0.676±0.429% | 0.367s | 23.380 | 1.131±0.355% | 1.594s |
| GP4CO [39] | GP-OS-SL | Sampling | MCTS (1s) | 16.673 | 0.768±0.437% | 1.063s | 23.621 | 2.176±0.828% | 1.219s |
| GP4CO [39] | GP-OS-SL | Random-16 | MCTS (1s) | 16.648 | 0.617±0.286% | 7.094s | 23.444 | 1.412±0.250% | 21.844s |
| GP4CO [18] | GP-OS-MAML(RL) | Greedy | – | 18.888 | 14.155±1.532% | 0.273s | 26.421 | 14.291±1.063% | 0.546s |
| GP4CO [18] | GP-OS-MAML(RL) | Greedy | Two-Opt | 17.655 | 6.707±1.121% | 0.314s | 24.906 | 7.735±0.823% | 0.662s |
| GP4CO [18] | GP-OS-MAML(RL) | Greedy | MCTS (1s) | 17.776 | 7.435±1.128% | 0.833s | 25.032 | 8.279±0.787% | 3.870s |
| GP4CO [18] | GP-OS-MAML(RL) | Sampling | MCTS (1s) | 17.735 | 7.189±1.219% | 0.917s | 24.947 | 7.912±0.744% | 4.210s |
| GP4CO [18] | GP-OS-MAML(RL) | Random-16 | MCTS (1s) | 18.024 | 8.932±0.778% | 30.610s | 25.477 | 10.204±0.586% | 168.103s |
| GP4CO [28] | GP-OS-UL | Greedy | – | 19.761 | 19.435±1.925% | 0.059s | 27.644 | 19.580±1.645% | 0.223s |
| GP4CO [28] | GP-OS-UL | Greedy | Two-Opt | 17.710 | 7.033±1.059% | 0.090s | 24.765 | 7.122±0.775% | 0.290s |
| GP4CO [28] | GP-OS-UL | Greedy | MCTS (1s) | 17.913 | 8.261±1.130% | 1.574s | 25.069 | 8.440±0.874% | 2.231s |
| GP4CO [28] | GP-OS-UL | Sampling | MCTS (1s) | 18.500 | 11.811±1.388% | 1.535s | 25.961 | 12.999±0.903% | 2.403s |
| GP4CO ($I_s$=1) [40] | GP-Gen-SL | Greedy | – | 18.176 | 9.840±4.093% | 0.050s | 26.039 | 12.632±3.301% | 0.203s |
| GP4CO ($I_s$=1) [40] | GP-Gen-SL | Greedy | Two-Opt | 16.750 | 1.233±0.687% | 0.055s | 23.481 | 1.571±0.412% | 0.211s |
| GP4CO ($I_s$=5) [40] | GP-Gen-SL | Greedy | – | 17.473 | 5.603±3.311% | 0.099s | 25.021 | 8.229±2.728% | 0.289s |
| GP4CO ($I_s$=5) [40] | GP-Gen-SL | Greedy | Two-Opt | 16.690 | 0.869±0.528% | 0.102s | 23.423 | 1.319±0.387% | 0.328s |
| GP4CO ($I_s$=5) [40] | GP-Gen-SL | 4×Greedy | – | 17.012 | 2.817±1.976% | 0.289s | 24.371 | 5.419±1.620% | 1.188s |
| GP4CO ($I_s$=5) [40] | GP-Gen-SL | 4×Greedy | Two-Opt | 16.627 | 0.490±0.294% | 0.320s | 23.339 | 0.954±0.214% | 1.141s |
| LC4CO [32] | LC-OS-SL | Greedy | – | 18.006 | 8.825±3.847% | 3.964s | 25.380 | 9.789±3.224% | 7.995s |
| LC4CO [32] | LC-OS-SL | Greedy | Two-Opt | 17.047 | 3.027±0.840% | 4.057s | 23.806 | 2.974±0.581% | 8.065s |
| LC4CO [32] | LC-OS-SL | Beam-16 | – | 17.279 | 4.433±2.576% | 4.940s | 24.558 | 5.494±1.812% | 17.260s |
| LC4CO [32] | LC-OS-SL | Beam-16 | Two-Opt | 16.923 | 2.278±0.720% | 4.960s | 23.681 | 2.434±0.466% | 17.465s |
| AE4CO ($D_s$=3, $I_s$=1) [37] | AE-Gen-SL | Greedy | – | 17.634 | 6.574±3.456% | 0.062s | 25.408 | 9.904±3.096% | 0.261s |
| AE4CO ($D_s$=3, $I_s$=1) [37] | AE-Gen-SL | Greedy | Two-Opt | 16.684 | 0.837±0.512% | 0.070s | 23.421 | 1.309±0.387% | 0.273s |
| AE4CO ($D_s$=3, $I_s$=5) [37] | AE-Gen-SL | Greedy | – | 17.119 | 3.466±3.012% | 0.231s | 24.698 | 6.834±2.617% | 0.691s |
| AE4CO ($D_s$=3, $I_s$=5) [37] | AE-Gen-SL | Greedy | Two-Opt | 16.626 | 0.487±0.388% | 0.242s | 23.337 | 0.946±0.409% | 0.703s |
| AE4CO ($D_s$=3, $I_s$=5) [37] | AE-Gen-SL | 4×Greedy | – | 16.776 | 1.395±1.851% | 0.650s | 24.081 | 4.165±1.902% | 2.412s |
| AE4CO ($D_s$=3, $I_s$=5) [37] | AE-Gen-SL | 4×Greedy | Two-Opt | 16.588 | 0.257±0.206% | 0.695s | 23.271 | 0.662±0.238% | 2.609s |

| METHOD | TYPE | SOLVING STAGE | | TSP-10K | | |
| --- | --- | --- | --- | --- | --- | --- |
| | | INITIALIZATION | OPTIMIZATION | OBJ.↓ | DROP↓ | TIME↓ |
| LKH (500) [6] | Heuristics | Random | K-Opt | 71.755* | 0.000±0.000% | 332.758s |
| GP4CO [39] | GP-OS-SL | Greedy | – | 107.645 | 50.018±3.885% | 26.409s |
| GP4CO [39] | GP-OS-SL | Greedy | Two-Opt | 73.019 | 1.761±0.166% | 28.294s |
| GP4CO ($I_s$=5) [40] | GP-Gen-SL | Greedy | – | 102.972 | 43.504±3.717% | 26.984s |
| GP4CO ($I_s$=5) [40] | GP-Gen-SL | Greedy | Two-Opt | 72.982 | 1.710±0.120% | 28.313s |
| LC4CO [32] | LC-OS-SL | Greedy | – | 82.249 | 14.627±1.910% | 95.579s |
| LC4CO [32] | LC-OS-SL | Greedy | Two-Opt | 74.595 | 3.958±0.231% | 102.369s |
| AE4CO ($D_s$=3, $I_s$=5) [37] | AE-Gen-SL | Greedy | – | 97.423 | 35.772±3.908% | 28.288s |
| AE4CO ($D_s$=3, $I_s$=5) [37] | AE-Gen-SL | Greedy | Two-Opt | 72.832 | 1.501±0.126% | 33.485s |

Table 13: Complete results on ATSP.

| METHOD | TYPE | SOLVING STAGE | | ATSP-50 | | | ATSP-100 | | |
| --- | --- | --- | --- | --- | --- | --- | --- | --- | --- |
| | | INITIALIZATION | OPTIMIZATION | OBJ.↓ | DROP↓ | TIME↓ | OBJ.↓ | DROP↓ | TIME↓ |
| LKH (1000) [6] | Heuristics | Random | K-Opt | 1.554 | 0.000±0.000% | 0.097s | 1.566 | 0.000±0.000% | 0.238s |
| GP4CO [39] | GP-OS-SL | Greedy | – | 1.667 | 7.261±5.299% | 0.007s | 1.666 | 6.408±3.498% | 0.008s |
| GP4CO [39] | GP-OS-SL | Greedy | Two-Opt | 1.613 | 3.795±2.881% | 0.008s | 1.629 | 4.004±2.095% | 0.011s |
| GP4CO ($I_s$=1) [40] | GP-Gen-SL | Greedy | – | 1.662 | 6.948±5.164% | 0.006s | 1.665 | 6.329±3.377% | 0.008s |
| GP4CO ($I_s$=1) [40] | GP-Gen-SL | Greedy | Two-Opt | 1.614 | 3.805±2.915% | 0.007s | 1.630 | 4.104±2.143% | 0.009s |
| GP4CO ($I_s$=5) [40] | GP-Gen-SL | Greedy | – | 1.602 | 3.043±4.021% | 0.029s | 1.628 | 3.935±2.374% | 0.033s |
| GP4CO ($I_s$=5) [40] | GP-Gen-SL | Greedy | Two-Opt | 1.580 | 1.616±2.294% | 0.029s | 1.608 | 2.665±1.687% | 0.034s |
| GP4CO ($I_s$=5) [40] | GP-Gen-SL | 4×Greedy | – | 1.562 | 0.485±1.412% | 0.033s | 1.595 | 1.827±1.650% | 0.161s |
| GP4CO ($I_s$=5) [40] | GP-Gen-SL | 4×Greedy | Two-Opt | 1.558 | 0.241±0.714% | 0.034s | 1.584 | 1.160±1.119% | 0.167s |
| LC4CO [31] | LC-OS-RL | Greedy | – | 1.576 | 1.373±1.108% | 0.032s | 1.620 | 3.456±1.291% | 0.053s |
| LC4CO [31] | LC-OS-RL | Greedy | Two-Opt | 1.574 | 1.256±1.050% | 0.032s | 1.616 | 3.199±1.225% | 0.057s |
| LC4CO [31] | LC-OS-RL | Beam-16 | – | 1.560 | 0.334±0.456% | 0.038s | 1.592 | 1.653±0.758% | 0.055s |
| LC4CO [31] | LC-OS-RL | Beam-16 | Two-Opt | 1.559 | 0.319±0.444% | 0.038s | 1.591 | 1.586±0.747% | 0.061s |
| LC4CO [32] | LC-OS-SL | Greedy | – | 1.595 | 2.567±2.715% | 0.395s | 1.644 | 5.010±2.098% | 0.723s |
| LC4CO [32] | LC-OS-SL | Greedy | Two-Opt | 1.587 | 2.079±2.058% | 0.396s | 1.632 | 4.195±1.736% | 0.724s |
| LC4CO [32] | LC-OS-SL | Beam-16 | – | 1.588 | 2.164±2.009% | 0.397s | 1.619 | 3.362±1.701% | 0.961s |
| LC4CO [32] | LC-OS-SL | Beam-16 | Two-Opt | 1.581 | 1.695±1.564% | 0.400s | 1.610 | 2.801±1.360% | 0.962s |
| AE4CO ($D_s$=3, $I_s$=1) [37] | AE-Gen-SL | Greedy | – | 1.637 | 5.304±4.226% | 0.019s | 1.646 | 5.086±3.145% | 0.023s |
| AE4CO ($D_s$=3, $I_s$=1) [37] | AE-Gen-SL | Greedy | Two-Opt | 1.601 | 3.029±2.543% | 0.019s | 1.619 | 3.361±1.996% | 0.024s |
| AE4CO ($D_s$=3, $I_s$=5) [37] | AE-Gen-SL | Greedy | – | 1.582 | 1.790±2.964% | 0.086s | 1.617 | 3.233±2.191% | 0.099s |
| AE4CO ($D_s$=3, $I_s$=5) [37] | AE-Gen-SL | Greedy | Two-Opt | 1.572 | 1.125±1.866% | 0.086s | 1.601 | 2.258±1.563% | 0.099s |
| AE4CO ($D_s$=3, $I_s$=5) [37] | AE-Gen-SL | 4×Greedy | – | 1.559 | 0.269±1.023% | 0.101s | 1.588 | 1.393±1.494% | 0.511s |
| AE4CO ($D_s$=3, $I_s$=5) [37] | AE-Gen-SL | 4×Greedy | Two-Opt | **1.557** | **0.171±0.611%** | 0.103s | 1.581 | **0.946±1.034%** | 0.516s |

| METHOD | TYPE | SOLVING STAGE | | ATSP-200 | | | ATSP-500 | | |
| --- | --- | --- | --- | --- | --- | --- | --- | --- | --- |
| | | INITIALIZATION | OPTIMIZATION | OBJ.↓ | DROP↓ | TIME↓ | OBJ.↓ | DROP↓ | TIME↓ |
| LKH (1000) [6] | Heuristics | Random | K-Opt | 1.565 | 0.000±0.000% | 0.724s | 1.573 | 0.000±0.000% | 4.376s |
| GP4CO [39] | GP-OS-SL | Greedy | – | 1.671 | 6.777±3.155% | 0.053s | 1.729 | 9.880±1.887% | 0.157s |
| GP4CO [39] | GP-OS-SL | Greedy | Two-Opt | 1.634 | 4.391±1.822% | 0.070s | 1.696 | 7.774±1.152% | 0.313s |
| GP4CO ($I_s$=1) [40] | GP-Gen-SL | Greedy | – | 1.678 | 7.280±2.475% | 0.037s | 1.675 | 6.443±1.454% | 0.127s |
| GP4CO ($I_s$=1) [40] | GP-Gen-SL | Greedy | Two-Opt | 1.652 | 5.608±1.958% | 0.046s | 1.653 | 5.088±1.098% | 0.231s |
| GP4CO ($I_s$=5) [40] | GP-Gen-SL | Greedy | – | 1.638 | 4.677±2.066% | 0.162s | 1.641 | 4.294±1.663% | 0.558s |
| GP4CO ($I_s$=5) [40] | GP-Gen-SL | Greedy | Two-Opt | 1.615 | 3.252±1.456% | 0.168s | 1.622 | 3.078±1.009% | 0.648s |
| GP4CO ($I_s$=5) [40] | GP-Gen-SL | 4×Greedy | – | 1.608 | 2.772±1.060% | 0.360s | 1.614 | 2.543±0.829% | 2.144s |
| GP4CO ($I_s$=5) [40] | GP-Gen-SL | 4×Greedy | Two-Opt | 1.597 | 2.094±0.842% | 0.391s | 1.605 | 1.980±0.611% | 2.484s |
| LC4CO [32] | LC-OS-SL | Greedy | – | 1.642 | 4.956±1.514% | 1.563s | 1.655 | 5.201±0.953% | 6.328s |
| LC4CO [32] | LC-OS-SL | Greedy | Two-Opt | 1.634 | 4.413±1.183% | 1.565s | 1.649 | 4.813±0.811% | 6.514s |
| LC4CO [32] | LC-OS-SL | Beam-16 | – | 1.614 | 3.172±1.154% | 4.704s | 1.639 | 4.190±0.629% | 63.388s |
| LC4CO [32] | LC-OS-SL | Beam-16 | Two-Opt | 1.610 | 2.917±0.938% | 4.748s | 1.636 | 3.954±0.760% | 63.571s |
| AE4CO ($D_s$=3, $I_s$=1) [37] | AE-Gen-SL | Greedy | – | 1.628 | 4.078±2.054% | 0.099s | 1.646 | 4.634±1.708% | 0.403s |
| AE4CO ($D_s$=3, $I_s$=1) [37] | AE-Gen-SL | Greedy | Two-Opt | 1.616 | 3.293±1.526% | 0.111s | 1.630 | 3.592±1.390% | 0.478s |
| AE4CO ($D_s$=3, $I_s$=5) [37] | AE-Gen-SL | Greedy | – | 1.615 | 3.200±1.461% | 0.547s | 1.623 | 3.128±1.402% | 1.669s |
| AE4CO ($D_s$=3, $I_s$=5) [37] | AE-Gen-SL | Greedy | Two-Opt | 1.605 | 2.563±1.158% | 0.517s | 1.611 | 2.408±0.888% | 1.766s |
| AE4CO ($D_s$=3, $I_s$=5) [37] | AE-Gen-SL | 4×Greedy | – | 1.593 | 1.790±0.953% | 1.073s | 1.604 | 1.965±0.660% | 6.267s |
| AE4CO ($D_s$=3, $I_s$=5) [37] | AE-Gen-SL | 4×Greedy | Two-Opt | **1.588** | **1.501±0.805%** | 1.097s | 1.598 | **1.568±0.487%** | 6.597s |

Table 14: Complete results on CVRP.

| METHOD | TYPE | SOLVING STAGE | | CVRP-50 | | | CVRP-100 | | |
|---|---|---|---|---|---|---|---|---|---|
| | | INITIALIZATION | OPTIMIZATION | OBJ.↓ | DROP↓ | TIME↓ | OBJ.↓ | DROP↓ | TIME↓ |
| HGS [75] | Heuristics | – | – | 10.366* | 0.000±0.000% | 1.005s | 15.563* | 0.000±0.000% | 20.027s |
| GP4CO ($I_s$=1) [40] | GP-Gen-SL | Greedy | – | 12.640 | 21.835±6.535% | 0.009s | 19.202 | 23.333±5.155% | 0.010s |
| GP4CO ($I_s$=1) [40] | GP-Gen-SL | Greedy | Classic-LS | 10.871 | 4.836±5.552% | 0.013s | 16.294 | 4.698±3.120% | 0.018s |
| LC4CO [41] | LC-OS-RL | Greedy | – | 10.769 | 3.891±1.640% | 0.087s | 16.220 | 4.241±1.131% | 0.166s |
| LC4CO [41] | LC-OS-RL | Greedy | Classic-LS | 10.565 | 1.910±1.435% | 0.168s | 15.933 | 2.379±1.001% | 0.173s |
| LC4CO [41] | LC-OS-RL | N×Greedy | – | 10.642 | 2.658±1.123% | 0.145s | 16.062 | 3.214±0.796% | 0.192s |
| LC4CO [41] | LC-OS-RL | N×Greedy | Classic-LS | 10.498 | 1.267±1.015% | 0.147s | 15.838 | 1.767±0.773% | 0.198s |
| LC4CO [41] | LC-OS-RL | N×Sampling | – | 10.638 | 2.617±1.080% | 0.152s | 16.063 | 3.220±0.767% | 0.195s |
| LC4CO [41] | LC-OS-RL | N×Sampling | Classic-LS | **10.489** | **1.179±0.976%** | 0.154s | **15.822** | **1.663±0.743%** | 0.202s |
| LC4CO [32] | LC-OS-SL | Greedy | – | 10.906 | 5.193±2.962% | 0.504s | 16.342 | 5.005±2.224% | 0.962s |
| LC4CO [32] | LC-OS-SL | Greedy | Classic-LS | 10.628 | 2.519±2.058% | 0.507s | 15.959 | 2.548±1.248% | 0.969s |
| LC4CO [32] | LC-OS-SL | Beam-16 | – | 10.622 | 2.444±1.654% | 0.549s | 15.991 | 2.743±1.310% | 1.278s |
| LC4CO [32] | LC-OS-SL | Beam-16 | Classic-LS | 10.525 | 1.513±1.325% | 0.552s | 15.837 | 1.757±0.945% | 1.286s |
| AE4CO ($D_s$=3, $I_s$=1) [37] | AE-Gen-SL | Greedy | – | 11.979 | 15.407±6.898% | 0.033s | 17.497 | 12.343±5.459% | 0.047s |
| AE4CO ($D_s$=3, $I_s$=1) [37] | AE-Gen-SL | Greedy | Classic-LS | 10.773 | 3.903±4.052% | 0.037s | 16.224 | 4.253±1.991% | 0.055s |

| METHOD | TYPE | SOLVING STAGE | | CVRP-200 | | | CVRP-500 | | |
|---|---|---|---|---|---|---|---|---|---|
| | | INITIALIZATION | OPTIMIZATION | OBJ.↓ | DROP↓ | TIME↓ | OBJ.↓ | DROP↓ | TIME↓ |
| HGS [75] | Heuristics | – | – | 19.630* | 0.000±0.000% | 60.024s | 37.154* | 0.000±0.000% | 360.376s |
| GP4CO ($I_s$=1) [40] | GP-Gen-SL | Greedy | – | 25.064 | 27.616±5.095% | 0.059s | 47.749 | 28.509±2.920% | 0.091s |
| GP4CO ($I_s$=1) [40] | GP-Gen-SL | Greedy | Classic-LS | 20.662 | 5.290±1.471% | 0.063s | 39.195 | 5.530±0.937% | 0.215s |
| LC4CO [41] | LC-OS-RL | Greedy | – | 20.662 | 5.274±0.926% | 0.320s | 40.382 | 8.723±0.810% | 0.769s |
| LC4CO [41] | LC-OS-RL | Greedy | Classic-LS | 20.193 | 2.880±0.854% | 0.341s | 38.700 | 4.173±0.606% | 0.883s |
| LC4CO [41] | LC-OS-RL | N×Greedy | – | 20.507 | 4.474±0.740% | 0.379s | 40.051 | 7.821±0.596% | 0.925s |
| LC4CO [41] | LC-OS-RL | N×Greedy | Classic-LS | 20.097 | 2.383±0.645% | 0.400s | 38.559 | 3.793±0.627% | 1.034s |
| LC4CO [41] | LC-OS-RL | N×Sampling | – | 20.556 | 4.734±0.741% | 0.405s | 40.437 | 8.862±0.680% | 1.024s |
| LC4CO [41] | LC-OS-RL | N×Sampling | Classic-LS | **20.091** | **2.359±0.701%** | 0.439s | 38.560 | 3.794±0.580% | 1.132s |
| LC4CO [32] | LC-OS-SL | Greedy | – | 21.439 | 9.263±2.504% | 1.978s | 38.846 | 4.587±1.159% | 7.095s |
| LC4CO [32] | LC-OS-SL | Greedy | Classic-LS | 20.396 | 3.936±1.492% | 2.037s | 38.118 | 2.620±0.609% | 7.312s |
| LC4CO [32] | LC-OS-SL | Beam-16 | – | 20.665 | 5.272±1.396% | 5.067s | 38.279 | 3.051±0.694% | 64.431s |
| LC4CO [32] | LC-OS-SL | Beam-16 | Classic-LS | 20.195 | 2.882±0.888% | 5.104s | **37.901** | **2.031±0.543%** | 64.589s |
| AE4CO ($D_s$=3, $I_s$=1) [37] | AE-Gen-SL | Greedy | – | 22.402 | 13.977±4.072% | 0.145s | 43.901 | 18.199±2.834% | 0.554s |
| AE4CO ($D_s$=3, $I_s$=1) [37] | AE-Gen-SL | Greedy | Classic-LS | 20.587 | 4.893±1.344% | 0.153s | 39.121 | 5.337±0.938% | 0.605s |

Table 15: Complete results on MIS.

| METHOD | TYPE | SOLVING STAGE | RB-SMALL OBJ↑ | DROP↓ | TIME↓ | RB-LARGE OBJ↑ | DROP↓ | TIME↓ |
|---|---|---|---|---|---|---|---|---|
| KaMIS [76] | Heuristics | – | 20.090 | 0.000±0.000% | 45.809s | 43.004 | 0.000±0.000% | 56.974s |
| Gurobi [5] | Heuristics | – | 20.090 | 0.000±0.000% | 0.538s | 42.192 | 1.829±2.942% | 33.843s |
| GP4CO [39] | GP-OS-SL | Greedy | 18.170 | 9.413±6.644% | 0.014s | 36.060 | 16.036±6.206% | 0.048s |
| GP4CO [20] | GP-OS-MAML(UL) | Greedy | 17.692 | 11.778±5.346% | 0.084s | 34.670 | 19.259±4.084% | 0.127s |
| GP4CO [26] | GP-OS-UL | Greedy | 17.308 | 13.709±7.679% | 0.016s | 34.002 | 20.779±6.753% | 0.029s |
| GP4CO ($I_s$=1) [40] | GP-Gen-SL | Greedy | 18.400 | 8.305±6.381% | 0.014s | 36.394 | 15.219±6.259% | 0.044s |
| GP4CO ($I_s$=20) [40] | GP-Gen-SL | Greedy | 19.330 | 3.710±3.952% | 0.192s | 38.936 | 9.388±4.740% | 0.714s |
| GP4CO ($I_s$=20) [40] | GP-Gen-SL | 4×Greedy | 19.742 | 1.690±2.467% | 0.332s | 40.066 | 6.742±3.971% | 2.398s |
| GP4CO [19] | GP-Gen-UL | Greedy | 19.200 | 4.369±3.965% | 0.470s | 38.490 | 10.428±3.552% | 4.712s |
| GP4CO [19] | GP-Gen-UL | 4×Greedy | 19.380 | 3.464±3.388% | 1.587s | 39.546 | 7.944±3.171% | 25.479s |
| LC4CO [32] | LC-OS-SL | Greedy | 18.268 | 8.959±5.419% | 0.182s | 37.596 | 12.476±4.125% | 3.962s |
| AE4CO ($D_s$=5, $I_s$=20) [37] | AE-Gen-SL | Greedy | 19.604 | 2.375±3.003% | 0.802s | 40.590 | 5.559±3.334% | 3.562s |
| AE4CO ($D_s$=20, $I_s$=1) [37] | AE-Gen-SL | Greedy | 19.662 | 2.088±2.891% | 0.112s | 41.234 | 4.060±2.822% | 0.624s |
| AE4CO ($D_s$=20, $I_s$=1) [37] | AE-Gen-SL | 4×Greedy | 19.706 | 1.880±2.691% | 0.120s | 41.438 | 3.582±2.379% | 1.298s |
| AE4CO ($D_s$=20, $I_s$=1) [37] | AE-Gen-SL | Greedy + RLSA | **20.070** | **0.093±0.655%** | 0.471s | **42.400** | **1.366±1.493%** | 1.816s |

| METHOD | TYPE | SOLVING STAGE | ER-700-800 OBJ↑ | DROP↓ | TIME↓ | SATLIB OBJ↑ | DROP↓ | TIME↓ |
|---|---|---|---|---|---|---|---|---|
| KaMIS [76] | Heuristics | – | 44.969 | 0.000±0.000% | 60.753s | 425.954 | 0.000±0.000% | 24.368s |
| Gurobi [5] | Heuristics | – | 38.781 | 13.749±3.017% | 60.489s | 425.924 | 0.007±0.074% | 3.953s |
| GP4CO [39] | GP-OS-SL | Greedy | 35.852 | 20.267±4.449% | 0.039s | 421.056 | 1.154±0.528% | 0.019s |
| GP4CO [20] | GP-OS-MAML(UL) | Greedy | 33.148 | 26.280±2.551% | 0.117s | 412.122 | 3.250±0.613% | 0.240s |
| GP4CO [26] | GP-OS-UL | Greedy | 33.813 | 24.804±4.235% | 0.029s | – | – | – |
| GP4CO ($I_s$=1) [40] | GP-Gen-SL | Greedy | 35.711 | 20.580±4.169% | 0.039s | 421.578 | 1.031±0.508% | 0.016s |
| GP4CO ($I_s$=20) [40] | GP-Gen-SL | Greedy | 40.195 | 10.618±3.117% | 0.641s | 424.556 | 0.329±0.247% | 0.190s |
| GP4CO ($I_s$=20) [40] | GP-Gen-SL | 4×Greedy | 41.430 | 7.867±2.460% | 2.094s | 425.276 | **0.160±0.161%** | 0.428s |
| LC4CO [32] | LC-OS-SL | Greedy | 40.477 | 9.986±2.784% | 2.296s | 421.882 | 0.957±0.374% | 49.842s |
| AE4CO [43] | AE-OS-RL | Greedy | 37.289 | 17.068±5.254% | 0.589s | – | – | – |
| AE4CO [43] | AE-OS-RL | 4×Greedy | 39.211 | 12.799±2.660% | 0.611s | – | – | – |
| AE4CO ($D_s$=5, $I_s$=20) [37] | AE-Gen-SL | Greedy | 41.992 | 6.610±2.577% | 3.188s | 424.734 | 0.287±0.228% | 1.034s |
| AE4CO ($D_s$=20, $I_s$=1) [37] | AE-Gen-SL | Greedy | 42.383 | 5.746±2.639% | 0.469s | 425.046 | 0.215±0.235% | 0.228s |
| AE4CO ($D_s$=20, $I_s$=1) [37] | AE-Gen-SL | 4×Greedy | 42.563 | 5.343±2.229% | 0.703s | 424.776 | 0.278±0.245% | 0.268s |
| AE4CO ($D_s$=20, $I_s$=1) [37] | AE-Gen-SL | Greedy + RLSA | **44.984** | **-0.041±1.389%** | 1.390s | **425.316** | 0.151±0.173% | 1.775s |

| METHOD | TYPE | SOLVING STAGE | ER-1400-1600 OBJ↑ | DROP↓ | TIME↓ | RB-GIANT OBJ↑ | DROP↓ | TIME↓ |
|---|---|---|---|---|---|---|---|---|
| KaMIS [76] | Heuristics | – | 50.938 | 0.000±0.000% | 60.824s | 49.260 | 0.000±0.000% | 60.960s |
| Gurobi [5] | Heuristics | – | 44.813 | 12.015±2.736% | 3602.519s | 48.560 | 1.377±2.452% | 3426.207s |
| GP4CO [39] | GP-OS-SL | Greedy | 37.680 | 26.021±3.738% | 0.117s | 38.760 | 21.224±5.395% | 0.240s |
| GP4CO [20] | GP-OS-MAML(UL) | Greedy | 37.555 | 26.262±2.578% | 0.267s | 38.840 | 21.039±3.670% | 0.739s |
| GP4CO [26] | GP-OS-UL | Greedy | 36.914 | 27.533±3.431% | 0.069s | 37.260 | 24.195±5.781% | 0.151s |
| GP4CO [19] | GP-Gen-UL | Greedy | – | – | – | 40.480 | 17.722±5.765% | 45.680s |
| GP4CO [19] | GP-Gen-UL | 4×Greedy | – | – | – | 42.120 | 14.352±4.582% | 339.010s |
| GP4CO ($I_s$=1) [40] | GP-Gen-SL | Greedy | 37.859 | 25.670±3.368% | 0.117s | 38.200 | 22.316±6.721% | 0.220s |
| GP4CO ($I_s$=50) [40] | GP-Gen-SL | Greedy | 44.094 | 13.430±3.135% | 5.117s | 38.380 | 21.995±6.499% | 8.980s |
| GP4CO ($I_s$=50) [40] | GP-Gen-SL | 4×Greedy | 46.117 | 9.446±2.654% | 23.781s | 40.260 | 18.120±6.109% | 43.760s |
| LC4CO [32] | LC-OS-SL | Greedy | 45.594 | 10.485±2.822% | 9.966s | 42.640 | 13.304±3.839% | 24.499s |
| AE4CO [43] | AE-OS-RL | Greedy | 39.523 | 22.400±6.333% | 1.468s | – | – | – |
| AE4CO [43] | AE-OS-RL | 4×Greedy | 42.477 | 16.601±3.340% | 1.482s | – | – | – |
| AE4CO ($D_s$=50, $I_s$=1) [37] | AE-Gen-SL | Greedy | 48.164 | 5.432±2.188% | 2.211s | 45.840 | 6.856±3.531% | 5.340s |
| AE4CO ($D_s$=50, $I_s$=1) [37] | AE-Gen-SL | 4×Greedy | 48.429 | 4.911±2.107% | 4.516s | 46.640 | 5.221±2.970% | 14.660s |
| AE4CO ($D_s$=50, $I_s$=1) [37] | AE-Gen-SL | Greedy + RLSA | **50.719** | **0.418±1.489%** | 4.102s | **47.880** | **2.741±2.001%** | 9.490s |

Table 16: Complete results on MCl.

| METHOD | TYPE | SOLVING STAGE | RB-SMALL | | | RB-LARGE | | |
|---|---|---|---|---|---|---|---|---|
| | | | OBJ.↑ | DROP↓ | TIME↓ | OBJ.↑ | DROP↓ | TIME↓ |
| Gurobi [5] | Heuristics | – | 19.082 | 0.000±0.000% | 0.900s | 40.182 | 0.000±0.000% | 276.657s |
| GP4CO [39] | GP-OS-SL | Greedy | 18.084 | 5.782±10.970% | 0.014s | 38.622 | 4.331±10.697% | 0.042s |
| GP4CO [39] | GP-OS-SL | Beam-16 | 18.592 | 2.950±6.595% | 0.023s | 39.540 | 1.857±5.579% | 0.104s |
| GP4CO [39] | GP-OS-SL | Beam-16 + RLSA | **19.082** | **0.000±0.000%** | 0.041s | 40.256 | **-0.275±1.721%** | 0.171s |
| GP4CO [20] | GP-OS-MAML(UL) | Greedy | 17.856 | 6.400±10.924% | 0.065s | 34.802 | 13.180±17.643% | 0.103s |
| GP4CO [26] | GP-OS-UL | Greedy | 15.084 | 20.720±18.362% | 0.016s | 29.052 | 26.604±21.115% | 0.029s |
| GP4CO [26] | GP-OS-UL | Beam-16 | 17.086 | 10.572±12.309% | 0.017s | 35.098 | 12.318±16.956% | 0.091s |
| GP4CO ($I_s$=1) [40] | GP-Gen-SL | Greedy | 18.258 | 4.870±10.775% | 0.016s | 36.402 | 9.809±16.700% | 0.042s |
| GP4CO ($I_s$=1) [40] | GP-Gen-SL | Beam-16 | 18.664 | 2.510±6.024% | 0.022s | 38.754 | 3.798±9.039% | 0.104s |
| GP4CO ($I_s$=20) [40] | GP-Gen-SL | Greedy | 18.608 | 2.850±6.945% | 0.200s | 35.384 | 12.672±19.874% | 0.708s |
| GP4CO ($I_s$=20) [40] | GP-Gen-SL | 4×Greedy | 18.982 | 0.607±2.521% | 0.328s | 38.706 | 4.197±11.586% | 2.386s |
| GP4CO [19] | GP-Gen-UL | Greedy | 15.142 | 18.245±18.185% | 0.559s | – | – | – |
| GP4CO [19] | GP-Gen-UL | 4×Greedy | 16.206 | 12.527±18.723% | 1.412s | – | – | – |
| LC4CO [32] | LC-OS-SL | Greedy | 15.682 | 17.628±17.431% | 0.171s | 29.382 | 26.156±21.529% | 3.148s |
| AE4CO ($D_s$=5, $I_s$=20) [37] | AE-Gen-SL | Greedy | 18.686 | 2.374±5.831% | 0.516s | 36.176 | 10.159±16.850% | 1.660s |
| AE4CO ($D_s$=20, $I_s$=1) [37] | AE-Gen-SL | Greedy | 18.766 | 1.892±4.479% | 0.046s | 36.752 | 8.817±15.326% | 0.150s |
| AE4CO ($D_s$=20, $I_s$=1) [37] | AE-Gen-SL | 4×Greedy | 18.922 | 1.018±3.009% | 0.056s | 39.170 | 2.722±7.875% | 0.414s |

| METHOD | TYPE | SOLVING STAGE | TWITTER | | | COLLAB | | |
|---|---|---|---|---|---|---|---|---|
| | | | OBJ.↑ | DROP↓ | TIME↓ | OBJ.↑ | DROP↓ | TIME↓ |
| Gurobi [5] | Heuristics | – | 14.210 | 0.000±0.000% | 0.276s | 42.113 | 0.000±0.000% | 0.063s |
| GP4CO [39] | GP-OS-SL | Greedy | 12.897 | 12.002±16.094% | 0.013s | 41.474 | 5.330±14.850% | 0.012s |
| GP4CO [39] | GP-OS-SL | Beam-16 | 13.210 | 7.931±9.718% | 0.020s | 41.660 | 3.219±8.523% | 0.012s |
| GP4CO [39] | GP-OS-SL | Beam-16 + RLSA | **14.210** | **0.000±0.000%** | 0.044s | **42.113** | **0.000±0.000%** | 0.024s |
| GP4CO [20] | GP-OS-MAML(UL) | Greedy | 13.815 | 2.714±5.375% | 0.063s | 41.024 | 2.094±10.665% | 0.065s |
| GP4CO [26] | GP-OS-UL | Greedy | 11.985 | 17.077±17.597% | 0.009s | 39.461 | 13.802±22.431% | 0.007s |
| GP4CO [26] | GP-OS-UL | Beam-16 | 12.369 | 12.953±12.775% | 0.015s | 40.732 | 7.354±12.027% | 0.008s |
| GP4CO ($I_s$=1) [40] | GP-Gen-SL | Greedy | 13.051 | 9.536±13.448% | 0.015s | 41.593 | 2.496±9.210% | 0.013s |
| GP4CO ($I_s$=1) [40] | GP-Gen-SL | Beam-16 | 13.303 | 6.504±8.451% | 0.021s | 41.772 | 1.695±5.767% | 0.014s |
| GP4CO ($I_s$=20) [40] | GP-Gen-SL | Greedy | 12.236 | 16.961±24.014% | 0.195s | 39.501 | 14.496±26.836% | 0.222s |
| GP4CO ($I_s$=20) [40] | GP-Gen-SL | 4×Greedy | 13.349 | 8.596±18.106% | 0.297s | 41.051 | 5.974±17.730% | 0.314s |
| LC4CO [32] | LC-OS-SL | Greedy | 13.262 | 7.124±9.726% | 0.141s | 41.775 | 2.189±6.839% | 0.408s |
| AE4CO ($D_s$=20, $I_s$=1) [37] | AE-Gen-SL | Greedy | 13.528 | 6.154±11.888% | 0.067s | 41.874 | 1.808±8.092% | 0.139s |
| AE4CO ($D_s$=20, $I_s$=1) [37] | AE-Gen-SL | 4×Greedy | 13.805 | 3.997±10.243% | 0.082s | 42.040 | 0.841±6.363% | 0.092s |

| METHOD | TYPE | TYPE | RB-GIANT | | |
|---|---|---|---|---|---|
| | | | OBJ.↑ | DROP↓ | TIME↓ |
| Gurobi [5] | Heuristics | – | 81.520 | 0.000±0.000% | 3606.201s |
| GP4CO [39] | GP-OS-SL | Greedy | 58.440 | 26.568±27.047% | 0.220s |
| GP4CO [39] | GP-OS-SL | Beam-16 | 77.460 | 3.119±22.916% | 0.406s |
| GP4CO [39] | GP-OS-SL | Beam-16 + RLSA | **85.380** | **-7.912±25.418%** | 4.342s |
| GP4CO [20] | GP-OS-MAML(UL) | Greedy | 77.160 | 1.613±30.471% | 0.683s |
| GP4CO [26] | GP-OS-UL | Greedy | 55.580 | 29.171±24.586% | 0.148s |
| GP4CO [26] | GP-OS-UL | Beam-16 | 76.360 | 4.433±20.950% | 0.333s |
| GP4CO ($I_s$=1) [40] | GP-Gen-SL | Greedy | 60.860 | 22.902±29.840% | 0.240s |
| GP4CO ($I_s$=1) [40] | GP-Gen-SL | 4×Greedy | 77.860 | 2.145±25.410% | 0.820s |
| GP4CO ($I_s$=1) [40] | GP-Gen-SL | Beam-16 | 77.340 | 2.732±24.924% | 0.420s |
| GP4CO ($I_s$=1) [40] | GP-Gen-SL | 4×Beam-16 | 84.120 | -6.424±26.720% | 1.520s |
| LC4CO [32] | LC-OS-SL | Greedy | 57.120 | 27.719±24.330% | 32.680s |
| AE4CO ($D_s$=50, $I_s$=1) [37] | AE-Gen-SL | Greedy | 64.640 | 19.005±22.747% | 1.062s |
| AE4CO ($D_s$=50, $I_s$=1) [37] | AE-Gen-SL | 4×Greedy | 74.920 | 5.339±26.102% | 3.386s |

#### Table 17: Complete results on MVC.

| METHOD | TYPE | SOLVING STAGE | RB-SMALL | | | RB-LARGE | | |
|---|---|---|---|---|---|---|---|---|
| | | | OBJ.↓ | DROP↓ | TIME↓ | OBJ.↓ | DROP↓ | TIME↓ |
| Gurobi [5] | Heuristics | – | 205.764 | 0.000±0.000% | 3.340s | 968.228 | 0.000±0.000% | 290.227s |
| GP4CO [39] | GP-OS-SL | Greedy | 207.554 | 0.871±0.653% | 0.014s | 976.112 | 0.817±0.292% | 0.068s |
| GP4CO [20] | GP-OS-MAML(UL) | Greedy | 208.126 | 1.149±0.514% | 0.065s | 977.914 | 1.002±0.193% | 0.097s |
| GP4CO [26] | GP-OS-UL | Greedy | 208.982 | 1.569±0.736% | 0.013s | 977.186 | 0.928±0.271% | 0.037s |
| GP4CO ($I_s$=1) [40] | GP-Gen-SL | Greedy | 207.460 | 0.827±0.575% | 0.016s | 974.950 | 0.696±0.276% | 0.044s |
| GP4CO ($I_s$=20) [40] | GP-Gen-SL | Greedy | 207.778 | 0.984±0.577% | 0.186s | 972.608 | 0.453±0.195% | 0.718s |
| GP4CO ($I_s$=20) [40] | GP-Gen-SL | 4×Greedy | 207.060 | 0.633±0.462% | 0.322s | 971.420 | 0.330±0.163% | 2.412s |
| LC4CO [32] | LC-OS-SL | Greedy | 207.260 | 0.728±0.482% | 0.182s | 972.532 | 0.447±0.174% | 3.950s |
| AE4CO ($D_s$=20, $I_s$=1) [37] | AE-Gen-SL | Greedy | 206.576 | 0.398±0.381% | 0.108s | 969.922 | 0.176±0.128% | 0.522s |
| AE4CO ($D_s$=20, $I_s$=1) [37] | AE-Gen-SL | 4×Greedy | 206.360 | 0.290±0.332% | 0.086s | 969.806 | 0.163±0.118% | 0.626s |
| AE4CO ($D_s$=20, $I_s$=1) [37] | AE-Gen-SL | Greedy + RLSA | **205.772** | **0.004±0.042%** | 0.612s | **968.398** | **0.018±0.103%** | 1.592s |

| METHOD | TYPE | SOLVING STAGE | TWITTER | | | COLLAB | | |
|---|---|---|---|---|---|---|---|---|
| | | | OBJ.↓ | DROP↓ | TIME↓ | OBJ.↓ | DROP↓ | TIME↓ |
| Gurobi [5] | Heuristics | – | 85.251 | 0.000±0.000% | 0.133s | 65.086 | 0.000±0.000% | 0.058s |
| GP4CO [39] | GP-OS-SL | Greedy | 86.369 | 1.222±1.419% | 0.009s | 65.178 | 0.141±0.566% | 0.009s |
| GP4CO [20] | GP-OS-MAML(UL) | Greedy | 92.518 | 8.262±3.709% | 0.064s | 66.172 | 2.170±2.834% | 0.061s |
| GP4CO [26] | GP-OS-UL | Greedy | 86.677 | 1.575±1.612% | 0.008s | 65.182 | 0.183±0.696% | 0.006s |
| GP4CO [26] | GP-OS-UL | Greedy + RLSA | **85.251** | **0.000±0.000%** | 0.115s | **65.086** | **0.000±0.000%** | 0.158s |
| GP4CO ($I_s$=20) [40] | GP-Gen-SL | Greedy | 86.231 | 1.071±1.383% | 0.198s | 65.131 | 0.093±0.519% | 0.206s |
| GP4CO ($I_s$=20) [40] | GP-Gen-SL | 4×Greedy | 85.585 | 0.332±0.599% | 0.233s | 65.091 | 0.010±0.145% | 0.267s |
| LC4CO [32] | LC-OS-SL | Greedy | 86.841 | 2.695±7.401% | 0.425s | 65.396 | 0.667±1.550% | 0.088s |
| AE4CO ($D_s$=20, $I_s$=1) [37] | AE-Gen-SL | Greedy | 85.574 | 0.326±0.574% | 0.138s | 65.121 | 0.047±0.303% | 0.056s |
| AE4CO ($D_s$=20, $I_s$=1) [37] | AE-Gen-SL | 4×Greedy | 85.446 | 0.204±0.481% | 0.062s | 65.107 | 0.028±0.227% | 0.038s |

| METHOD | TYPE | SOLVING STAGE | RB-GIANT | | |
|---|---|---|---|---|---|
| | | | OBJ.↓ | DROP↓ | TIME↓ |
| Gurobi [5] | Heuristics | – | 2396.780 | 0.000±0.000% | 1813.786s |
| GP4CO [39] | GP-OS-SL | Greedy | 2407.940 | 0.465±0.105% | 0.365s |
| GP4CO [20] | GP-OS-MAML(UL) | Greedy | 2406.360 | 0.402±0.083% | 0.555s |
| GP4CO [26] | GP-OS-UL | Greedy | 2408.320 | 0.482±0.102% | 0.221s |
| GP4CO ($I_s$=1) [40] | GP-Gen-SL | Greedy | 2407.540 | 0.450±0.110% | 0.240s |
| GP4CO ($I_s$=50) [40] | GP-Gen-SL | Greedy | 2405.060 | 0.346±0.096% | 9.008s |
| GP4CO ($I_s$=50) [40] | GP-Gen-SL | 4×Greedy | 2403.580 | 0.284±0.078% | 37.506s |
| LC4CO [32] | LC-OS-SL | Greedy | 2401.740 | 0.208±0.083% | 24.956s |
| AE4CO ($D_s$=50, $I_s$=1) [37] | AE-Gen-SL | Greedy | 2400.600 | 0.160±0.061% | 4.360s |
| AE4CO ($D_s$=50, $I_s$=1) [37] | AE-Gen-SL | 4×Greedy | 2400.360 | 0.149±0.052% | 8.200s |
| AE4CO ($D_s$=50, $I_s$=1) [37] | AE-Gen-SL | Greedy + RLSA | **2397.360** | **0.026±0.079%** | 8.590s |

#### Table 18: Complete results on MCut.

| METHOD | TYPE | SOLVING STAGE | BA-SMALL | | | BA-LARGE | | |
|---|---|---|---|---|---|---|---|---|
| | | | OBJ.↑ | DROP↓ | TIME↓ | OBJ.↑ | DROP↓ | TIME↓ |
| Gurobi [5] | Heuristics | – | 727.844 | 0.000±0.000% | 60.612s | 2936.886 | 0.000±0.000% | 300.214s |
| GP4CO [39] | GP-OS-SL | Greedy | 705.330 | 3.102±1.370% | 0.014s | 2827.480 | 3.744±0.851% | 0.016s |
| GP4CO [20] | GP-OS-MAML(UL) | Greedy | 672.748 | 7.563±1.760% | 0.063s | 2748.310 | 6.425±0.965% | 0.090s |
| GP4CO [26] | GP-OS-UL | Greedy | 700.972 | 3.706±1.198% | 0.017s | 2884.086 | 1.815±0.716% | 0.019s |
| GP4CO ($I_s$=1) [40] | GP-Gen-SL | Greedy | 702.376 | 3.504±1.841% | 0.014s | 2783.834 | 5.231±1.156% | 0.016s |
| GP4CO ($I_s$=20) [40] | GP-Gen-SL | Greedy | 725.624 | 0.319±0.557% | 0.172s | 2948.112 | -0.369±0.580% | 0.194s |
| GP4CO ($I_s$=20) [40] | GP-Gen-SL | 4×Greedy | 728.316 | -0.049±0.371% | 0.202s | 2960.664 | -0.797±0.494% | 0.362s |
| GP4CO (FT) ($I_s$=1) [40] | GP-Gen-SL | Greedy | 718.592 | 1.281±0.765% | 0.014s | 2941.740 | -0.153±0.583% | 0.016s |
| GP4CO (FT) ($I_s$=20) [40] | GP-Gen-SL | Greedy | 726.538 | 0.195±0.476% | 0.172s | 2980.508 | -1.467±0.513% | 0.196s |
| GP4CO (FT) ($I_s$=20) [40] | GP-Gen-SL | 4×Greedy | 728.272 | -0.043±0.386% | 0.196s | 2987.142 | -1.693±0.474% | 0.364s |
| GP4CO (FT) ($I_s$=20) [40] | GP-Gen-SL | Greedy + RLSA | – | – | – | **2994.118** | **-1.932±0.449%** | 0.999s |
| GP4CO [19] | GP-Gen-UL | Greedy | 726.900 | 0.146±0.483% | 0.197s | 2986.932 | -1.688±0.480% | 0.654s |
| GP4CO [19] | GP-Gen-UL | 4×Greedy | 727.534 | 0.061±0.462% | 0.610s | 2989.458 | -1.773±0.466% | 2.701s |
| AE4CO ($D_s$=20, $I_s$=1) [37] | AE-Gen-SL | Greedy | 727.526 | 0.058±0.419% | 0.182s | 2978.200 | -1.387±0.517% | 0.204s |
| AE4CO ($D_s$=20, $I_s$=1) [37] | AE-Gen-SL | 4×Greedy | 726.798 | 0.159±0.442% | 0.064s | 2961.500 | -0.821±0.544% | 0.114s |
| AE4CO ($D_s$=20, $I_s$=1) [37] | AE-Gen-SL | Greedy + RLSA | **729.706** | **0.240±0.313%** | 0.727s | – | – | – |
| AE4CO (FT) ($D_s$=20, $I_s$=1) [37] | AE-Gen-SL | Greedy | 726.382 | 0.219±0.455% | 0.184s | 2975.712 | -1.304±0.511% | 0.202s |
| AE4CO (FT) ($D_s$=20, $I_s$=1) [37] | AE-Gen-SL | 4×Greedy | 726.232 | 0.238±0.467% | 0.064s | 2980.652 | -1.475±0.477% | 0.114s |

| METHOD | TYPE | SOLVING STAGE | BA-GIANT | | |
|---|---|---|---|---|---|
| | | | OBJ.↑ | DROP↓ | TIME↓ |
| Gurobi [5] | Heuristics | – | 7217.900 | 0.000±0.000% | 3601.342s |
| GP4CO [39] | GP-OS-SL | Greedy | 6979.260 | 3.291±0.613% | 0.060s |
| GP4CO [20] | GP-OS-MAML(UL) | Greedy | 6712.120 | 6.994±1.473% | 0.535s |
| GP4CO [26] | GP-OS-UL | Greedy | 7124.880 | 1.287±0.425% | 0.059s |
| GP4CO ($I_s$=1) [40] | GP-Gen-SL | Greedy | 6860.220 | 4.935±0.841% | 0.060s |
| GP4CO ($I_s$=50) [40] | GP-Gen-SL | Greedy | 7308.260 | -1.258±0.403% | 0.700s |
| GP4CO ($I_s$=50) [40] | GP-Gen-SL | 4×Greedy | 7329.420 | -1.553±0.339% | 1.760s |
| GP4CO (FT) ($I_s$=1) [40] | GP-Gen-SL | Greedy | 7264.660 | -0.656±0.448% | 0.060s |
| GP4CO (FT) ($I_s$=50) [40] | GP-Gen-SL | Greedy | 7372.100 | -2.134±0.330% | 0.720s |
| GP4CO (FT) ($I_s$=50) [40] | GP-Gen-SL | 4×Greedy | 7381.920 | -2.276±0.327% | 1.760s |
| GP4CO (FT) ($I_s$=50) [40] | GP-Gen-SL | Greedy + RLSA | **7389.300** | **-2.383±0.310%** | 2.228s |
| GP4CO [19] | GP-Gen-UL | Greedy | 7384.020 | -2.306±0.326% | 2.480s |
| GP4CO [19] | GP-Gen-UL | 4×Greedy | 7387.760 | -2.358±0.325% | 10.760s |
| AE4CO ($D_s$=20, $I_s$=1) [37] | AE-Gen-SL | Greedy | 7369.980 | -2.111±0.330% | 0.720s |
| AE4CO (FT) ($D_s$=20, $I_s$=1) [37] | AE-Gen-SL | Greedy | 7361.800 | -1.997±0.318% | 0.740s |

Table 19: TSPLIB

| INSTANCE | SCALE | EXACT | GP-OS-SL | | GP-GEN-SL | | AE-GEN-SL | | BEST | |
|---|---|---|---|---|---|---|---|---|---|---|
| | | | OBJ.↓ | DROP↓ | OBJ.↓ | DROP↓ | OBJ.↓ | DROP↓ | OBJ.↓ | DROP↓ |
| eil51 | TSP-100 | 6.701 | 6.703 | 0.026% | 6.711 | 0.143% | 6.701 | 0.000% | 6.701 | 0.000% |
| berlin52 | TSP-100 | 4.348 | 4.348 | 0.000% | 4.349 | 0.004% | 4.349 | 0.004% | 4.349 | 0.004% |
| st70 | TSP-100 | 6.839 | 6.840 | 0.000% | 6.840 | 0.000% | 6.840 | 0.000% | 6.840 | 0.000% |
| eil76 | TSP-100 | 7.561 | 7.668 | 1.414% | 7.561 | 0.000% | 7.561 | 0.000% | 7.561 | 0.000% |
| pr76 | TSP-100 | 5.518 | 5.606 | 1.582% | 5.518 | 0.000% | 5.518 | 0.000% | 5.518 | 0.000% |
| rat99 | TSP-100 | 5.671 | 5.671 | 0.000% | 5.675 | 0.079% | 5.675 | 0.079% | 5.671 | 0.000% |
| kroA100 | TSP-100 | 5.408 | 5.408 | 0.000% | 5.408 | 0.000% | 5.408 | 0.000% | 5.408 | 0.000% |
| kroB100 | TSP-100 | 5.626 | 5.668 | 0.742% | 5.668 | 0.742% | 5.668 | 0.742% | 5.668 | 0.742% |
| kroC100 | TSP-100 | 5.276 | 5.276 | 0.000% | 5.452 | 3.330% | 5.452 | 3.330% | 5.276 | 0.000% |
| kroD100 | TSP-100 | 5.431 | 5.431 | 0.000% | 5.431 | 0.000% | 5.431 | 0.000% | 5.431 | 0.000% |
| kroE100 | TSP-100 | 5.557 | 5.641 | 1.495% | 5.567 | 0.178% | 5.570 | 0.232% | 5.567 | 0.178% |
| rd100 | TSP-100 | 8.065 | 8.065 | 0.000% | 8.065 | 0.000% | 8.065 | 0.000% | 8.065 | 0.000% |
| eil101 | TSP-100 | 8.536 | 8.536 | 0.000% | 8.536 | 0.000% | 8.536 | 0.000% | 8.536 | 0.000% |
| lin105 | TSP-100 | 4.756 | 4.756 | 0.000% | 4.793 | 0.779% | 4.756 | 0.000% | 4.756 | 0.000% |
| pr107 | TSP-100 | 3.778 | 3.778 | 0.000% | 3.865 | 2.292% | 3.816 | 1.005% | 3.778 | 0.000% |
| pr124 | TSP-100 | 5.009 | 5.013 | 0.075% | 5.009 | 0.000% | 5.009 | 0.000% | 5.009 | 0.000% |
| bier127 | TSP-100 | 6.106 | 6.165 | 0.953% | 6.176 | 1.140% | 6.215 | 1.771% | 6.106 | 0.953% |
| ch130 | TSP-100 | 8.751 | 8.771 | 0.235% | 8.751 | 0.000% | 8.751 | 0.000% | 8.751 | 0.000% |
| pr136 | TSP-100 | 7.767 | 7.875 | 1.391% | 7.767 | 0.010% | 7.767 | 0.010% | 7.767 | 0.010% |
| pr144 | TSP-100 | 4.538 | 4.591 | 1.175% | 4.557 | 0.417% | 4.555 | 0.386% | 4.555 | 0.386% |
| ch150 | TSP-100 | 9.342 | 9.429 | 0.939% | 9.342 | 0.000% | 9.342 | 0.000% | 9.342 | 0.000% |
| kroA150 | TSP-100 | 6.693 | 6.723 | 0.444% | 6.775 | 1.230% | 6.733 | 0.595% | 6.723 | 0.444% |
| kroB150 | TSP-100 | 6.635 | 6.667 | 0.482% | 6.647 | 0.188% | 6.647 | 0.189% | 6.647 | 0.189% |
| pr152 | TSP-500 | 5.234 | 5.391 | 2.997% | 5.291 | 1.097% | 5.244 | 0.191% | 5.244 | 0.191% |
| u159 | TSP-500 | 6.574 | 6.574 | 0.000% | 6.574 | 0.000% | 6.574 | 0.000% | 6.574 | 0.000% |
| rat195 | TSP-500 | 7.993 | 8.049 | 0.705% | 8.026 | 0.419% | 8.049 | 0.706% | 8.026 | 0.419% |
| d198 | TSP-500 | 3.924 | 3.998 | 1.884% | 3.950 | 0.649% | 3.944 | 0.488% | 3.944 | 0.488% |
| kroA200 | TSP-500 | 7.437 | 7.475 | 0.513% | 7.491 | 0.723% | 7.441 | 0.056% | 7.441 | 0.056% |
| kroB200 | TSP-500 | 7.467 | 7.481 | 0.197% | 7.573 | 1.421% | 7.569 | 1.373% | 7.569 | 1.373% |
| ts225 | TSP-500 | 10.554 | 10.740 | 1.759% | 10.754 | 1.893% | 10.675 | 1.150% | 10.675 | 1.150% |
| tsp225 | TSP-500 | 7.895 | 7.989 | 1.191% | 7.911 | 0.200% | 7.960 | 0.822% | 7.911 | 0.200% |
| pr226 | TSP-500 | 5.279 | 5.288 | 0.169% | 5.308 | 0.551% | 5.315 | 0.683% | 5.308 | 0.551% |
| gil262 | TSP-500 | 12.050 | 12.166 | 0.963% | 12.075 | 0.215% | 12.070 | 0.166% | 12.070 | 0.166% |
| pr264 | TSP-500 | 6.200 | 6.245 | 0.731% | 6.200 | 0.000% | 6.200 | 0.000% | 6.200 | 0.000% |
| a280 | TSP-500 | 9.238 | 9.412 | 1.883% | 9.247 | 0.096% | 9.238 | 0.000% | 9.238 | 0.000% |
| pr299 | TSP-500 | 6.638 | 6.743 | 1.569% | 6.729 | 1.367% | 6.725 | 1.306% | 6.725 | 1.306% |
| lin318 | TSP-500 | 10.170 | 10.305 | 1.327% | 10.209 | 0.386% | 10.244 | 0.727% | 10.209 | 0.386% |
| rd400 | TSP-500 | 15.344 | 15.492 | 0.964% | 15.407 | 0.412% | 15.359 | 0.100% | 15.359 | 0.100% |
| fl417 | TSP-500 | 6.286 | 6.355 | 1.112% | 6.413 | 2.033% | 6.354 | 1.095% | 6.354 | 1.095% |
| pr439 | TSP-500 | 8.991 | 9.255 | 2.936% | 9.065 | 0.822% | 9.123 | 1.475% | 9.065 | 0.822% |
| pcb442 | TSP-500 | 13.364 | 13.543 | 1.337% | 13.450 | 0.645% | 13.441 | 0.576% | 13.441 | 0.576% |
| d493 | TSP-500 | 9.350 | 9.509 | 1.696% | 9.510 | 1.706% | 9.469 | 1.265% | 9.469 | 1.265% |
| u574 | TSP-500 | 12.023 | 12.097 | 0.613% | 12.149 | 1.706% | 12.092 | 0.577% | 12.092 | 0.577% |
| rat575 | TSP-500 | 13.619 | 13.694 | 0.551% | 13.791 | 1.260% | 13.671 | 0.382% | 13.671 | 0.382% |
| p654 | TSP-1000 | 7.196 | 7.516 | 4.454% | 7.240 | 0.624% | 7.240 | 0.621% | 7.240 | 0.621% |
| d657 | TSP-1000 | 12.212 | 12.323 | 0.911% | 12.334 | 1.003% | 12.288 | 0.620% | 12.288 | 0.620% |
| u724 | TSP-1000 | 14.437 | 14.598 | 1.119% | 14.593 | 1.087% | 14.582 | 1.009% | 14.582 | 1.009% |
| rat783 | TSP-1000 | 15.247 | 15.373 | 0.828% | 15.342 | 0.625% | 15.418 | 1.127% | 15.342 | 0.625% |
| pr1002 | TSP-1000 | 16.397 | 16.645 | 1.517% | 16.561 | 1.001% | 16.490 | 0.571% | 16.490 | 0.571% |
| *mean* | – | **8.062** | **8.140** | **0.916%** | **8.115** | **0.649%** | **8.105** | **0.519%** | **8.095** | **0.356%** |

Table 20: Generalization experiment of the *LC-OS-RL* solver on CVRPLIB. CVRP-X denotes the model trained on instances of size X. Settings: N×Sampling + Classic-LS.

| INSTANCE | EXACT | CVRP-100 | | CVRP-200 | | CVRP-500 | | AVERAGE | | BEST | |
|---|---|---|---|---|---|---|---|---|---|---|---|
| | | OBJ.↓ | DROP↓ | OBJ.↓ | DROP↓ | OBJ.↓ | DROP↓ | OBJ.↓ | DROP↓ | OBJ.↓ | DROP↓ |
| X-n101-k25 | 27.905 | 30.625 | 9.744% | 31.702 | 13.605% | 30.563 | 9.524% | 30.963 | 10.958% | 30.563 | 9.524% |
| X-n106-k14 | 26.362 | 26.969 | 2.300% | 26.802 | 1.668% | 26.955 | 2.250% | 26.909 | 2.073% | 26.802 | 1.668% |
| X-n110-k13 | 14.979 | 15.097 | 0.784% | 15.219 | 1.599% | 15.523 | 3.629% | 15.280 | 2.004% | 15.097 | 0.784% |
| X-n115-k10 | 12.933 | 13.896 | 7.448% | 14.112 | 9.115% | 13.574 | 4.957% | 13.860 | 7.173% | 13.574 | 4.957% |
| X-n120-k6 | 13.437 | 13.611 | 1.296% | 13.672 | 1.749% | 13.908 | 3.509% | 13.730 | 2.185% | 13.611 | 1.296% |
| X-n125-k30 | 56.164 | 59.041 | 5.122% | 62.810 | 11.833% | 62.514 | 11.307% | 61.455 | 9.420% | 59.041 | 5.122% |
| X-n129-k18 | 29.005 | 29.476 | 1.625% | 29.356 | 1.211% | 29.777 | 2.662% | 29.536 | 1.833% | 29.356 | 1.211% |
| X-n134-k13 | 11.351 | 11.633 | 2.479% | 11.810 | 4.045% | 11.674 | 2.841% | 11.706 | 3.122% | 11.633 | 2.479% |
| X-n139-k10 | 13.678 | 14.048 | 2.708% | 13.804 | 0.920% | 14.074 | 2.894% | 13.975 | 2.174% | 13.804 | 0.920% |
| X-n143-k7 | 15.776 | 16.533 | 4.798% | 16.602 | 5.239% | 16.702 | 5.868% | 16.612 | 5.301% | 16.533 | 4.798% |
| X-n148-k46 | 43.671 | 53.319 | 22.092% | 45.190 | 3.478% | 55.014 | 25.973% | 51.175 | 17.181% | 45.190 | 3.478% |
| X-n153-k22 | 21.572 | 31.742 | 47.146% | 36.105 | 67.371% | 28.809 | 33.547% | 32.219 | 49.354% | 28.809 | 33.547% |
| X-n157-k13 | 17.572 | 17.708 | 0.775% | 17.870 | 1.696% | 17.811 | 1.365% | 17.796 | 1.279% | 17.708 | 0.775% |
| X-n162-k11 | 14.412 | 14.705 | 2.029% | 14.707 | 2.044% | 14.732 | 2.220% | 14.715 | 2.098% | 14.705 | 2.029% |
| X-n167-k10 | 20.766 | 21.244 | 2.303% | 21.287 | 2.510% | 21.684 | 4.421% | 21.405 | 3.078% | 21.244 | 2.303% |
| X-n172-k51 | 46.974 | 65.767 | 40.007% | 79.562 | 69.372% | 59.677 | 27.041% | 68.335 | 45.473% | 59.677 | 27.041% |
| X-n176-k26 | 47.816 | 51.999 | 8.748% | 56.172 | 17.475% | 52.487 | 9.770% | 53.553 | 11.998% | 51.999 | 8.748% |
| X-n181-k23 | 26.809 | 27.067 | 0.963% | 27.092 | 1.057% | 27.074 | 0.990% | 27.078 | 1.003% | 27.067 | 0.963% |
| X-n186-k15 | 24.251 | 24.720 | 1.933% | 24.721 | 1.936% | 25.753 | 6.193% | 25.065 | 3.354% | 24.720 | 1.933% |
| X-n190-k8 | 17.037 | 17.817 | 4.580% | 17.418 | 2.238% | 17.687 | 3.817% | 17.641 | 3.545% | 17.418 | 2.238% |
| X-n195-k51 | 44.274 | 45.772 | 3.385% | 60.037 | 35.604% | 50.881 | 14.924% | 52.230 | 17.971% | 45.772 | 3.385% |
| X-n200-k36 | 60.084 | 62.080 | 3.324% | 61.860 | 2.957% | 61.834 | 2.914% | 61.925 | 3.065% | 61.834 | 2.914% |
| X-n204-k19 | 19.588 | 20.166 | 2.951% | 20.034 | 2.279% | 19.967 | 1.936% | 20.056 | 2.389% | 19.967 | 1.936% |
| X-n209-k16 | 30.691 | 31.481 | 2.575% | 31.107 | 1.355% | 31.561 | 2.835% | 31.383 | 2.255% | 31.107 | 1.355% |
| X-n214-k11 | 10.879 | 11.543 | 6.100% | 11.278 | 3.665% | 11.413 | 4.903% | 11.411 | 4.889% | 11.278 | 3.665% |
| X-n219-k73 | 117.601 | 124.696 | 6.033% | 126.506 | 7.572% | 129.738 | 10.320% | 126.980 | 7.975% | 124.696 | 6.033% |
| X-n223-k34 | 40.478 | 41.308 | 2.051% | 42.331 | 4.576% | 41.637 | 2.862% | 41.759 | 3.163% | 41.308 | 2.051% |
| X-n228-k23 | 25.747 | 28.368 | 10.180% | 27.065 | 5.119% | 26.867 | 4.350% | 27.433 | 6.550% | 26.867 | 4.350% |
| X-n233-k16 | 19.297 | 19.822 | 2.721% | 19.731 | 2.249% | 20.114 | 4.235% | 19.889 | 3.068% | 19.731 | 2.249% |
| X-n237-k14 | 27.078 | 27.573 | 1.828% | 27.397 | 1.178% | 28.176 | 4.057% | 27.715 | 2.354% | 27.397 | 1.178% |
| X-n242-k48 | 82.757 | 84.827 | 2.502% | 84.427 | 2.018% | 84.730 | 2.384% | 84.661 | 2.301% | 84.427 | 2.018% |
| X-n247-k50 | 37.966 | 59.721 | 57.301% | 63.388 | 66.958% | 55.808 | 46.994% | 59.639 | 57.084% | 55.808 | 46.994% |
| X-n251-k28 | 38.885 | 39.904 | 2.621% | 39.798 | 2.348% | 40.056 | 3.011% | 39.919 | 2.660% | 39.798 | 2.348% |
| X-n256-k16 | 18.942 | 19.784 | 4.444% | 19.379 | 2.306% | 19.432 | 2.582% | 19.532 | 3.111% | 19.379 | 2.306% |
| X-n261-k13 | 26.591 | 29.077 | 9.348% | 27.586 | 3.741% | 27.591 | 3.759% | 28.085 | 5.616% | 27.586 | 3.741% |
| X-n266-k58 | 75.489 | 81.892 | 8.482% | 78.439 | 3.908% | 78.007 | 3.336% | 79.446 | 5.242% | 78.007 | 3.336% |
| X-n270-k35 | 35.432 | 37.837 | 6.786% | 36.133 | 1.978% | 37.645 | 6.244% | 37.205 | 5.003% | 36.133 | 1.978% |
| X-n275-k28 | 22.924 | 24.085 | 5.068% | 23.488 | 2.464% | 23.829 | 3.948% | 23.801 | 3.827% | 23.488 | 2.464% |
| X-n280-k17 | 33.503 | 34.597 | 3.265% | 35.739 | 6.673% | 36.544 | 9.074% | 35.627 | 6.337% | 34.597 | 3.265% |
| X-n284-k15 | 20.421 | 21.564 | 5.598% | 21.252 | 4.071% | 21.472 | 5.148% | 21.429 | 4.939% | 21.252 | 4.071% |
| X-n289-k60 | 95.351 | 103.478 | 8.523% | 107.669 | 12.919% | 105.407 | 10.546% | 105.518 | 10.663% | 103.478 | 8.523% |
| X-n294-k50 | 47.315 | 51.281 | 8.383% | 48.783 | 3.102% | 49.012 | 3.588% | 49.692 | 5.025% | 48.783 | 3.102% |
| X-n298-k31 | 34.247 | 35.659 | 4.122% | 35.464 | 3.553% | 35.698 | 4.236% | 35.607 | 3.971% | 35.464 | 3.553% |
| X-n303-k21 | 21.961 | 22.907 | 4.307% | 22.659 | 3.179% | 22.889 | 4.225% | 22.818 | 3.903% | 22.659 | 3.179% |
| X-n308-k13 | 25.980 | 27.164 | 4.556% | 26.723 | 2.859% | 26.891 | 3.505% | 26.926 | 3.640% | 26.723 | 2.859% |
| X-n313-k71 | 94.437 | 103.575 | 9.676% | 110.157 | 16.646% | 107.451 | 13.780% | 107.061 | 13.367% | 103.575 | 9.676% |
| X-n317-k53 | 78.456 | 86.456 | 10.198% | 87.043 | 10.946% | 93.404 | 19.053% | 88.968 | 13.399% | 86.456 | 10.198% |
| X-n322-k28 | 29.846 | 30.892 | 3.506% | 30.590 | 2.492% | 31.861 | 6.753% | 31.114 | 4.250% | 30.590 | 2.492% |
| X-n327-k20 | 27.681 | 28.768 | 3.928% | 28.546 | 3.127% | 28.733 | 3.799% | 28.682 | 3.618% | 28.546 | 3.127% |
| X-n331-k15 | 31.155 | 31.993 | 2.690% | 32.028 | 2.804% | 32.349 | 3.834% | 32.123 | 3.109% | 31.993 | 2.690% |
| X-n336-k84 | 139.283 | 159.225 | 14.318% | 169.684 | 21.827% | 157.244 | 12.896% | 162.051 | 16.347% | 157.244 | 12.896% |
| X-n344-k43 | 42.194 | 43.796 | 3.795% | 43.225 | 2.442% | 43.542 | 3.194% | 43.521 | 3.144% | 43.225 | 2.442% |
| X-n351-k40 | 25.948 | 27.200 | 4.822% | 32.179 | 24.011% | 27.259 | 5.051% | 28.879 | 11.295% | 27.200 | 4.822% |
| X-n359-k29 | 51.512 | 53.260 | 3.394% | 52.817 | 2.535% | 53.347 | 3.562% | 53.141 | 3.164% | 52.817 | 2.535% |
| X-n367-k17 | 24.741 | 25.836 | 4.426% | 25.777 | 4.187% | 25.711 | 3.920% | 25.775 | 4.178% | 25.711 | 3.920% |
| X-n376-k94 | 147.881 | 148.786 | 0.612% | 149.060 | 0.797% | 148.848 | 0.654% | 148.898 | 0.688% | 148.786 | 0.612% |
| X-n384-k52 | 66.086 | 67.866 | 2.693% | 67.689 | 2.426% | 68.387 | 3.482% | 67.981 | 2.867% | 67.689 | 2.426% |
| X-n393-k38 | 38.386 | 40.077 | 4.404% | 39.938 | 4.043% | 39.702 | 3.427% | 39.906 | 3.958% | 39.702 | 3.427% |
| X-n401-k29 | 66.204 | 67.696 | 2.253% | 67.783 | 2.385% | 68.101 | 2.865% | 67.860 | 2.501% | 67.696 | 2.253% |
| X-n411-k19 | 19.741 | 20.966 | 6.210% | 21.030 | 6.534% | 21.476 | 8.791% | 21.158 | 7.178% | 20.966 | 6.210% |
| X-n420-k130 | 108.137 | 134.019 | 23.935% | 145.475 | 34.528% | 152.736 | 41.243% | 144.077 | 33.235% | 134.019 | 23.935% |
| X-n429-k61 | 65.777 | 72.105 | 9.620% | 67.992 | 3.368% | 70.841 | 7.698% | 70.313 | 6.895% | 67.992 | 3.368% |
| X-n439-k37 | 36.484 | 41.280 | 13.146% | 39.235 | 7.541% | 38.941 | 6.735% | 39.819 | 9.140% | 38.941 | 6.735% |
| X-n449-k29 | 55.282 | 58.464 | 5.756% | 56.635 | 2.447% | 59.498 | 7.627% | 58.199 | 5.277% | 56.635 | 2.447% |
| X-n459-k26 | 25.279 | 27.058 | 7.035% | 26.603 | 5.236% | 26.814 | 6.070% | 26.825 | 6.114% | 26.603 | 5.236% |
| X-n469-k138 | 222.364 | 254.665 | 14.526% | 258.329 | 16.174% | 272.896 | 22.725% | 261.963 | 17.808% | 254.665 | 14.526% |
| X-n480-k70 | 89.842 | 98.216 | 9.321% | 92.585 | 3.053% | 92.732 | 3.217% | 94.511 | 5.197% | 92.585 | 3.053% |
| X-n491-k59 | 66.591 | 69.121 | 3.799% | 68.596 | 3.010% | 71.887 | 7.952% | 69.868 | 4.921% | 68.596 | 3.010% |
| X-n502-k39 | 69.262 | 80.609 | 16.383% | 86.878 | 25.433% | 74.836 | 8.049% | 80.774 | 16.622% | 74.836 | 8.049% |
| X-n513-k21 | 24.273 | 25.612 | 5.513% | 25.416 | 4.705% | 25.259 | 4.061% | 25.429 | 4.760% | 25.259 | 4.061% |
| *mean* | **45.183** | **49.159** | **7.590%** | **49.994** | **8.878%** | **49.672** | **7.730%** | **49.608** | **8.066%** | **48.263** | **5.469%** |

