# OpenReview forum: "ML4CO-Bench-101: Benchmark Machine Learning for Classic Combinatorial Problems on Graphs"
_NeurIPS.cc/2025/Datasets_and_Benchmarks_Track — NeurIPS 2025 Datasets and Benchmarks Track poster_

### Official Review · Reviewer_zcew · 2025-06-25

**Rating:** 5
**Confidence:** 3

**Summary:**

The paper rightfully justify the need for a more standardized benchmark in ML4CO and propose to
categorize the different neural solvers into a tri-leveled *paradigm-model-learning taxonomy*. It is
indeed important to properly evaluate a neural solver by taking into account the effect of the
post-processing or solution decoding and to analyze the quality of a solution w.r.t. its need for an
intensive decoding step or the number of iterations required to produce the solution.

They propose a dataset made of 7 COPs. TSP, ATSP and CVRP that are edge-oriented problems and MIS,
MCut, MVC and MCl that are node-oriented problems. They release for each COP a set of synthetic
instances as well as real-world ones. For every instance, an approximate solution is also given,
using known solvers.

**Additional Feedback:**

Typos:

- 12: edge-oriented -> node-oriented
- 13 MCut written twice, replace one MCut with MCl
- 152: "and" written twice, remove the first occurance
- Algorithm 1 & 2: Grpah -> Graph

Some abbreviations are missing a proper introduction:

- RLSA (introduced line 182)
- BA graphs (introduced line 53)
- RB graphs (introduced line 54)

The sentence line 78 to 80 looks like it should be rephrased.

SATLIB is missing a link, similar to TSPLIB and CVRPLIB.

Some citations are badly formatted (ex: 8, 71) or are badly sourced (ex: 33).

**Dataset Code Accessibility:**

Partly

**Dataset Code Comments:**

The dataset is available on HuggingFace, but no documentation is available. I would have liked a
README file with a small tour of the files and a snippet of code to get started.

**Ethical Considerations:**

No, there are no or only very minor ethics concerns

**Final Justification:**

The authors have adequately addressed the concerns raised in the review.

**Limitations Weaknesses:**

This taxonomy is laid out on three different aspects:

- **Paradigm**: number of iterations over the model required to produce a solution
- **Model**: either a one-shot solver (single pass) or a generative one (denoising the solution)
- **Learning**: learning approach to train the solver, either supervised, unsupervised, reinforcement or meta-learning

While I think a taxonomy is indeed important, I don't think the proposed taxonomy is a good one. The
distinction between *paradigm* and *model* is unclear. I would have merged both categories into a
*compute* category, that specifies in general the computed required by the neural solver to produce
its solution, independently of the actual architecture used underneath. On the *learning* side, I
wonder why the meta-learning is specifically mentioned here. It seems to me that meta-learning can
generally use any of the SL, UL or RL approach. See
[DIMES](https://papers.nips.cc/paper_files/paper/2022/hash/a3a7387e49f4de290c23beea2dfcdc75-Abstract-Conference.html)
(RL-based MAML) and [Meta-EGN](https://openreview.net/forum?id=-ENYHCE8zBp) (UL-based MAML) for
example.

They differentiate the decoding strategy with the post-processing technique. From what I understand
the decoding strategy is the method used to generate the solution out of the neural solver
prediction and the post-processing technique is used to refine the solution once it has been
generated. But then why don't MCTS or any sampling optimization method fall into the decoding
strategy? It looks like the line separating both concepts is blurry.

**Strengths Contributions:**

The field of Neural Combinatorial Optimization typically uses randomly generated instances for both
training **and testing**. Releasing a dataset is an important step towards a more accurate
comparison between papers. They also kept the already used real instances which is good.

Another important point is to present a *taxonomy* of neural solvers, allowing us properly compare
methods within its own category and to compare general categories with each other. For example,
if a neural solver is generating a solution step-by-step, it shouldn't be directly compared
to a solver generating the solution all at once.

They also call out that the *decoding strategies* used by different papers make the comparisons even
more difficult. I agree with the authors with the importance of using a simple greedy decoding to
remove the impact of any specific decoding method.

Finally, they train and evaluate multiple neural solvers to give a first sight about the
performances of specific choices over the taxonomy and the decoding strategies.

---

> ### Author Rebuttal · Authors · 2025-07-31
>
> Dear Reviewer zcew,
>
> Thanks for your review and recognition! Below we respond to the five major points you have mentioned.
>
> > **Q1：I would have merged both categories into a compute category, that specifies in general the computed required by the neural solver to produce its solution, independently of the actual architecture used underneath.**
>
> > **A1:** Thank you very much for your suggestion. This may be another reasonable way of classification, but please allow me to share with you the initial reasons why we categorize it into ``paradigm-model-learning``.
> >* ``paradigm``: ***On the paradigm level, the focus is on how to invoke the model to determine the values of decision variables***. This can involve: **1)** invoking the model once to determine all decision variables (corresponding to GP), **2)** invoking the model to determine one decision variable at a time (corresponding to LC), **3)** or invoking the model multiple times to determine subsets of decision variables each time (corresponding to AE).
> >* ``model``: ***On the model level, the focus is on how to call the neural network to obtain prediction results.*** This can involve **1)** a single inference to get the prediction result (corresponding to OS), **2)** or leveraging the principles of generative methods, with multiple iterations of adding and reducing noise to obtain the prediction result (corresponding to Gen).
> >* ``learning``: ***On the learning level, the foucs is on how the model is trained to learn predictions.*** This can involve SL, UL, MAML, RL, etc.
>
> > The reason we distinguish between paradigm and model is partly to ***differentiate the number of times a model is called $D_s$ and the number of inference steps $I_s$ in generative models.***
>
> ---
>
> > **Q2：On the learning side, I wonder why the meta-learning is specifically mentioned here. It seems to me that meta-learning can generally use any of the SL, UL or RL approach. See DIMES (RL-based MAML) and Meta-EGN (UL-based MAML) for example.**
>
> > Thank you very much for raising this issue. After careful consideration, we have decided to make certain modifications. We continue to keep MAML as a separate category because we believe that categorizing DIMES and Meta-EGN under RL and UL respectively could obscure the characteristics of these two methods as their core features remain those of MAML. However, we have added annotations specifying the exact learning forms they employ, ***categorizing DIMES as MAML(RL) and Meta-EGN as MAML(UL)***. In summary, we remain open to taxonomies favored from different perspectives, and as a benchmark, we aim to provide as comprehensive information as possible, leaving room for researchers to explore and interpret.
>
> ---
>
> > **Q3：Why don't MCTS or any sampling optimization method fall into the decoding strategy? It looks like the line separating both concepts is blurry.**
>
> > Firstly, your understanding of ``decoding strategy`` and ``post-processing`` is correct. In our view, ``decoding strategy`` refers to how to make determinations (assign values to decision variables) using the prediction results from neural networks for ***uncomplete solution***; whereas ``post-processing`` refers to optimization starting from a ***complete initial solution***. You can see from Appendix B that both MCTS and RLSA start from an initial solution, with the former optimizing by continuously swapping edges and the latter optimizing by continuously flipping the values of nodes. Therefore, they clearly fall under the category of ``post-processing`` rather than ``decoding strategy``.
>
> ---
>
> > **Q4：The dataset is available on HuggingFace, but no documentation is available. I would have liked a README file with a small tour of the files and a snippet of code to get started.**
>
> > **A4:** When submitting, we have provided the source code hosted on GitHub as well as the dataset hosted on Hugging Face. We have also included corresponding README files. We place more detailed introductions and instructions in the code repository, while the data repository only contains a brief introduction to the dataset and a link pointing to the code repository. Therefore, we hope you can visit our code repository, where there are very detailed steps.
>
> ---
>
> > **Q5: Paper presentation issues**
>
> > **A5:** Thank you for pointing out these minor issues. We have carefully addressed all the typos (correcting "edge-oriented" to "node-oriented", replacing the duplicate "MCut" with "MCl", removing the redundant "and", and fixing "Grpah" to "Graph" in Algorithms 1 & 2). We have also added proper introductions for abbreviations like RLSA, BA graphs, and RB graphs at their first occurrences. The sentence in lines 78–80 has been rephrased for clarity, and a link for SATLIB has been included to align with the formatting of TSPLIB and CVRPLIB. Additionally, we have corrected the formatting of problematic citations (e.g., 8, 71) and verified the sourcing of citation 33 to ensure accuracy. These revisions have been incorporated into the updated manuscript.
>
> ---
> Thank you again for your valuable support. We hope our explanations have been sufficient to address your concerns, and always stay willing to provide further clarifications if any question remains!
>
> Best regards,
>
> The Authors

---

> > ### Comment · Reviewer_zcew · 2025-08-06
> >
> > Thank you for your detailed rebuttal. I’m satisfied with it.

---

### Official Review · Reviewer_pcUr · 2025-07-02

**Rating:** 4
**Confidence:** 4

**Summary:**

The paper proposes a standardized benchmarking framework that enables a fair comparison of past research literature on solving combinatorial optimization problems. The proposed framework involves three solvers, namely GP4CO, LC4CO and AE4CO, corresponding to the global prediction, local construction and adaptive expansion solving paradigms, and standardized datasets for 7 combinatorial optimization problems. The proposed framework isolates factors that could affect the performance of the model. For instance, the removal of heuristics pre and post inference ensures that the performance can be attributed to the model rather than the heuristic.

**Additional Feedback:**

If the authors can alleviate the concerns about the main contribution of the paper not being explained clearly, I can consider increasing my score.

**Dataset Code Accessibility:**

Partly

**Dataset Code Comments:**

The code and dataset can be easily found. Instructions to run the code are clear. However, the Algorithm Design Correspondence Table for DIMES (Qiu et al., 2022) and GOAL (Drakulic et al., 2025) do not tally with Table 1. This difference requires a justification.
Reproducibility may be an issue because the method of dividing into training and validation sets is not clearly mentioned. The authors have mentioned ‘freely divide’, which lacks specificity. If the training and validation sets are not consistent with the training and validation sets used in the paper, then the results may be hard to reproduce, even though the testing set is consistent.

**Ethical Comments:**

For the minor concern, please refer to point 2 of ‘Paper Presentation Issues’.

**Ethical Considerations:**

No, there are no or only very minor ethics concerns

**Final Justification:**

The paper is well-motivated. Novelty of approach is lacking, but this is far compensated by the anticipated impact this will bring to future generations of researchers when benchmarking the work. Overall, my recommendation is 4.

**Limitations Weaknesses:**

1. The paper has clear motivation, but novelty is lacking and new insights gained are brief. Paper shows effort to standardize evaluation protocols for fair assessment. Major weakness is lack of clarity in explaining the main contribution of the paper, especially on the proposed GP4CO, LC4CO and AE4CO solvers.
2.	Relative lack of analysis of results, confined to lines 257 to 281.

Paper presentation issues:
1.	Use of many acronyms makes paper far less readable for those in adjacent fields. It would be better to define the acronyms first, then use the acronym throughout. It is hard to find the definitions of the acronyms used in the paper. It would be, for instance, better if CO were to be notated as combinatorial optimization (CO) in the first mention. Then, you can use CO subsequently.
2.	GP4CO, LC4CO and AE4CO are not well-explained. The exact method used lacks elaboration. Are you referring to the methods used in reproduced papers? If so, cite it, and at least provide a legend to denote which method corresponds to which paper. Otherwise, explain GP4CO, LC4CO and AE4CO on a different section. In table 11, the use of GP4CO method for different types, with each type corresponding to a different source, confused me. In contrast, in table 1, GP4CO is stated as yours, sometimes extending beyond the union of problems considered for past literature. But for GP4CO to be stated as your own method, you have to explain clearly how exactly you did the grouping of existing solvers and how you adapted existing methods to propose your own method, which there is no clear indication. This raises an ethical issue of ownership of GP4CO, which is the main reason why I am giving 1/6.
3.	There is some analysis of the results on the paper, which should have been on the abstract as those are key findings of the paper.
4.	From lines 37 to 76, even though bold subheadings are used, readability remains slightly hindered because new points are not on new lines.
5.	Typo in line 13: MCut problem duplicated and MCl missed out. The seven problems are, (read from the appendix), TSP, ATSP, CVRP, MIS, MCl, MVC and MCut.
6.	Line 95: ‘constraints’ is misspelled

**Strengths Contributions:**

1.	The motivation behind writing the paper is clearly written.
2.	Experiments are comprehensively conducted, and results are transparently reported in the appendix.

---

> ### Author Rebuttal · Authors · 2025-07-31
>
> Dear Reviewer pcUr,
>
> Thank you for your meticulous review and constructive feedback on our work. Below we address each of your concerns point-by-point.
>
> ---
>
> > **Q1: The paper has clear motivation, but novelty is lacking and new insights gained are brief. Paper shows effort to standardize evaluation protocols for fair assessment. Major weakness is lack of clarity in explaining the main contribution of the paper, especially on the proposed GP4CO, LC4CO and AE4CO solvers.**
>
> **A1:** We sincerely apologize for any confusion that our description of the contributions in this paper may have caused. We will elaborate on our main contributions from the two aspects below, hoping to resolve your doubts.
>
> * **Foundational Toolkit:** In recent years, numerous ML4CO studies have emerged. However, we observe a lack of unified evaluation standards across these works, reflected in the use of different test sets, evaluation metrics, and baseline comparisons. To address this gap, we introduce the first benchmark toolkit ***ML4CO-Kit*** specifically designed for the ML4CO field. This toolkit comprises several key components below aimed at standardizing evaluation protocols for a wide range of combinatorial optimization problems. ***Our goal is to establish a solid foundation that allows future researchers to concentrate on algorithm design, without the overhead of data generation, preprocessing, data I/O operations, or evaluation.***
>
>     * ``algorithm``: common post-processing algorithms.
>     * ``data``: common test datasets and our generated traning dataset.
>     * ``draw``: visualization of problems and solutions.
>     * ``evaluate``: evaluator for problems and solvers.
>     * ``generator``: data generation of various distributions.
>     * ``learning``: implemented base classes that facilitate method development for ML4CO.
>     * ``solver``: solvers' base classes and mainstream traditional solvers.
>     * ``utils``: general or commonly used functions and classes.
>
> * **Systematic benchmark:** Our work encompasses 7 major categories of COPs, for which we have extensively collected, organized, and curated a substantial body of related literature. In addition, we have open-sourced all relevant code and datasets. ***We believe that this effort provides a sufficiently comprehensive and practical benchmark for the community***. To address your confusion regarding GP4CO, LC4CO, and AE4CO, we first need to explain the tri-leveled ``paradigm-model-learning`` taxonomy our proposed.
>     * ``paradigm``: ***On the paradigm level, the focus is on how to invoke the model to determine the values of decision variables***. This can involve: **1)** invoking the model once to determine all decision variables (corresponding to GP), **2)** invoking the model to determine one decision variable at a time (corresponding to LC), **3)** or invoking the model multiple times to determine subsets of decision variables each time (corresponding to AE).
>     * ``model``: ***On the model level, the focus is on how to call the neural network to obtain prediction results.*** This can involve **1)** a single inference to get the prediction result (corresponding to OS), **2)** or leveraging the principles of generative methods, with multiple iterations of adding and reducing noise to obtain the prediction result (corresponding to Gen).
>     * ``learning``: ***On the learning level, the foucs is on how the model is trained to learn predictions.*** This can involve SL, UL, MAML, RL, etc.
>
>    We can see that these three concepts belong to three different levels, from top to bottom in a hierarchical manner. As shown in Table 1, we have used this tri-leveled taxonomy to categorize the collected representative methods into 9 categories. ***For ease of presentation, we have selected the highest level, paradigm, as their overarching category names, thus generating the three solver categories of GP4CO, LC4CO, and AE4CO***. Additionally, we would like to clarify several points here.
>     * As pointed out in line 104, the concepts of paradigms (GP, LC, AE, etc.) are ***derived from COExpander*** [1].
>     * Regarding Table 1, we would like to clarify that we have not proposed any entirely new methods. ***The label "ours" is used to indicate that these are methods we have reorganized and reproduced to distinguish them from previous methods.*** In fact, the implementation of any solver category marked as "ours" is derived from its corresponding past method. For example, GP4CO (GP-OS-SL) is a synthesis of methods 1 through 4. Moreover, you can see from the table that for the same category, such as GP4CO (GP-OS-SL), previous methods only covered two types of problems: TSP and MIS. In our reorganization and reproduction, ***we have extended the support of this category of methods to include ATSP, MCl, MCut, and MVC problems. This is precisely the significance and contribution of our work in creating a benchmark.***
>     * Regarding the issue of "with each type corresponding to a different source" in Table 11, ***this arises because we only referred to the paradigm when naming.*** Our original intention was to be concise, but it now appears that it may have caused you considerable confusion. In fact, it is the "TYPE" column that truly differentiates the various solvers. Additionally, it should be noted that ***the citations used in Table 11 correspond to the existing works in Table 1. If we have extended a particular work to more problems, we have also labeled it as that work.***
>
> We hope that our clarifications can resolve any confusion you may have regarding the contributions of our work and the ownership of the methods.
>
> [1] COExpander: Adaptive Solution Expansion for Combinatorial Optimization. ICML-2025
>
>
> ---
>
> > **Q2: Relative lack of analysis of results, confined to lines 257 to 281**
>
> That's a great suggestion. We believe the analysis regarding time efficiency is self-evident, and we will conduct the performance analysis here.
>
> * **Paradigm and Model.** Generally speaking, we would anticipate that the AE paradigm might exhibit better performance compared to the GP paradigm, due to its approach of utilizing multiple model invocations to secure a more efficient determination process. Furthermore, based on research such as Fast-T2T and DIFUSCO, it is commonly assumed that generative models tend to outperform one-shot models. However, it should be noted that ***there is actually no strict hierarchy of superiority or inferiority between different paradigms and different models.*** For example, the experimental results from MCl (Table 3) do not support this, and even more noise addition and subtraction steps brought negative performance impacts. ***The reasons for this are currently unknown to us. We present these findings as part of our results, hoping that future researchers will be able to provide an explanation.***
> * **Learning.** you can refer to our response **A4 to Reviewer zcew**.
>
> ---
>
> > **Q3: Paper presentation issues**
>
> Thank you for pointing out these minor issues. We have completed the modification. 1) line-13: "MCut" to "MCl"; 2) line-95: "constrains" to "constraints".
>
> ---
>
> > **Q4: However, the Algorithm Design Correspondence Table for DIMES (Qiu et al., 2022) and GOAL (Drakulic et al., 2025) do not tally with Table 1. This difference requires a justification.**
>
> **A4:** We sincerely apologize for not fully understanding the issue you've raised. We have carefully reviewed Table 1 and did not find any inconsistencies. If you could provide further clarification on this matter, we would be more than happy to address your concerns in subsequent discussions.
>
> ---
>
> > **Q5: Reproducibility may be an issue because the method of dividing into training and validation sets is not clearly mentioned. The authors have mentioned ‘freely divide’.**
>
> **A5:** Thank you for this important question!
> - Please be assured that we, in the first place, acknowledge the significance of reproducibility. To ensure credible replication, we have 1) open-sourced our division of validation datasets (in addition to the training and test sets which are already available), 2) provided all pre-trained weights for the neural models studied in this work, and 3) manually fixed a global random seed for both training and evaluation, confirming numerical stability among different executions.
> - As for the term "freely divide", we stated it to mean that in broad ML practices the division of training-validation-test data is virtually at the researchers' disposal given our generator APIs, while ours, as a standard benchmark, have all our experiments performed on fixed validation sets since the beginning with guaranteed reproducibility.
> - In summary, we regret any confusion the words might have caused and have updated the released assets to include our division of validation instances. Thank you for bringing this to our attention!
>
> ---
> Once again, we express our sincere gratitude for your time and review. We earnestly hope that our responses have adequately addressed your concerns and clarified possible misunderstandings. We would be truly appreciative if, with your valuable reassessment, deeper consensus could be reached regarding the overall contribution our work could have merited in this community. We look forward to your reply and any further discussions!
>
> Best regards,
>
> The Authors

---

> > ### Comment · Reviewer_pcUr · 2025-08-05
> > **Increase my score to 4.**
> >
> > The paper is well-motivated. Novelty of approach is lacking, but this is far compensated by the anticipated impact this will bring to future generations of researchers when benchmarking the work. The standardized benchmark is expected to bring in a fair comparison of the effectiveness of various proposed methods in different combinatorial problems, while saving researchers' time in generating synthetic datasets. While parts of the paper have been presented in a confusing manner, the severity is minor.
> >
> > The issues in the review have generally been adequately explained, apart from the issue that was confusing to you. The "Algorithm Design Correspondence Table" can be found on GitHub. In that table, the past method 'dimes' supports only TSP, and the past method 'meta_egn' supports MIS, MCl, MVC and MCut. But in Table 1, DIMES supports TSP and MIS, while Meta-EGN supports MCl and MVC, a different set of problems from the "Algorithm Design Correspondence Table". Does it mean that your implementation supports a different set of problems from the original papers, sometimes an extension of the original and sometimes a more simplified version of the original?
> >
> > I still have one more minor issue that I have missed earlier. In Table 2 MVC-RB-LARGE, OBJ.↑ should be OBJ.↓ since it is a minimum vertex cover problem.

---

> > > ### Author Response · Authors · 2025-08-05
> > > **The response regarding the "Algorithm Design Correspondence Table"**
> > >
> > > Dear Reviewer pcUr,
> > >
> > > Thank you very much for your recognition of our work, and we have already corrected the minor errors in the Table 2 that you point out. Below we will address the issue related to the "*Algorithm Design Correspondence Table*".
> > >
> > > >* **Table 1.** Firstly, Table 1 in the paper is compiled based on the ***original papers and the open-source codes (github)*** provided for each method. For example, ``DIMES``[1] method indeed supports TSP and MIS, ``Meta-EGN``[2] supports MCl and MVC, ``GOAL`` supports ATSP, CVRP, MIS, MVC. Besides, it should be noted that for ``Meta-EGN``, although the term "MIS" appears in the original paper, there is no explicit experimental table in the main body. Moreover, "MIS" does not appear in the open-source code either. Therefore, in the Table 1, we do not conclude that Meta-EGN support MIS.
> > >
> > > >* **Algorithm Design Correspondence Table.** In the source code we submitted, you can find that the "*Algorithm Design Correspondence Table*" (in ``README.md``) is a table ***corresponding to the code level.*** ***All the methods in this table are either reproduced or extended by us, corresponding to all the rows marked with "ours" in Table 1.*** For example, in the "*Algorithm Design Correspondence Table*," ``meta_egn`` supports MIS, MCl, MVC, and MCut. This means that based on the original implementation, we have extended it to support MIS and MCut.
> > >
> > > >* **The reason for causing confusion.** We apologize for the confusion caused. This is mainly related to our categorization. In Table 1, we have listed 9 solvers marked with "ours", among which ``GP-OS-MAML`` supports TSP, MIS, MCl, MVC, and MCut. In the code level, ``GP-OS-MAML`` is a combined result of ``DIMES`` and ``Meta-EGN``. In the code we submitted, ``DIMES`` is responsible for the TSP code support, while ``Meta-EGN`` covers the remaining support. In other words, since ``Meta-EGN`` already supports MIS, and both ``DIMES`` and ``Meta-EGN`` fall under the category of ``GP-OS-MAML``, we do not provide code for ``DIMES`` supporting MIS. However, Reviewer zcew points out that there is some issue with our categorization, and that ``DIMES`` and ``Meta-EGN`` should correspond to ``MAML(RL)`` and ``MAML(UL)``, respectively. ***Anyway, we will supplement the code for ``DIMES`` supporting MIS in the future.***
> > >
> > > [1] *DIMES: A Differentiable Meta Solver for Combinatorial Optimization Problems*
> > >
> > > [2] *Unsupervised Learning for Combinatorial Optimization needs Meta Learning*
> > >
> > > Hopefully our responses alleviate your concerns with satisfaction, and we stay open for any further interactions!
> > >
> > > Best regards,
> > >
> > > The Authors

---

### Official Review · Reviewer_HEtv · 2025-07-02

**Rating:** 5
**Confidence:** 4

**Summary:**

This paper presents ML4CO-Bench-101, a benchmark framework addressing the challenges of methodological fragmentation, implementation inconsistency, and evaluation irregularity in ML4CO. The authors categorize existing methods via a tri-level "paradigm-model-learning" taxonomy, integrate 34 datasets across 7 combinatorial optimization problems, and design unified solvers for Global Prediction (GP), Local Construction (LC), and Adaptive Expansion (AE). Key contributions include standardized evaluation protocols, reproducible implementations, and systematic analyses of generative models and decoding strategies.

**Dataset Code Accessibility:**

Yes

**Dataset Code Comments:**

The code and dataset can be found in https://github.com/Thinklab-SJTU/ML4CO-Bench-101

**Ethical Considerations:**

No, there are no or only very minor ethics concerns

**Final Justification:**

This paper focuses on combinatorial problems on graphs and integrates 7 mainstream problems into the benchmark, which holds high application value in the field. Given that the authors have supplemented the relevant rebuttal content, I recommend accepting this paper.

**Limitations Weaknesses:**

1. Datasets primarily use uniform, RB, and BA graphs, lacking complex structures (Holme-Kim, Watts-Strogatz).
2. The benchmark omits Divide-and-Conquer (D&C) and Learning to Search methods (e.g., GLOP/UDC), which excel in large-scale problems (e.g., JSSP), compromising comprehensiveness .
3. The framework focuses on traditional graphs, ignoring hypergraphs (e.g., multi-node edges in MIS). Compared to LLM4Hypergraph, it lacks support for high-order relations, restricting applications in social network community detection.
4. Why does Supervised Learning (SL) significantly outperform Unsupervised Learning (UL) in edge-oriented tasks (such as TSP and CVRP), while UL performs better in node-oriented tasks?

**Strengths Contributions:**

1. The tri-level taxonomy provides a unified framework, particularly the AE paradigm balancing efficiency and solution quality.
2. The benchmark covers 7 problem classes (e.g., edge-oriented TSP/CVRP and node-oriented MIS/MCut) across scales (TSP-50 to TSP-10K).
3. The released code (GitHub) and datasets (Hugging Face) include detailed model configurations (e.g., GCN layers, training hyperparameters) and post-processing steps (K-Opt/RLSA), aligning with NeurIPS reproducibility standards.

---

> ### Author Rebuttal · Authors · 2025-07-31
>
> Dear Reviewer HEtv,
>
> Thanks for your valuable recognition and conducive comments. Below we respond to your concerns in detail.
>
> > **Q1: Datasets primarily use uniform, RB, and BA graphs, lacking complex structures (Holme-Kim, Watts-Strogatz).**
>
> > **A1:** Thank you for this observation. Over the past few months, we have addressed the limitation of dataset diversity by expanding our datasets to include more complex structures (e.g., Holme-Kim [HK], Watts-Strogatz [WS]) and making them open-source. Below is a summary of the newly added datasets, with subsets used for generalization testing:
>
> >* **TSP:** Gaussian (50, 100, 200, 500), Cluster (50, 100, 200, 500)
> >* **ATSP:** HCP (50, 100, 200, 500), SAT (54, 102, 200, 507)
> >* **MIS:** BA (SMALL, LARGE), HK (SMALL, LARGE), WS (SMALL, LARGE)
> >* **MCl:** BA (SMALL, LARGE), HK (SMALL, LARGE), WS (SMALL, LARGE), Twitter, ER-700-800
> >* **MCut:** RB (SMALL, LARGE), HK (SMALL, LARGE), WS (SMALL, LARGE), ER-700-800
>
> > | Problem | Dataset | Baseline| Pretrain    | GP-OS-SL (greedy + mcts) |
> > | -- | -- | -- | ----------- |-- |
> > | TSP     | Gaussian-100 | Concorde, 34.03, 0.44s  | Uniform-100 | 34.10, 0.20%, 0.02s      |
> > | TSP     | Gaussian-500 | Concorde, 77.52, 19.95s | Uniform-500 | 78.22, 0.90%, 0.49s      |
> > | TSP     | Cluster-100  | Concorde, 5.53, 0.29s   | Uniform-100 | 5.538, 0.21%, 0.02s      |
> > | TSP     | Cluster-500  | Concorde, 10.72, 5.07s  | Uniform-500 | 11.050, 3.05%，0.68s     |
>
>
> > | Problem | Dataset  | Baseline              | Pretrain   | AE-Gen-SL (greedy)    |
> > | ------- | -------- | --------------------- | ---------- | --------------------- |
> > | MIS     | HK-SMALL | KaMIS, 79.37, 54.17s  | ER-700-800 | 78.79, 0.74%, 0.17s   |
> > | MIS     | HK-LARGE | KaMIS, 330.95, 67.27s | ER-700-800 | 327.81, 0.94%, 0.27s  |
> > | MIS     | WS-SMALL | KaMIS, 76.90, 51.49s  | ER-700-800 | 76.26, 0.84%, 0.18s   |
> > | MIS     | WS-LARGE | KaMIS, 262.57, 37.79s | ER-700-800 | 249.768, 4.88%, 0.23s |
>
> > These additions enhance the diversity of graph structures in our evaluations, with results included in the revised manuscript.
>
> ---
>
> > **Q2: The benchmark omits Divide-and-Conquer (D&C) and Learning to Search methods (e.g., GLOP/UDC), which excel in large-scale problems (e.g., JSSP), compromising comprehensiveness.**
>
> > **A2:** As noted in Appendix E, our work does not focus on Divide-and-Conquer (D&C) or Learning to Search (L2S) methods for two key reasons:
> > 1. **Complementarity.** Our approach is orthogonal to these methods and can integrate with them. For D&C, which emphasizes problem partitioning and subproblem solving, our framework can serve as an effective subproblem solver. For L2S methods (e.g., GLOP/UDC), which iteratively refine solutions via neural-guided actions, our work can provide high-quality initial solutions to accelerate their optimization.
> > 2. **Scope of focus.** The ML4CO community already has extensive research on D&C and L2S. Our work instead centers on core solution acquisition and post-processing optimization, aiming to clarify differences across paradigms, models, and learning approaches. Discussing D&C and L2S in depth would constitute a distinct line of inquiry, which goes beyond the scope of this paper.
>
> ---
>
> > **Q3: The framework focuses on traditional graphs, ignoring hypergraphs (e.g., multi-node edges in MIS). Compared to LLM4Hypergraph, it lacks support for high-order relations, restricting applications in social network community detection.**
>
> > **A3:** We acknowledge that our current framework focuses on traditional graphs and does not explicitly address hypergraphs or high-order relations (e.g., as in LLM4Hypergraph). That said, the core methods we propose may have transferable potential to hypergraphs. For instance, multi-node edge scenarios in problems like MIS can be transformed into equivalent formulations with nodes $\mathcal{V}$ and edges $\mathcal{E}$ in traditional graph terms, allowing our machine learning approaches to be adapted. Also, we recognize the importance of hypergraphs for applications such as social network community detection and view extending our framework to support high-order relations as a valuable direction for future work.
>
> ---
>
> > **Q4: Why does Supervised Learning (SL) significantly outperform Unsupervised Learning (UL) in edge-oriented tasks (such as TSP and CVRP), while UL performs better in node-oriented tasks?**
>
> > **A4:** This is a great question, and we will address it from the following three aspects.
> >* **Supervising Data Quality:** The importance of the quality of supervised data for supervised learning goes without saying. When the supervised data is weak, the quality of supervised learning is also affected. In the experiments of this paper, the solvers (Concorde, LKH, HGS) used for edge-oriented tasks have a high solving quality, while node-oriented tasks mainly use Gurobi for solving. It can be seen that when the number of nodes is high, even if the solving time limit is extended, the quality of the solution still drops significantly for Gurobi. As can be seen from Table 8, in many node-oriented problems, the ML4CO method easily outperforms Gurobi.
> >* **Uniqueness of the Solution:** The solutions to edge-oriented tasks are often unique, whereas solutions to node-oriented problems typically have many possibilities. This also affects the quality of SL.
> >* **Design of the Loss Function:**  Node-oriented problems can beell-suited for UL through the formulation of , nergy functions (see Appendix. A). For the 4 node-oriented problems addressed in this paper, their energy fu%nctions all incorporate the objective values of the original problems. Particularly in the case of MCut, there are no constraint terms, allowing the loss function to consistently decrease in a single direction. However, in edge-oriented problems, the number of decision variables is quadratically related to the problem size, which significantly increases the complexity. Moreover, due to more intricate constraint relationships, designing the loss function often requires optimizing multiple objectives, which can lead to poor training outcomes.
>
> ---
> We hope our point-by-point clarificaitons have satisfactorily addressed your concerns, while remaining fully committed to involving further discussions with you. Thanks again for your support!
>
> Best regards,
>
> The Authors

---

> > ### Author Response · Authors · 2025-08-08
> > **Looking forward to your reply**
> >
> > Dear Reviewer HEtv,
> >
> > Thank you very much for your recognition and support of our work! As the discussion period is drawing to a close, we are eager to know whether our responses have addressed your concerns. Additionally, we kindly remind you that the ``Mandatory Acknowledgement`` has still not been completed.
> >
> > Thank you once again for your support!
> >
> > Best regards,
> >
> > The Authors

---

### Official Review · Reviewer_ezor · 2025-07-03

**Rating:** 4
**Confidence:** 2

**Summary:**

This paper proposes ML4CO-Bench-101, a comprehensive framework for benchmarking machine learning methods on combinatorial problems. Specifically, the authors first split the works in this field into two categories: Algorithm Design and Heuristic Design, in which the former is further divided into Paradigm, Model, and Learning; while the latter is categorized into Decoding Strategies and Post-Processing Techniques. Under this framework, extensive experiments are conducted to validate the existing ML4CO methods on both synthetic and real-world datasets.

**Dataset Code Accessibility:**

Yes

**Dataset Code Comments:**

The benchmark is hosted on GitHub, where the authors provide a detailed README.md file to run their code.

**Ethical Considerations:**

No, there are no or only very minor ethics concerns

**Final Justification:**

The author's rebuttal addressed my concerns, and I will maintain my previous positive evaluation of the work.

**Limitations Weaknesses:**

(1) As pointed out by the authors in Appendix E, there are other common combinatorial optimization problems used in the literature (e.g. JSSP and CVRPTW), which are not covered by this paper.

(2) This is maybe a more general question on the scope of ML4CO. While the objective functions discussed here are all white-boxed and easy to evaluate, there are also other well-known combinatorial optimization problems in graphs where the objective functions are black-boxed or expensive to evaluate. For example, in Influence Maximization [1] and Opinion Dynamics [2], the goal is to optimize for the seed node subset such that the simulation result (i.e., influence) is maximized. Some recent works also use Bayesian optimization (BOCS [3], COMBO [4], GraphComBO [5]) to solve black-box combinatorial problems on graphs. It would be good if the authors could briefly clarify the scope of the current work.

Reference:

[1] Maximizing the spread of influence through a social network. KDD-2003

[2] Reaching a Consensus. Journal of the American Statistical Association

[3] Bayesian Optimization of Combinatorial Structures. ICML-2018

[4] Combinatorial Bayesian Optimization using the Graph Cartesian Product. NeurIPS-2019

[5] Bayesian Optimization of Functions over Node Subsets in Graphs. NeurIPS-2024

**Strengths Contributions:**

(1) This paper provides a detailed categorization of the existing ML4CO methods in the literature, which, in my opinion, may also serve as a survey paper to some extent.

(2) The proposed benchmark features a unified framework for testing ML4CO baselines, which largely addresses the inconsistent implementation problem across different works.

(3) The experiments are extensive, where the author discussed various “combinations” of conditions across different paradigms.

(4) The paper is also well-written from the following perspectives: the motivation is derived from the key limitations in the literature (as identified by the authors); the presentation is clear with sufficient task descriptions and intuitive figures; and the flow is smooth and easy to understand with a clear logic and framework.

---

> ### Author Rebuttal · Authors · 2025-07-31
>
> Dear Reviewer ezor,
>
> We appreciate your valuable recognition and constructive advice. Below we respond to the two major points mentioned in your comments.
>
> ---
>
> >**Q1: As pointed out by the authors in Appendix E, there are other common combinatorial optimization problems used in the literature (e.g. JSSP and CVRPTW), which are not covered by this paper.**
>
> > **A1:** We acknowledge that our current work focuses on 7 types of COPs, selected for their representativeness and extensive prior study. However, this is not a fixed scope, as we are actively expanding our research. Over recent months, we have enhanced the ML4CO-Kit (used in the ML4CO-Bench-101 project) to include additional problems, such as the ***Knapsack Problem (KP), Orienteering Problem (OP), Prize Collection Traveling Salesman Problem (PCTSP), and Stochastic Prize Collection Traveling Salesman Problem (SPCTSP), Job Shop Scheduling Problem (JSSP)***, with ongoing efforts to broaden coverage further.
>
> ---
>
>
> >**Q2: It would be good if the authors could briefly clarify the scope of the current work.**
>
> > **A2:**  Thank you very much for this suggestion. To clarify, our current work focuses on seven types of white-box combinatorial optimization problems (CVRP, TSP, ATSP, MCut, MCl, MIS, MVC). These problems are characterized by straightforward evaluation: each solution can be directly mapped to an exact objective function value. While for those hard-to-evaluate black-box COPs, such as the influence maximization problem, our work does not take them into consideration. In our view, for the evaluation of these COPs, we could consider Monte Carlo simulations, random walks, and even specialized simulation software. However, it should be noted that ***our work primarily focuses on how to obtain solutions to COPs, and the ease or difficulty of evaluation is not the main focus of our work.*** Setting aside the evaluation aspect, these COPs share commonalities in terms of problem modeling and solution acquisition. This is precisely what we hope to leverage by using these 7 types of COPs as representatives and further extending them to other COPs.
>
> ---
> Hopefully our responses alleviate your concerns with satisfaction, and we stay open for any further interactions!
>
> Best regards,
>
> The Authors

---

### Author Response · Authors · 2025-08-09
**Summary of Rebuttal**

Dear PCs, SACs, ACs, and Reviewers,

We would like to express our sincere appreciation for the reviewers' time and thoughtful feedback. As the author-reviewer discussion phase comes to a close, we would like to provide a general summary of the ***main reviews*** and outline the efforts we have made during the discussion phase.

---

### Reviewers ezor

>**Q1: To cover more COPs in the future.**

**A1:** Recently, we have enhanced the ML4CO-Kit (used in the ML4CO-Bench-101 project) to include additional problems, including ***KP, OP, PCTSP, SPCTSP, JSSP***, with ongoing efforts to broaden coverage further.


>**Q2: To clarify the scope of the current work.**

**A2** Our current work focuses on 7 types of white-box COPs, without considering the hard-to-evaluate black-box COPs, such as the influence maximization problem. ***Our primarily focus is on how to obtain solutions to COPs, rather than the ease or difficulty of evaluation.*** Setting aside the evaluation aspect, these COPs share commonalities in terms of problem modeling and solution acquisition. This is precisely what we hope to leverage by using these 7 types of COPs as representatives to extend our work to other COPs.

---

### Reviewers HEtv


>**Q1: More diverse datasets to support generalizability.**

**A1:** We have addressed the limitation of dataset diversity by expanding our datasets to ***include more complex structures*** (e.g., Holme-Kim [HK], Watts-Strogatz [WS], Guassian, Cluster) ***and making them open-source***.


>**Q2: The benchmark omits D&C and L2S methods.**

**A2:** Firstly, ***our approach is orthogonal and complementary to these methods***. For D&C, our framework can serve as an effective subproblem solver, while for L2S methods, our work can provide high-quality initial solutions to accelerate optimization. Secondly, the ML4CO community already has extensive research on D&C and L2S, and ***our focus is on core solution acquisition and post-processing optimization, aiming to clarify differences across paradigms, models, and learning approaches.*** Therefore, we think that in-depth discussion of D&C and L2S would go beyond the scope of this paper.

>**Q3: Ignoring hypergraphs.**

**A3:** ***The core methods we propose may have transferable potential to hypergraphs.*** For instance, multi-node edge scenarios in problems like MIS can be transformed into equivalent formulations with nodes $\mathcal{V}$ and edges $\mathcal{E}$ in traditional graph terms, allowing our machine learning approaches to be adapted. Also, we recognize the importance of hypergraphs for applications such as social network community detection and view extending our framework to support high-order relations as a valuable direction for future work.

>**Q4: Question about SL and UL performance.**

**A4:** We believe that the influencing factors include: ***the quality of supervised data*** (the solver Gurobi for node-oriented problems has relatively weaker performance), ***the uniqueness of the solution*** (the solutions of node-oriented problems typically have multiple possibilities), and ***the design of the loss function*** (the node-oriented problems can be transformed into the form of energy functions, which are more suitable for UL).

---

### Note · Authors · 2025-08-16

We truly thank the chairs and reviewers for their time and valuable feedbacks.

---

Our work is regarded as a ***unified framework*** (ezor, HEtv) for ML4CO, encompassing an ***extensive review of the literature***, ***comprehensive experiments*** (ezor, pcUR), and the provision of source ***code and datasets*** (Hetv, zcew).

Our work proposes a ***tri-level taxonomy*** that enables clear and proper comparisons both within and across different categories (zcew). Our work is also highly praised for its ***well writing*** (pcUR, ezor), with a ***clear motivation*** grounded in the limitations of existing literature (ezor).

Besides, Review zcew highly commends our emphasis on the importance of using a simple ***greedy decoding*** to remove the impact of any specific decoding method, agreeing that the varying decoding strategies used by different papers make comparisons even more difficult.

----

During the discussions, we addressed all concerns in detail. Key items:

1. **Expansion of the research scope and datasets.** We have enhanced the ***ML4CO-Kit*** to include ***KP, OP, PCTSP, SPCTSP, JSSP***, and expanded our datasets to include ***HK, WS, Guassian, Cluster***.

2. **Discussion on the limitations of the research content in the paper.** We believe our approach is orthogonal and complementary to D&C and L2S methods, and the core methods we discussed may have transferable potential to hypergraphs.

3. **Discussion on the experimental results.** We have supplemented the analysis, especially the comparison between SL and UL, from three aspects: ***supervised data, the uniqueness of the solution, and the design of the loss function***.

4. **Clarification of other confusion.** We have explained:
    * The scope of discussion in our paper: focus on how to obtain solutions to COPs, rather than the ease or difficulty of evaluation
    * The core contributions of our work: the first benchmark toolkit, a systematic benchmark for 7 major COPs with a tri-leveled taxonomy
    * The inconsistencies in the Algorithm Design Correspondence Table: We have revised the learning categories of ``DIMES`` and ``Meta-EGN`` to ``MAML(RL)`` and ``MAML(UL)``, respectively, following the suggestions of **Reviewer zcew**. We also promise to supplement the code for DIMES supporting MIS in the future.

---

After the discussion period, we are grateful that ***all reviewers have reached consensus recommending acceptance***. Our sincere thanks again to the chairs and reviewers!

---

### Decision · Program_Chairs · 2025-09-18

**Decision:**

Accept (poster)

**Comment:**

This paper introduces ML4CO-Bench-101, a comprehensive benchmark framework for machine learning for combinatorial optimization (ML4CO). It unifies evaluation across 7 canonical COPs (edge- and node-oriented), providing standardized datasets, reproducible solvers, and a tri-level taxonomy (paradigm–model–learning) that organizes prior methods into three solver families: GP4CO (Global Prediction), LC4CO (Local Construction), AE4CO (Adaptive Expansion). The benchmark covers 34 datasets, incorporates both synthetic and real-world instances, and standardizes decoding/post-processing.

All four reviewers converged on acceptance after rebuttal. The paper is technically solid and potentially impactful: while novelty is limited, the standardization and breadth of this benchmark are critical infrastructure for ML4CO research. Remaining issues (taxonomy debates, hypergraph extension, deeper analyses, D&C/L2S integration) are valid but not blockers—they represent natural future directions.

Camera-Ready Suggestions.

- Tighten the contribution statement to emphasize the benchmark (ML4CO-Kit + taxonomy + standardized evaluation), while clearly noting that GP4CO/LC4CO/AE4CO are categories, not new methods.

- Present taxonomy and solver mapping more cleanly (possibly move correspondence details to appendix/code).

- Include a short dedicated discussion on limitations: D&C/L2S scope, hypergraphs, taxonomy debates.

- Keep acronyms/acronym definitions consistent and ensure main results tables are easily readable.

Recommendation: Accept.